# A tighter constraint on Earth-system sensitivity from long-term temperature and carbon-cycle observations

Tony E. Wong [1]✉, Ying Cui[2]✉, Dana L. Royer[3] & Klaus Keller[4,5]

The long-term temperature response to a given change in $CO_2$ forcing, or Earth-system sensitivity (ESS), is a key parameter quantifying our understanding about the relationship between changes in Earth's radiative forcing and the resulting long-term Earth-system response. Current ESS estimates are subject to sizable uncertainties. Long-term carbon cycle models can provide a useful avenue to constrain ESS, but previous efforts either use rather informal statistical approaches or focus on discrete paleoevents. Here, we improve on previous ESS estimates by using a Bayesian approach to fuse deep-time $CO_2$ and temperature data over the last 420 Myrs with a long-term carbon cycle model. Our median ESS estimate of 3.4 °C (2.6-4.7 °C; 5-95% range) shows a narrower range than previous assessments. We show that weaker chemical weathering relative to the a priori model configuration via reduced weatherable land area yields better agreement with temperature records during the Cretaceous. Research into improving the understanding about these weathering mechanisms hence provides potentially powerful avenues to further constrain this fundamental Earth-system property.

[1] School of Mathematical Sciences, Rochester Institute of Technology, Rochester, NY, USA. [2] Department of Earth and Environmental Studies, Montclair State University, Montclair, NJ, USA. [3] Department of Earth and Environmental Sciences, Wesleyan University, Middletown, CT, USA. [4] Department of Geosciences, The Pennsylvania State University, University Park, PA, USA. [5] Earth and Environmental Systems Institute, The Pennsylvania State University, University Park, PA, USA. ✉email: aewsma@rit.edu; cuiy@montclair.edu

Understanding the relationship between changes in atmospheric carbon dioxide ($CO_2$) concentration and global surface temperatures has been a scientific quest for more than a century[1]. The current uncertainty surrounding this relationship poses considerable challenges for the design of climate change policies[2]. Of particular interest is the equilibrium response of global mean surface temperature to a doubling of $CO_2$ relative to preindustrial conditions, termed the "equilibrium climate sensitivity"[3] (ECS). The ECS is critical for mapping changes in radiative forcing, including $CO_2$ and other greenhouse gases, to changes in global temperature. ECS is based on "fast" feedback responses to changes in radiative forcing, including changes in water vapor, lapse rate, cloud cover, snow/sea-ice albedo, and the Planck feedback[4]. Even with detailed constraints from the instrumental period, ECS estimates based on the historic record alone are still subject to large uncertainties[5–8]. Based on the understanding of feedback processes, historical climate and paleoclimate records, a recent summary by Sherwood et al.[9] concluded that the most likely range (66% confidence) for the effective sensitivity (defined in terms of the 150-year temperature response to a quadrupling of $CO_2$ forcing in the context of their general circulation model experiments) is 2.6–3.9 °C. Similar to the ECS, the effective sensitivity does not include long-term feedbacks, such as ice sheets, vegetation, and carbon cycle (ref. [9] and references therein).

In contrast to the shorter-term ECS that responds to relatively fast feedback processes, consideration of longer-term responses offers a glimpse into the deep-time paleoclimate evolution of the sensitivity of the Earth-system temperature response to both fast and slow feedbacks. In particular, a deep-time perspective offers insight into the "Earth-system sensitivity" (ESS)—the long-term equilibrium surface temperature response to a given $CO_2$ forcing, including all Earth-system feedbacks[10]. Sherwood et al. estimate the ESS as their effective sensitivity multiplied by an inflation factor, $(1 + f_{ESS})$, where $f_{ESS}$ is sampled from a normal distribution with mean value of 0.5 and standard deviation of 0.25 (refs. [10,11]). A growing body of evidence suggests covariations in $CO_2$ and temperature during the last 420 million years (Myr; ref. [12]). This long-term record enables improved quantification of ESS and insights into factors affecting the climate response across a wide range of climate states, including both icehouse and greenhouse conditions[10,13–18]. This wide range of states and variations in temperature and $CO_2$ is also important to help distinguish the long-term climate signal from the noise.

Previous studies estimate ESS over geological timescales using varying combinations of global climate models, long-term carbon-cycle models, and proxy data for temperature and atmospheric $CO_2$. Royer et al.[19] combines a geochemical model and $CO_2$ proxies from the past 420 Myr, and concludes that ESS falls between 1.6 and 5.5 °C (95% confidence). In addition, during glacial periods, a given $CO_2$ forcing will lead to a stronger temperature change due to the land ice-albedo feedback. Thus, estimates of ESS that do not explicitly account for land-ice feedbacks will necessarily be higher than those that do. Arguments in (for example) Park and Royer[14] and Hansen et al.[18] support such a "glacial amplification" in ESS, giving 6 °C or more warming per doubling of $CO_2$. The former study uses model time steps of 10 Myr, so mechanisms such as orbital forcings, which operate on timescales of 10–100s of thousands of years, are averaged out and are not explicitly represented. Many studies suggest that ESS >1.5 °C is a general feature of the Phanerozoic[10,13,16,20], although these studies generally vary in the types of external forcings they consider and the confidence levels for the ranges they report. By assuming different sets of external feedbacks, forcings and (sea) surface temperatures, these previous studies report different kinds of Earth-system sensitivities[4]. It is therefore necessary to distinguish between various flavors of ESS[4,15]. For example, the geochemical model from Royer et al.[19] uses a form of ESS that computes the overall global mean surface temperature response by explicitly accounting for forcings from changes in $CO_2$, solar luminosity, and paleogeography. In the notation of Rohling et al.[4], this ESS is based on the specific paleoclimate sensitivity $S_{[CO2, geog, solar]}$. Krissansen-Totton and Catling[21] also account explicitly in their model for $CO_2$, solar, and paleogeographic forcing over the past 100 Myr, and compute a median ESS of 5.6 °C (3.7–7.5 °C 90% credible interval). By contrast, Anagnostou et al.[22] account explicitly for $CO_2$, solar luminosity, paleogeography, and land ice, and find ESS estimates varying from ~5–7 °C 53 Myr ago to about 2 °C 30 Myr ago. Following the argument above, we expect that the inclusion of land-ice feedbacks leads the ESS estimate of ref. [22] (based on $S_{[CO2, geog, solar]}$) to be lower than that of ref. [21] (based on $S_{[CO2, geog, solar, land ice]}$).

In long-term carbon-cycle models, many uncertainties stem from how $CO_2$ proxy data can best be used to improve estimates of carbon-cycle model parameters[14,19]. Specifically, the errors in proxy $CO_2$ data are often asymmetric, where it is typical for the upper error bound to be farther from the mean than the lower error bound[15,23]. In addition, there is a complex interactive relationship among the model parameters and their combined effect on modeled $CO_2$ concentrations. Previous assessments do not fully account for these model parameter interactions[15] or neglect the asymmetric error structure[21]. This raises the related questions of how these assumptions affect estimates of ESS, and which research has the greatest promise to reduce biases and constrain ESS, given this common model framework.

Here, we expand on previous work[14,15,19] by improving the uncertainty characterization of both proxy $CO_2$, surface temperature, and associated parameters in a commonly used long-term geochemical model. First, we consider interactions among model parameters via a Monte Carlo precalibration approach to account for uncertainties in the model parameters, and the surface temperature and $CO_2$ proxy data. We generate model ensembles that agree with $CO_2$ proxy data only, temperature reconstructions only, and both, by imposing constraints on the goodness-of-fit of the model simulations. This experimental setup allows us to characterize the ability of each source of information to better constrain estimates of model parameters, and to examine the correlations among model parameters and periods of bias in the model output. We demonstrate that improved constraint on the ESS model parameter $\Delta T_{2x}$ will result from both (i) improved constraint on the $CO_2$ and/or temperature hindcast and from (ii) using both $CO_2$ and temperature data to constrain estimates of $\Delta T_{2x}$.

## Results

**The GEOCARBSULFvolc model.** GEOCARBSULFvolc is a long-term carbon and sulfur cycle model that simulates atmospheric concentrations of $CO_2$ and $O_2$ based on mass and isotopic balance over the past 570 Myr. The GEOCARBSULFvolc model (henceforth, "GEOCARB") and its previous incarnations[24,25] have been widely used in previous studies (e.g., refs. [14,15,19,26,27]), and includes a version of ESS where the only independent radiative forcings are $CO_2$, solar evolution, and changing geography. In the notation of refs. [4,10], this ESS would be computed from the specific paleoclimate sensitivity, $S_{[CO2, geog, solar]}$. However, within GEOCARB and other such models (e.g., ref. [21]), an ESS model parameter links $CO_2$ radiative forcing to the associated temperature response, but also accounts internally for other forcings in computing the total temperature response to radiative forcing. In GEOCARB, the ESS model parameter $\Delta T_{2x}$ corresponds to the long-term temperature change resulting from doubling $CO_2$ relative to preindustrial levels, accounting for

changes in solar evolution and continental geography. GEOCARB assumes a linear increase in solar luminosity over time, corresponding to the parameter $W_s$, and uses results from general circulation model output to simulate the land temperature change, resulting solely from changes in paleogeography (GEOG; see Supplementary Fig. 1)[28,29]. Thus, appropriate choices for the ESS parameter within GEOCARB are influenced by the balance of forcing between $CO_2$, solar luminosity, and paleogeographic changes. For brevity, we will use $\Delta T_{2x}$ when referring to ESS within the GEOCARB model, and reserve the term "ESS" for discussion of Earth-system sensitivity more generally.

While other long-term carbon-cycle model choices are available[21,30,31], we focus on the GEOCARB model due to its extensive use as an inverse modeling tool for leveraging $CO_2$ proxy data to constrain ESS and other geophysical uncertainties[14,15,19,27]. The inverse approach generates model simulations using many different plausible values for $\Delta T_{2x}$ to determine which values for $\Delta T_{2x}$ are likely, given the (mis)match between the proxy data for $CO_2$ and temperature and model simulation output for these quantities. The model structure assumes that the atmosphere and ocean is a single system, where the weathering of organic-rich sediments and volcanic degassing deliver carbon to the atmosphere–ocean system, while carbon is lost via the burial of organic-rich sediments and carbonates[24,32]. The shape of the modeled $CO_2$ curve is well-characterized, with high values (>1000 p.p.m.) between 540 and 400 Myr and ~250 Myr[15], consistent with the lower solar luminosity in the early Phanerozoic[26].

There are 68 GEOCARB model parameters, of which 56 are constants and 12 are time series parameters. The constant parameters have well-defined prior distributions from previous work, and the time series parameters have central estimates and independent uncertainties defined for each time point[15]. Previous efforts to constrain the uncertainty in the GEOCARB model parameters relied on several important, but limiting, assumptions[15]. The prior distribution centers are held fixed in their Monte Carlo resampling strategy and only the widths are adjusted; if a parameter sample leads to model failure (e.g., through unphysical carbon or sulfur fluxes or unphysical $O_2$ or $CO_2$ concentrations), then the input range is considered unlikely and rejected. This resampling approach risks missing key parameter interactions, and propagating biases in the centers of parameters' distributions.

**Model configuration.** Our adopted GEOCARB model[15] is structurally identical to the model as presented in Royer et al.[15]. GEOCARB assumes a single ESS parameter (called $\Delta T_{2x}$ within the model and in previous studies[14,15,21]) for the past 420 Myrs of non-glacial periods, and includes a parameter (GLAC) to amplify the ESS during the late Paleozoic (330–260 Myr) and late Cenozoic (40–0 Myr) glacial periods. During glacial periods, the effective ESS within GEOCARB is then $GLAC \times \Delta T_{2x}$. The two stable states, glacial and non-glacial, for ESS within GEOCARB provide a simple representation of the type II state dependence described by von der Heydt et al.[33]. However, temporal variation in ESS within each of those stable states is not represented in GEOCARB[22,34]. Some previous modeling efforts have assumed a single value of ESS for multiple climate states (e.g., glacial and non-glacial)[19,21]. This will generally increase the uncertainty in the resulting scalar parameter estimate due to using a single parameter to represent a quantity that is changing over time.

We briefly discuss the temperature and carbon mass balance calculations within GEOCARB below, but further details on the GEOCARB model structure and parameterizations may be found in ref. [35]. The temperature in GEOCARB is computed as

$$T(t) - T(0) = \Delta T_{2x} \frac{\ln(\mathrm{RCO_2}(t))}{\ln(2)} - W_s \frac{t}{570} + \mathrm{GEOG}(t), \quad (1)$$

where $T(t)–T(0)$ denotes the global mean surface temperature at time $t$ (Myr ago) relative to present ($t = 0$) and $\mathrm{RCO_2}(t)$ is the mass of atmospheric $CO_2$ at time $t$ relative to present. The time series parameter GEOG describes the change in land temperature attributable to changes in paleogeography and the parameter $W_s$ accounts for the linear trend in solar forcing over time. We follow ref. [35] and assume a present (past 5 Myr) mean global surface temperature of $T(0) = 15\,°C$. In GEOCARB, a mass balance governs changes in carbon over time in the surficial system, as given in Eq. 1 of ref. [15]:

$$\frac{dM_c}{dt} = F_{wc} + F_{wg} + F_{mc} + F_{mg} - F_{bc} - F_{bg}, \quad (2)$$

where $M_c$ is the total carbon mass in the surficial system, $F_{wc}$ represents the flux due to weathering of calcium and magnesium carbonates, $F_{wg}$ represents the weathering flux of sedimentary organic carbon, $F_{mc}$ represents the degassing flux from carbonates, $F_{mg}$ represents the degassing from organic carbon, $F_{bc}$ represents the burial of carbonate, and $F_{bg}$ represents the burial of organic carbon. An associated carbon isotopic mass balance accompanies this as an additional constraint. Thus, the weathering processes ($F_{wc}$ and $F_{wg}$) are parameterized to capture the average balance among these carbon sources and sinks, assuming a steady state balance over the course of a 10 Myr time step. For modeling the long-term carbon cycle, no perturbations around the steady state can persist for >500,000 years[36], including for alkalinity.

**Parameter precalibration.** The essence of our precalibration method to fuse the GEOCARB model with data is to sample a large number of model parameter sets from their prior distributions—these are the a priori parameter values, taken before any data are fused with the model. Then, we rule out any combinations of parameters that yield simulations that do not agree well with the $CO_2$ proxy or temperature data, given their uncertainties. What remains are the a posteriori ensembles of parameters, including $\Delta T_{2x}$. We use Latin hypercube sampling to draw samples of the constant parameters and inverse Wishart sampling to account for uncertainty and autocorrelation in the time series parameters (see "Methods" and Supplementary Fig. 2). This method improves on previous GEOCARB-based ESS estimates by updating the centers of all parameters' distributions.

In this setting, precalibration is preferable to formal calibration methods (e.g., Markov chain Monte Carlo) to avoid potentially overconstraining the system with a large and diverse calibration data set. For example, data points with relatively lower uncertainty can dominate the goodness-of-fit measure, leading to poor agreement with the other data points. Here, the $CO_2$ data uncertainties scale roughly with $CO_2$ concentration, so we employ precalibration to avoid a low-$CO_2$ bias (see Supplementary Fig. 3). We establish a maximal $+/-1\sigma$ window around all of the time series data for each of temperature and $CO_2$. For the $CO_2$ data, we use the proxy compilation of Foster et al.[26], and for the temperature data, we use the Phanerozoic temperature compilation of Mills et al.[12]. As a goodness-of-fit measure, we use the percentage of time steps in which a model simulation is outside the range of the precalibration windows around the data, termed "%outbound" following Mills et al.[12]. We create ensembles of model simulations that match $CO_2$ data, temperature data, or both simultaneously by imposing limits of at most 50, 45, 40, 35, and 30% of time steps to be out-of-bounds (for a total of 15 main experiments). Unless otherwise stated, we present results for the

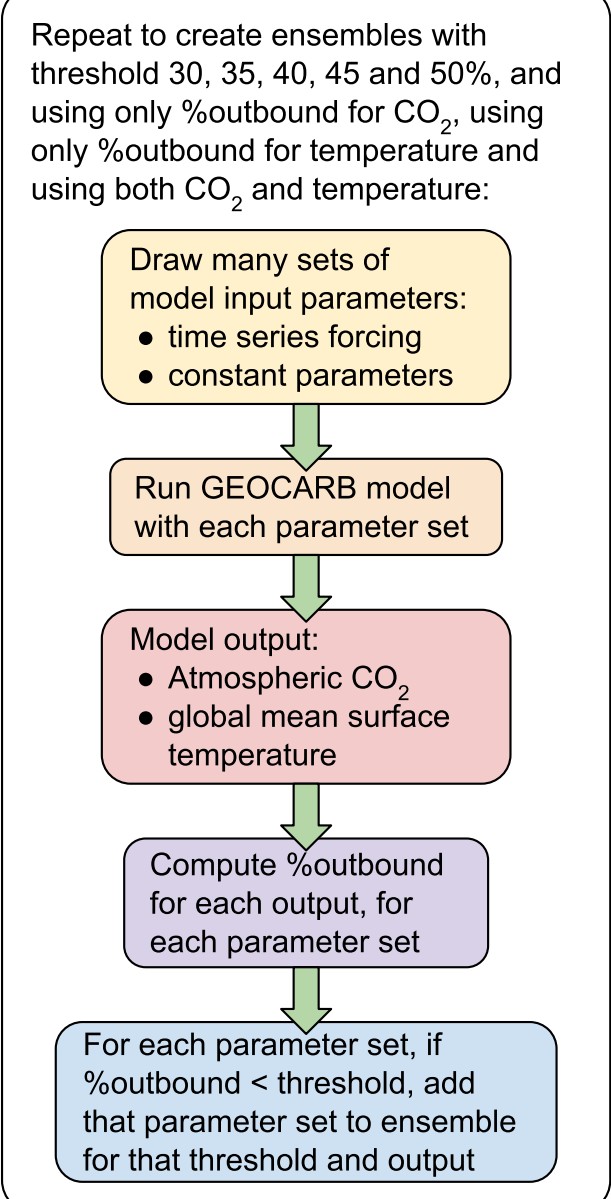

**Fig. 1 Schematic of the precalibration workflow.** This workflow is repeated to produce ensembles by varying the %outbound threshold and the data sets employed.

30 %outbound experiment, using both $CO_2$ and temperature data. Figure 1 gives a schematic depicting the precalibration workflow.

**Inference for Earth-system sensitivity**. We find an a posteriori ensemble median $\Delta T_{2x}$ of 3.4 °C per doubling of $CO_2$ (mean is 3.5 °C and 5–95% credible range is 2.6–4.7 °C; Fig. 2). Our estimates further improve constraint on the upper tail of the distribution for $\Delta T_{2x}$ from previous GEOCARB work: 2.8 °C (1.6–5.5 °C 95% confidence range) from Royer et al.[19] and 3.8 °C (1.6–7.6 °C 5–95% probability range) from Park and Royer[14]. We find 0.1% probability associated with non-glacial $\Delta T_{2x}$ >6 °C, in contrast to 16% in Park and Royer[14] ("PR2011" in Fig. 2).

The fact that the $\Delta T_{2x}$ estimate of Krissansen-Totton and Catling[21] (3.7–7.5 °C 5–95% probability range) is centered higher and has a wider uncertainty range than our study can be attributed largely to their selection of a single constant sensitivity

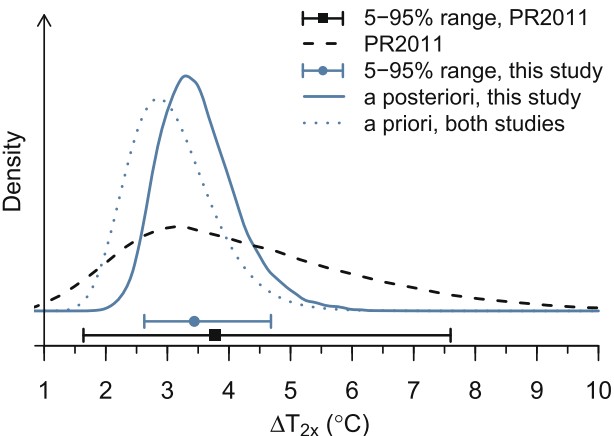

**Fig. 2 A posteriori probability density for Earth-system sensitivity parameter ($\Delta T_{2x}$, solid blue line).** The posterior density is plotted relative to the prior density (dotted line) and the results from Park and Royer[14] (PR2011, dashed black line). At bottom, the points provide median estimates from this study (blue circle) and from Park and Royer[14] (black square) and whiskers denote the 5–95% probability ranges.

value (see Supplementary Fig. 4). Our a posteriori estimates for the glacial scaling factor, GLAC, are centered at 2.1 (ensemble median; 5–95% credible range: 1.4–2.9), which is consistent with the central value of 2 used in previous work[14]. This leads to our estimated distribution for the net glacial period $\Delta T_{2x}$ to be centered at 7.1 °C (mean is 7.3 °C and 5–95% credible range is 4.4–11.0 °C). This result is centered slightly higher than the estimate of 6–8 °C from a previous GEOCARB analysis[14], although still within the uncertainty ranges for that and other glacial period $\Delta T_{2x}$ estimates[13]. Our results thus reconcile the distribution of ESS between estimates that place more probability weight <2.5 °C (refs. [4,14,19]) and the high-end estimates of Krissansen-Totton and Catling[21], whose posterior $\Delta T_{2x}$ values represent a mix of the glacial and non-glacial estimates presented here.

As we consider increasingly tighter bounds on acceptable $CO_2$ hindcasts without the use of temperature data, the corresponding constraint on $\Delta T_{2x}$ does not noticeably improve (Fig. 3). As we progressively tighten constraint on temperature hindcasts, however, the associated estimates of $\Delta T_{2x}$ become better-constrained: the uncertain ranges for $\Delta T_{2x}$ become narrower. This improvement is most prominent when $CO_2$ and temperature are used as complementary constraints (Fig. 3, bottom). In addition, the ensemble median estimate of $\Delta T_{2x}$ increases as constraint on paleo global mean surface temperature improves (Fig. 3, middle). Thus, two important related conclusions emerge: (i) temperature provides an important constraint on $\Delta T_{2x}$, in addition to $CO_2$, and (ii) improved estimates of paleotemperatures likely lead to tighter estimates of $\Delta T_{2x}$. These results highlight the importance of temperature data, in order to improve estimates of ESS more generally.

**Constraint of paleo $CO_2$ evolution**. We find that the assimilation of the $CO_2$ and temperature proxy data provides a tight constraint on the evolution of modeled paleoclimate $CO_2$ and surface temperature (Fig. 4). As expected, there is notable improvement in the simulation of paleoclimate $CO_2$ concentration when both temperature and $CO_2$ data are used for precalibration, as compared to when only temperature data are used (Fig. 4a). When we use only temperature data to constrain the model simulations, 10% of the 10,000 ensemble members are in agreement with the proxy $CO_2$ compilation at a %outbound level of 25% or better. By

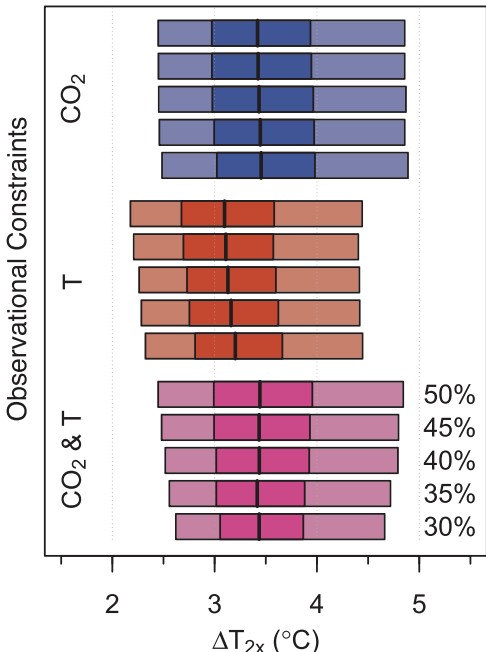

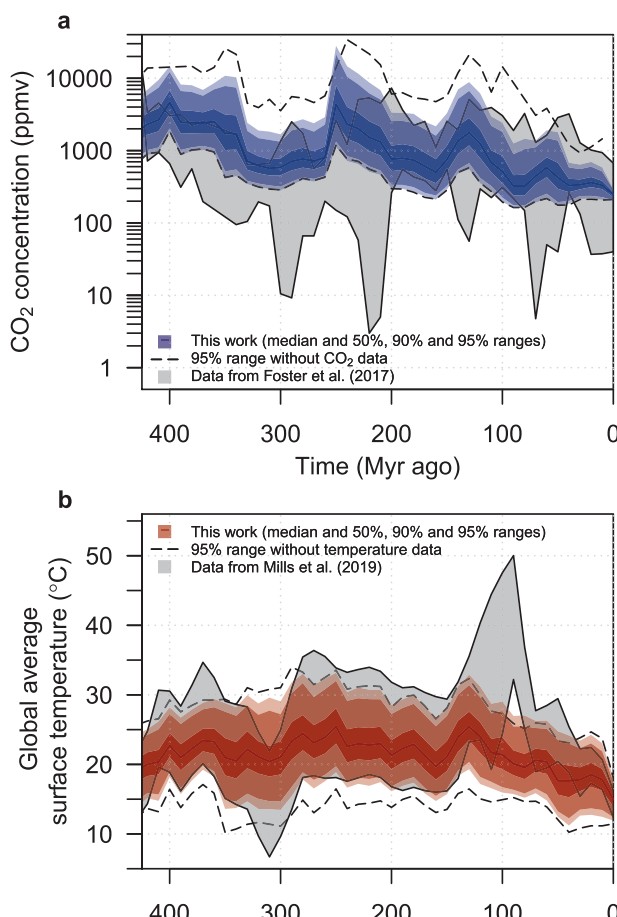

**Fig. 3 Medians and 50 and 90% credible intervals of the ESS model parameter, $\Delta T_{2x}$.** We use as observational constraints $CO_2$ only (top set of shaded blue boxplots), temperature only (middle set of shaded red boxplots), or both $CO_2$ and temperature (bottom set of shaded purple boxplots). Each boxplot shows the 5–95% (light shading) and 25–75% (dark shading) credible ranges and ensemble medians (solid lines). Within each set of boxplots, from top to bottom, the boxplots depict the credible ranges for the experiments using different thresholds for %outbound, starting with %outbound of 50% (top row in each set) and ending with 30% (bottom row in each set).

including $CO_2$ data in addition to temperature data, the number of simulations that agree at the 25 %outbound level or better improves to 85%. We focus on the 25 %outbound error level here because that is roughly the lowest error magnitude reported by Mills et al.[12] (c.f. Fig. 11 in that work). We also observe dramatic improvement in the temperature simulation: without temperature data, only 0.14% of the 10,000 ensemble members have an error <25 %outbound; by including temperature data in addition to $CO_2$ data, 3.8% of the ensemble members attain error margins <25 %outbound in temperature. While 3.8% seems like a low proportion of success, we note that (i) 25 %outbound in temperature is comparable to the best error margins for the tuned simulations of Mills et al.[12], and (ii) this constitutes an order of magnitude improvement relative to the model simulations that do not employ temperature data.

**Controls on Cretaceous temperature biases.** Despite this improvement in the match to paleotemperatures, it is still striking that it is so rare to attain error margins that are <25 %outbound for temperature. These results, taken together with the results from the work of Mills et al.[12], who also found it difficult to further improve on the temperature simulation, highlight the importance of examining the controls on paleotemperature within the GEOCARB model structure. Specifically, during the early Cretaceous (~100 Myr ago), both our results and those of Mills et al.[12] display a substantial cool bias in temperature relative to the proxies.

In light of these biases, we perform an additional sensitivity experiment to investigate the controls on early Cretaceous (140–90 Myr ago) temperature, using the GEOCARB model.

**Fig. 4 Model hindcast, using both $CO_2$ and temperature data, for precalibration and a %outbound threshold of 30% (shaded regions).** The gray-shaded regions show the data compilations for $CO_2$ (ref. [26]) and temperature[12]. The lightest colored shaded regions denote the 95% probability range from the precalibrated ensemble, the medium shading denotes the 90% probability range, the darkest shading denotes the 50% probability range, and the solid-colored lines show the ensemble medians. To depict the marginal value of each data set, the dashed lines depict the 95% probability range from the precalibrated ensemble, when only temperature data is used (**a**) and when only $CO_2$ data is used (**b**).

First, the point of this exercise is to examine the relationship between Cretaceous temperatures and the model parameters (in particular, the ESS parameter $\Delta T_{2x}$), so we relax the %outbound threshold from 30 to 50%. This change allows more variation in the model's temperature simulations. Later, after making further changes to improve the goodness-of-fit in the Cretaceous temperature simulations, we tighten the error margin back to 30%, to show that the GEOCARB model is indeed quite capable of matching well the Cretaceous temperature record. Our initial sensitivity experiment is similar to the 50 %outbound experiment from our main set of simulations, where the ensemble for analysis consists only of simulations that match the $CO_2$ temperature data windows in at least 50% of the time steps. In our new experiment, however, we retain only those simulations that pass through the temperature data window at 90 Myr ago. This time step was chosen because it corresponds to the peak in the temperature time series (Fig. 4b, gray-shaded region).

The Cretaceous-matching calibration experiment leads to an increase in the estimated distribution for $\Delta T_{2x}$ by ~0.2 °C relative to the original results for the 50 %outbound experiment (median

of 3.6 °C as compared to 3.4 °C in the original 50 %outbound experiment). We examine the distributions of model input parameters for the Cretaceous-matching experiment and find no substantial changes in any of the 56 constant parameters. However, several of the time series parameters' distributions change substantially. Specifically, we find that changes were required in the time series for the land area relative to present ($f_A$), global river runoff relative to present ($f_D$), the response of temperature change on river runoff (RT), and the fraction of land area that undergoes chemical weathering relative to present ($f_{AW}/f_A$). Not surprisingly, the main changes to these time series parameters occur primarily in the 90 Myr time step (see Supplementary Fig. 5). In order to match the Cretaceous temperatures during that time, we observe slight decreases in $f_A$, $f_D$, and RT 90 Myr ago. However, we observe a sizable decrease in $f_{AW}/f_A$ (the weatherable land surface area), which is not well-supported by paleoclimate modeling studies[28,29].

To remove the effect of arguably unphysical parameter choices, we generate a new set of 10,000 simulations that all match the Cretaceous temperature 90 Myr ago. We sample the time series parameters by changing the centers of their multivariate normal distributions to match the mean time series shown in Supplementary Fig. 5 (dashed lines). We revert to using the 30 % outbound threshold, in order to assess the degree to which our best ESS estimates (the 30 %outbound experiments) are influenced by biases in the Cretaceous temperatures, and to improve these estimates by accounting for both the Cretaceous temperature bias and the plausibility of forcing parameter values. By restricting our set of simulations to only those in which the $f_{AW}/f_A$ time series does not stray too far from its original central value, our updated set of Cretaceous-matching simulations has a median $\Delta T_{2x}$ of 3.3 °C, as compared to 3.4 °C in the original set of experiments. The 5–95% probability range also shifts ~0.1–0.2 °C cooler at 2.5–4.5 °C, as compared to 2.6–4.7 °C in the original 30 %outbound experiments (Fig. 5). From the fact that the distribution of estimated $\Delta T_{2x}$ changes by <0.2 °C, we conclude that our estimates of ESS are not unduly influenced by biases in the temperature simulation. Further, we conclude that GEO-CARB is indeed capable of matching the temperature data, although these results highlight that sampling via brute force Monte Carlo requires a very large number of samples and some

statistical care is needed, in order to bring the modeled and proxy temperatures into better agreement.

## Discussion

We make a number of improvements relative to previous work using the GEOCARBSULFvolc model[14,19], which reduce the ESS uncertainty compared to these previous studies[14]. This change can be explained by our improved calibration approach and our use of temperature data in addition to $CO_2$. Specifically, we find that a constraint on paleotemperature is critical for tightening our estimates of the GEOCARB ESS parameter, $\Delta T_{2x}$; reducing the uncertainty surrounding paleo $CO_2$ concentrations on its own is not sufficient. In addition, we include a larger $CO_2$ proxy data record[26] and conduct a set of sensitivity experiments to analyze the parametric controls on simulated $CO_2$ concentrations and global mean surface temperatures. Our results refine the characterization of the Earth-system surface temperature response to changes in atmospheric $CO_2$ concentrations and can provide guidance on where to focus future research to better understand and quantify this relationship.

We adopt a well-studied, state-of-the-art, yet still relatively simple model. This model simplicity provides the advantages of transparency and the ability to perform careful and exhaustive uncertainty and sensitivity analyses[37]. These advantages come, however, with several caveats that point to fruitful research directions. One key caveat stems from the fact that GEOCARB is a coarse-resolution and highly parameterized model with a long (10 Myr) time step and many (68) parameters (including 12 time series). A second related caveat arises from the still highly stylized representation of feedbacks and processes that is characteristic of such models (e.g., refs. [21,30]). As previously discussed (e.g., refs. [12,22,33]), the current assumption in the model of using a constant $\Delta T_{2x}$ for each of the glacial and non-glacial stable climate states risks missing processes leading to gradual changes in $\Delta T_{2x}$ within one of the larger stable climate states. The work of ref. [12] further points to the potential importance of capturing this type I state dependence in $\Delta T_{2x}$, because their results indicate an increasing trend in $\Delta T_{2x}$ beginning ~130 Myr ago. In the GEO-CARB model, however, the $\Delta T_{2x}$ ESS parameter is assumed to be constant at its non-glacial value from 260 to 40 Myr ago, then shifts immediately to its glacial value from 40 to 0 Myr ago. We evaluate the impacts of this type I state dependence in an experiment, where we linearly increase $\Delta T_{2x}$ from its non-glacial value 130 Myr ago to its glacial value 40 Myr ago; the parameter remains constant at its glacial value from 40 to 0 Myr ago. This linear change in $\Delta T_{2x}$ (as opposed to the step function transitions in the base-case version of the model) has little effect on the temperature hindcast (Supplementary Fig. 6). This simple experiment, of course, scratches only the surface of the challenge to represent type I state dependence for ESS. This result suggests, however, that a simple refinement of type I state dependency does not substantially impact our results.

The assumed time series of forcing parameters may also introduce biases. For example, uncertainty in paleogeographical changes, such as the opening of the Drake Passage, while not explicitly represented in the GEOCARB inputs or processes, indeed contributes to uncertainty in such parameters as GEOG (the temperature change resulting from changes in paleogeography, assuming fixed $CO_2$ and solar luminosity). In addition, GEOCARB does not explicitly account for non-$CO_2$ greenhouse gases or aerosols. This limitation of GEOCARB and other similar models (e.g., ref. [21]) may risk overestimating $\Delta T_{2x}$ by assuming that all of the observed temperature change is attributable to the $CO_2$ forcing (along with paleogeography and solar luminosity in the case of GEOCARB). However, our experiment examining the

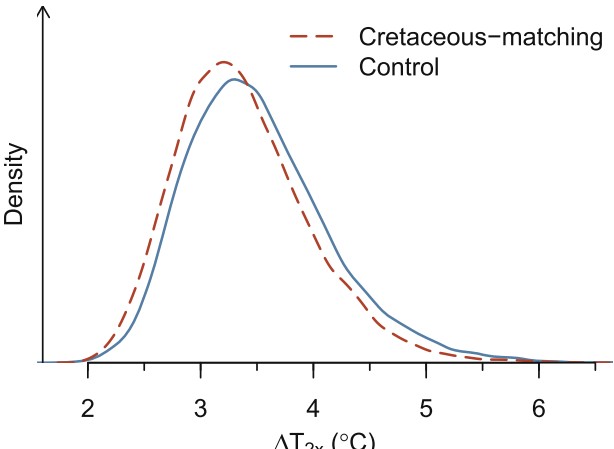

**Fig. 5 A posteriori probability density for Earth-system sensitivity parameter ($\Delta T_{2x}$).** Shown are the densities for the original calibration experiment using the 30 %outbound threshold and both $CO_2$ and temperature data (solid blue line) and the experiment, where the simulated temperature is forced to agree with the data compilation of ref. [12] in the 90 Myr ago time step (dashed red line).

Cretaceous cool temperature bias suggests that our estimates of ESS are robust to these variations associated with improving the Cretaceous temperatures.

Our use of a precalibration approach to avoid overfitting data points with low-$CO_2$ concentrations minimizes the low-$CO_2$ bias found throughout the Mesozoic Era characteristic of previous GEOCARB analyses[14,15]. Indeed, when we fit a mixture model distribution to the $CO_2$ proxy data, this distribution reveals strong multimodality in the $CO_2$ proxy record (see Supplementary Fig. 7). This multimodality is a likely culprit for the low-$CO_2$ bias observed in previous work[14,15], as formal calibration procedures (e.g., Markov chain Monte Carlo[21]) may improve the model fit to the data by tuning the model to better represent modes in the data that have narrower uncertainty ranges at the expense of adequately representing data points with higher uncertainties (Supplementary Fig. 7a).

We find that the efficiency of chemical weathering, as modulated by weatherable land surface area and riverine discharge to oceans, offers an avenue to improve the representation of paleotemperature in GEOCARB. Given the important role of temperature in obtaining better-constrained estimates of $\Delta T_{2x}$, this highlights the importance of these weathering mechanisms for constraining ESS, thereby improving our understanding of the relationship between atmospheric $CO_2$ concentrations and changes in Earth's climate.

## Methods

**Parameter precalibration**. We use the parameter means and uncertainty ranges given by Park and Royer[14]. There are 68 parameters in total: 56 constant parameters and 12 time series parameters. The time series parameters include isotopic ratios for strontium ($^{87}Sr/^{86}Sr$, to track the weathering fraction of volcanic rocks), carbon and sulfur isotope ratios ($\delta^{13}C$ and $\delta^{34}S$, to track burial, degassing, and weathering fluxes); paleogeographical factors (including continental relief, total land area, land area susceptible to weathering, land area covered by carbonates, river runoff, and the effect of paleogeographical changes on temperature); and degassing and seafloor spreading. The parameters are described along with their prior and posterior ranges in the Supplemental Material accompanying this work, and in much greater detail in Royer et al.[15]. The essence of any Bayesian calibration scheme is to update our a priori beliefs about probable parameter values in light of the available data. Our a priori beliefs about the parameters' probable values and their uncertainties are characterized by assigning the parameters prior distributions. The constant parameters are assigned Gaussian prior distributions, with the exception of the Earth-system sensitivity parameter, $\Delta T_{2x}$, which we assign a log-normal prior distribution[14]. Each of the time series parameters takes on distinct values at each of the 58 model time steps. Following previous work, we assume the model and forcing time series parameters are in steady state between model time steps[14]. Each time series parameter is sampled from a 58-dimensional (number of time steps) multivariate normal distribution, whose mean is taken to match the central estimates from previous work[15]. The covariance matrix for this multivariate normal distribution is sampled from an inverse Wishart distribution. We choose the degrees of freedom for the inverse Wishart distributions such that the widths of the prior distributions match those from Royer et al.[15]. We update the time series for seafloor spreading rate ($f_{SR}$) to match the more recent work of Domeier and Torsvik[38], and evaluate the sensitivity of our results to this improvement in a set of supplemental experiments (see Supplementary Fig. 1). In our adopted GEOCARB model, we have fixed an error that was noted in previous GEOCARB versions[27], wherein the forcing time series for the fraction of land area that undergoes chemical weathering relative to present (the parameter $f_{AW}/f_A$) was previously not normalized to 1 relative to the final model time step (which roughly represents present-day conditions).

**Model–data fusion**. Using the $CO_2$ proxy data set as in Foster et al.[26], containing 1215 proxy data points, we first discard two data points with unphysical negative $CO_2$ concentration values. For each model time step (10 Myr) we construct a precalibration window as follows (see Supplementary Fig. 8). We pool all data points within 5 Myr of the given time step's center. We compute the upper and lower $1\sigma$ bounds on each of the data points within the given time step. From the set of upper $1\sigma$ bounds, we take the maximum as the upper limit of the precalibration window for this time step. Similarly, we use the minimum of the data points' lower $1\sigma$ bounds for the lower bound for each of the windows. Any time steps that have no $CO_2$ proxy data points within them are assigned a window of 0–50,000 p.p.m.v. $CO_2$ (ref. [15]). For paleoclimate global mean surface temperature reconstructions, we use the reconstruction of Mills et al.[12]. The gray-shaded regions in Fig. 4 correspond to the time series of precalibration windows. We measure a model simulation's goodness-of-fit to the proxy data using the percentage of time steps, in which the model hindcast time series is outside of the precalibration windows around the data, termed "%outbound" following Mills et al.[12]. We use thresholds of %outbound varying from 30 to 50%, in order to evaluate the impacts of improved fit to the data. As examples, a %outbound threshold of 100% amounts to sampling from the prior distributions, and a 0 %outbound threshold requires the model simulations to go through all of the precalibration windows. For each of the % outbound thresholds between 30 and 50% (in increments of 5%), we generate model ensembles that agree with $CO_2$ proxy data only, with temperature reconstructions only, and both data sources.

**Parameter sampling**. We use a Latin hypercube approach to sample from the prior distributions of the model parameters, and use the precalibration windowing procedure described above to cull the prior samples down to only those that match the data (temperature, $CO_2$ or both) to within the desired %outbound threshold. We use an initial sample size of $2 \times 10^7$ parameter sets, but cease sampling once we achieve at least 10,000 samples that are within the %outbound threshold for the given experiment. Experiments adjusting the final sample size confirmed that our a posteriori estimates of $\Delta T_{2x}$ are insensitive to changes in sample size beyond ~1000 samples (see Supplementary Fig. 9).

In our experiment examining the Cretaceous temperature bias, we sample the time series parameters by changing the centers of their multivariate normal distributions to the a posteriori means from a set of simulations that are forced to agree with the temperature data at the 90 Myr ago time step. We retain only the plausible simulations in our experiment by removing any simulations where the value for the $f_{AW}/f_A$ time series at 90 Myr ago was more than one standard deviation away from its original central value. This leaves 2139 simulations out of the original 10,000.

## Data availability

All input data sets are provided with the model codes and are freely available from https://doi.org/10.5281/zenodo.4562996. Model output results files used for analysis are freely available from https://doi.org/10.5281/zenodo.4563019. All are provided under the GNU general public license.

## Code availability

All model codes and analysis codes used for analysis are freely available from https://doi.org/10.5281/zenodo.4562996, and are distributed under the GNU general public license. Large model output data sets are linked in the "Data availability" section.

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

## Acknowledgements

This work was co-supported by the National Science Foundation through the Network for Sustainable Climate Risk Management (SCRiM) under NSF cooperative agreement GEO-1240507, the Penn State Center for Climate Risk Management, and the RIT College of Science Dean's Research Initiation Grants program. Y.C. thanks support from NSF award #1603051, the National Science Foundation of China (Grant #41888101) and travel support from RCN NSF award #OCE-16-36005 to Bärbel Hönisch and Pratigya Polissar. Any opinions, findings, and conclusions or recommendations expressed in this material are those of the authors and do not necessarily reflect the views of the funding entities. We thank Nathan Urban, Irene Schaperdoth, and Benjamin Mills for their contributions.

## Author contributions

Y.C., K.K., D.L.R., and T.E.W. contributed to the study design; T.E.W. carried out the experiments; T.E.W. and Y.C. wrote the first draft of the manuscript; and Y.C., K.K., D.L.R., and T.E.W. contributed to the final version of the manuscript.

## Competing interests

The authors declare no competing interests.
