## [Peer Review File · Nature Communications]

Reviewers' comments:

Reviewer #1 (Remarks to the Author):

[see comments page 6]

Reviewer #2 (Remarks to the Author):

Review Wong et al. "Evidence for higher ESS from long-term carbon-cycle observations"

Let me start off with this: I think this is a novel study (while at the same time well-founded in previous work, which underlines its relevance) that has strong potential, and an important message. Eventually, I would like to see this published, and Nature Communications is a good outlet for it, with a broad readership. I will dig deep in the following, but this should not be seen as an attempt to reject the paper. On the contrary, I'm hoping to expose where the paper needs some extra thought, or where it needs extra clarity or detail to make it stand-alone and accessible to a broad readership. If/When I get things completely wrong, this likely is the case because I couldn't follow or reproduce in my head what was being done; so, in those cases, better explanation is needed. Also, it seems to me that a structural rethink (maybe with a flow diagram) is needed to bring out exactly what the model starts with (and show this "master" time series), what it then does, and what is diagnosed for this study (CO₂, T, ESS). Then, a better/clearer comparison of those outputs with proxy data is warranted, including more thought on what mismatches may mean for the study's conclusions – that level of critical (self-)assessment seems to be too weak to me.

So, I think the following aspects need in-depth attention:

First, the paper contains more technical jargon than it really needs to. I suggest going through it with a fine comb to minimise it to what is absolutely necessary. Bear in mind, that you're aiming for a broad readership. A good example (but not the only one) of excessive jargon is in lines 93-99, where "prior distribution centers in the Monte Carlo resampling strategy" are mentioned before the Bayesian approach is even introduced, followed by some arcane sentences that make sense only to someone who has operated the specific model referred to. Such passages need to be written in a more accessible manner. Other examples are the captions of Suppl. Figures 1 and 2 – they are so terse that they are entirely inaccessible. And the end of line 180: "including the assumed time series of isotopic forcing"; what does that mean? Which isotopes (carbon I presume). Which forcing? And how is a time-series assumed? Is it based on data, or just made up? Please clarify in less terse and more intelligible terms.

The paper also is presented too much in a manner that assumes that people know as much as the authors about the exact setup of the experiments. For example, tables of parameters are given, but not the values used, or the uncertainties. It is mentioned that values and uncertainty ranges were used, but the reader is nowhere shown what these values and ranges are in the experiments. That makes the work essentially unreproducible, and that's not good enough in my view. Parameters are mentioned in the text using their acronyms, but these are not always introduced/spelled out. This ought to be added, and/or reference needs to be made in appropriate places to the tables of parameter explanations.

And the paper migrates too freely (in a non-specific manner) across terms like ESS and climate sensitivity, especially in the first several paragraphs. Ranges are compared between studies, where some are ESS, and others determined over centuries and therefore clearly NOT ESS, but more like Equilibrium Climate Sensitivity (ECS) (lines 34-43). This level of non-specific description and comparison is confusing. For Figure 1, a supplementary table is needed that documents exactly what values are used (e.g., some studies determined SST, not mean global surface temperature, so that a correction must have been made to generate Figure 1, but no details are

provided). In short here, the terminology needs to be sorted out very carefully throughout the manuscript. And composite figures like Fig 1 need to be backed up with a supplementary table that documents exactly how the plotted values were obtained. It is very important that this is checked carefully to ensure that the plotted values are indeed ESS (not ECS), and that any conversion that were made between the two are documented in the supplementary table. This is all too "loose" at the moment.

Similarly, I think that the model code (including all modifications such as GLAC) needs to be made publicly available, and if it is available somewhere already, then a URL needs to be provided.

Second, I would like to see the temperature values used in this study to calculate ESS. I think they're all just obtained from the model somehow. But how do they compare with data? And then, what is the confidence in the temperature responses, and how does the model determine this? Also, how good is the temperature determination from the model on a structural level; for example, does it assume constant and similar efficacies between all radiative influences to determine temperature (i.e., how does the model cope with feedback efficacy changes and/or potential non-linear addition of feedback processes)? In the complete absence of discussion on this aspect, and especially of the rather large uncertainties that are likely associated with inferred T changes and/or their comparison with data (not shown), I am uncomfortable with the conclusions as they stand. It may well be that all of this is covered and perfectly in order. But I am given absolutely nothing to judge it by. Importantly, simple referring to a bunch of previous studies when revising will not be good enough; the arguments should be at least synthesised in the supplement, so that all critical information is brought together nicely for the reader. This should not be too onerous, even if should be done in a transparent manner that is well-referenced and presented with as little jargon as possible.

Third, and this comes back a bit (in different wording) to my second point, I remain confused about the serious amount of mismatch between the data and the model hindcast for CO₂ (Figure 3), which is also visible throughout Supplementary Figure 1. The model hindcast seems to strongly underestimate major, dense clusters of data. Yet this lack of agreement is not discussed. The study seems to simply proceed with the model hindcast. Does this reduced variability in hindcast CO₂ relative to the data not risk underestimating ESS ($\Delta T/F_{2xCO_2}$)? Lines 113-116 give us some vague inkling on what's happening: "the essence of the calibration method is to iteratively sample model parameters, and "steer" the sampling toward values of parameters (including ESS) that yield simulations that agree well with the CO₂ proxy data." This suggests to me that the model effectively varies (for calibration) a bunch of parameters (as is well described, and also in the tables), which include ESS. It then uses the calibration to determine which CO₂+Temp histories best agree with the "assumed time series of isotopic forcing", which is mentioned only as late as Line 180. In this sense, it appears to me that it is the carbon isotope record that the model really tries to fit, and that CO₂, ESS, and T histories are what's diagnosed (and compared with the CO₂ proxy data; but why not also compared with T data?). This – or the correct description, if I got it wrong – really, really needs explaining. At the moment, the paper is like a bowl of spaghetti that twists and twirls around without apparent beginning or end. Moreover, where can I see the model's grand master time series, the "assumed time series of isotopic forcing"? It's downright strange not to show such a crucial piece of information, and to not explain (briefly) how it was developed.

If the above is a reasonable approximation of what the model in its most basic form does, then CO₂ and T are per definition assumed to be highly correlated in the model, with only the glacial parameter (GLAC) imposing a change in time in that correlation. This brings me to a Fourth issue: such an imposed correlation may be flawed because there are two types of potential state dependence in climate sensitivity, and I am concerned that the model may be wrapping potentially (strongly) state-dependent estimates over a very long period of Earth history into one ESS range. The GLAC parameter helps only to some extent with this problem. GLAC deals with so-called Type II state dependence (von der Heydt et al., 2016 Current Climate Change Reports; a study that really needs to be incorporated/addressed in some detail in the present work), which concerns bifurcations/multiple stable states issues between glaciated and non-glaciated states. However, the other type (Type I) of state dependence of von der Heydt et al (2016) is not considered at all well, in my view. Type I concerns the potential for changes in feedback efficacies, which can be gradual, or step-wise. For example, there may be water vapour and shortwave cloud feedback

efficacy jumps in warm states (Caballero and Huber 2013 PNAS; Zhu et al., 2019 Science Advances; and refs therein), or ice-albedo efficacy decreases under increasing cold states (Stap et al., 2019 Earth System Dynamics; and refs therein). Such Type I efficacy changes would sit folded within the warm state (warm versus very warm) or cold state (cold versus very cold). This would cause changes/spread in the diagnosed ESS for warm states and cold states because the temperature responses to CO₂ changes would be strengthened or weakened. One might say: "so the estimates found for warm and cold states can be further refined/narrowed by looking in detail inside them." But that would be true only if the amalgamated warm or cold state values were randomly representative for the entire warm or cold range; i.e., if the information presented in the current manuscript had no bias at all into any direction within the warm state (e.g., toward very warm, or just warm), or within the cold state (e.g., toward very cold, or just cold). And I am not convinced about that. The model CO₂ seems to sit systematically below major clusters of proxy CO₂ in several intervals, suggesting that the model ESS may be systematically biased to low-forcing states in those intervals.

Fifth, I sense that the sensitivity tests discussed around Figure 4 are all about instantaneous relationships within given time slices. However, for some of the slower processes, would you not expect the correlations to strengthen with some appreciable amount of time-lag? Could lagged correlations be included in the sensitivity tests?

I think a lot of my problems/confusions come from the opaque presentation of the paper, which (1) does not outline how the model does what it does, (2) does not present the "master timeseries" from which the model through all its manipulations determines T, CO₂ and ESS, (3) does not carefully and consistently use terminology throughout; (4) obscures important information by using unnecessary jargon; (5) does not rigorously discuss the large offsets between model hindcast CO₂ and proxy CO₂, and how these mismatches may affect the conclusions; and (6) shows no temperature outputs with comparison to proxy data (or analysis of differences between these). All of this can be fixed. And then this could be a really important contribution.

Finally, there is a really weird statement in line 169: "further investigation is outside the scope of this study." Really? This statement is about overconfidence in the model's CO₂ reconstruction, and offsets from proxy data. Those matters could blow out the reconstructed ESS ranges by quite some margin. I would argue that some of this comes into the Type I state dependence sector. If the authors wish to present believable ESS ranges, then this is exactly what should be tackled in the present study – it cannot be put off to a next study as it is fundamental to the ESS ranges reported, which are the central conclusion of the study.

A most final niggle: The reference list is a bit of a mess, and needs consistent formatting, with correct journal abbreviations, and with the "2" in CO₂ and O₂ written in subscript. Also, in Ref 21, eocene should be capitalised.

=====

Eelco Rohling

Reviewer #3 (Remarks to the Author):

This paper describes a nice approach to fitting a long-term carbon-cycle model to Phanerozoic CO₂ estimates while varying a multitude of parameters in a MCMC procedure. Included in those parameters is the Earth System Sensitivity (ESS - the long term climate response to a doubling of carbon dioxide levels), and so a posterior pdf on ESS can be constructed. Overall this is worthwhile and interesting, but there are a few issues that could be better addressed.

1) There is an overall lack of clarity in what is being done (or rather, how the study is framed). ESS is not dependent on the carbon cycle - it is the theoretical long-term response to fixed CO₂ change. The work here uses variations in the CC to get a better match to the inferred CO₂ levels that then allow a best guess ESS to be determined. Thus this procedure to constrain ESS depends

on the CC, but the ESS sensu stricto does not. There are multiple places where the text seems to imply there is a direct influence of the carbon cycle on ESS.

2) There are a few issues with the model setup that I think could be more usefully explained:

i) Why is the response to solar insolation (W_s) different from ESS? (in some suitably scaled way). These should scale. If one wanted the flexibility to have the response to solar be more or less efficient than CO_2 , it would be better to set $W_s = \text{Seff} * \text{ESS}$ - but the prior on Seff would be quite tight. Otherwise one might end up with a situation where high ESS is bizarrely matched to low W_s which does not seem physical. What is W_s in any case? (I note the SM does not include units nor the 'best guess' values for any of the parameters other than ESS. This could usefully be added I think).

ii) GEOG is the input time-series of normalized land temperature over 570 Ma. How are the uncertainties in this dealt with? It would seem to be that this is an order one uncertainty and yet it is not discussed at all (the inclusion of it in figure 4 is not particularly illuminating). I doubt that the authors are claiming that we know such temperatures exactly... Also how was it constructed since presumably one needs to know ESS and W_s to normalise actual temperatures to present day CO_2 and solar? Looking at the Cenozoic portion for instance, it appears that the authors assume that all Cenozoic variability at 10 My steps is driven by CO_2 - but isn't that begging the question? Why isn't the normalization part of the forward model? That geography can only have such a small impact is a little surprising - how does one deal with the isolation of Antarctica by the opening of Drake passage and the impact that has on glaciation?

iii) there are other time-series used as input (i.e. line 212+). What does the uncertainty sampling mean for them? Is each time point varied independently or is there some auto-correlation? What are the sources for the baseline values? This could all be usefully added to the SM.

3) I understand that the authors want to not only improve constraints, but also focus attention on key uncertainties. I am not sold on figure 4 as the main vehicle for showing this though. It doesn't explain how the parameters co-vary, merely that they do. Does a bigger GYM imply a bigger ESS or smaller? Is there a compensation between W_s and ESS? etc. A matrix of sensitivities might be more informative (i.e. a bar graph for all 25 parameters showing the sensitivity for ESS over the prior range of that value).

4) I'm a little concerned that the model doesn't have physics to properly match everything that is reflected in the target CO_2 estimates, and that the residuals are in some sense irreducible because of that. For instance, at the K/Pg boundary something happens to perturb the CO_2 (i.e. an asteroid) that can't be captured (I think) in the model setup (or is it wrapped into GEOG - how though?). There could be a tendency for MCMC procedure to try and match that through other means, perhaps biasing the whole result. Similarly, at the P/T and whatever is happening around 300 Myr if any of that survives the next point.

5) In the methods section the authors discuss the time-scale mismatch between the CO_2 observations where there is a lot of (relatively) short-term variability and the 10 My timestep in the model. Yet all of the results show the raw CO_2 data, not the CO_2 timeseries that is actually been fit. This could be misleading - please show the actual CO_2 curve that the procedure is targeting. My ability to fit a skew-normal mixture model to estimate the joint likelihood for each timestep by eye is a little rusty.

6) line 237. I don't understand this line. Nowhere in the text above have the authors described the paleo-temperature series influence, let alone a tight link - does this refer to GEOG? or some other input? Or indeed, the output? If temperature is being output then we should see it and have it compared to some independent estimates.

7) nowhere is there any discussion of non- CO_2 forcings - CH_4 , N_2O , aerosols etc. The authors might well be assuming that they are either zero, or scale proportionally to CO_2 , but either way this assumption should be explicitly stated and the consequences for the eventual interpretation of the ESS noted.

Minor points:

line 29: "carbon burial rates". No. this is not part of the ESS definition regardless of what the PALEOSENS paper said. (c.f. Lunt et al, 2010, Hansen et al. 2010).

line 29-32. Not sure I understand this sentence.

line 41. This is not the same quantity as the ESS being discussed above. It is equivalent to assuming that orbital forcings have no effect on the glacial-interglacial changes.

Figure 1. 'glacial' boxes are not discussed in the caption. And I don't think they are credible in any case.

line 169. Seems too important to leave out entirely though...

SM fig 4. This could be usefully converted into a joy plot (i.e. like the famous Joy Division cover) of stacked pdfs to allow for more data to be shown (i.e. one every 30 Ma or so).

Review on

Evidence for higher Earth-system sensitivity from long-term carbon- cycle observations

by **T E Wong et al**

submitted to *Nature Communications*, article reference: NCOMMS-19-975121-T

Date: December 2, 2019

This paper uses an extended version of the GEOCARB model, which simulates the long-term (multi-million years) evolution of the carbon cycle. Applying Bayesian statistics all model parameter are systemtically altered followed by a model application over the last 420 Myr in order to give a revised estimated of the so-called “Earth system sensitivity (ESS)”, the annual and global mean long-term (millennial-scale and longer) temperature change due to a doubling of atmospheric CO₂. They claim, that they find an updated ESS, that is higher and narrower than previous estimates.

I have some fundamental concerns with this study listed below, from which I have to conclude that I can not recommend to publish the paper.

Fundamental remarks:

1. ESS in their approach is not an output of the model, but one of the parameters (input). Thus, the improvement reported here is, how Bayesian statistic improves one of the model parameters. The target for model improvement is the CO₂ reconstruction from data (Fig 3), which is visibly shown in comparison to an older (2014) version (parametrisation) of the same model. Maybe I missed it, but I can not remember a quantitative criterion which tells the reader that and by how much the new parametrization leads to simulated CO₂, that agrees better than the previous parametrization with the CO₂ reconstructions. Maybe the Nash-Sutcliffe efficiency (methods) is used for that, but I can not find any numbers, especially comparing both parametrizations. What disturbs me further are the findings in Foster et al.

(2017) (their ref 28, on which one of the authors here (D Royer) is coauthor), in which presumably the same CO₂ data set is analysed by constructing a fitting time series, that is then compared in the Supplementary Figure 2 of ref 28 with (to my understanding) yet another version of GEOCARB, showing large discrepancies between the data fit and the model. As is I can not see if the model version presented here is the best of these 3 GEOCARB version, but it seems to me that all are still a bit away from meeting the fitted (mean) of the data. This model/data misfit of course is part of every forward modeling approach, that simulates one variable (here CO₂) independently from the reconstructions. However, since the agreement of simulated CO₂ with reconstructed CO₂ is used as a measure for model improvement, from which then the revised model parameter for ESS is taken as the main finding of the study, this still large disagreement bothers me, because it implies that ESS might also be still a way off from its “true” value, but this offset in ESS is not even shown in the given uncertainty range (which is claimed to have been narrowed).

2. What the authors analysed is not “pure” ESS, but the climate response to CO₂, solar evolution and changing geography, which, in the notation of the PALAEOSENS project (their ref 4) would have been $S_{[CO_2,geo,solar]}$, as such precisely mentioned in lines 62,63. However, the reader knows nothing about their assumptions on solar evolution and changing geography. This, in underlying changes and resulting radiative forcing needs to be shown, and then set in relation to CO₂ radiative forcing. It is also problematic, that throughout the paper the authors talk about ESS, while what they in fact analyse is $S_{[CO_2,geo,solar]}$. It also needs to be clarified which versions of S (which are the assumed forcing changes X in the notation of $S_{[X]}$?) are compiled in Supplementary Table 1.
3. Another way of independently validating the results would have been to compare simulated temperature change with reconstructed temperature change. However, this has not been done so far, apart from a few points for ESS entered in Fig 1, for which it would be important to know if they are estimates of pure ESS or of $S_{[CO_2,geo,solar]}$. However, all data points in Fig 1 are younger than 50 Myr

(+ one assumption around 220 Myr). I believe that more temperature change estimates have been published, e.g. Krissansen-Totton and Catling (2017), their ref 20, contains data points until 100 Myr ago, and there are surely others, e.g. 150 Myr in Farnsworth et al. (2019). Without such independent temperature estimates the numbers on ESS given here are difficult to evaluate.

4. I am not very familiar with the GEOCARB models, but what I understood from reading a few of the paper is, that ocean alkalinity is not explicitly considered. Please correct me if I am wrong. However, especially weathering (one of the processes explicitly mentioned) changes atmospheric CO_2 via the alkalinity input. Higher alkalinity leads to an increased oceanic uptake of atmospheric CO_2 , ultimately leading via carbonate compensation to a feedback of the ocean-sediment fluxes of carbonate. It might be that these effects are somehow “hidden” in the mentioned “long-term” effects, which are only considered. However, from my understanding atmospheric CO_2 is determined by the ocean (more precisely the surface ocean) state of the carbonate system, that has 2 degrees of freedom, therefore not only the carbon content itself, but also alkalinity needs to be known. This at least has been done in a different study (Krissansen-Totton and Catling, 2017), their ref 20, but to my understanding not in GEOCARB. For me, it is therefore difficult to understand how weathering effects can be represented at all within GEOCARB. However, since this a very general comment on all studies using GEOCARB I have the feeling, that the authors might have an answer for it, but right now I can not see evidence that would convince me to trust the results connected with any processes that change ocean alkalinity (and weathering is just one of these processes).

5. Long-term effects: It is said in the abstract that ESS is the long-term global warming to CO_2 doubling, where long-term implies “millennial-scale” and longer. The effects investigated with GEOCARB are, however, more multi-million years with one time step being 10 Myr, which makes sense since weathering (one of the processes within GEOCARB) operates on million-year time-scales. However, this is a factor of 1000 or more longer than the mentioned millennial-scale mentioned for ESS. It might

therefore indeed be worth to name it differently (see # 2 above), or reframe this time-scales importance. Furthermore, the question is, how relevant multi-million years effects are for ongoing anthropogenic future warming, for which mainly the so-called Charney climate sensitivity considering only fast feedbacks is of importance. As such I find it hard to consider the calculated ESS to be an important policy relevant issue, since no connection between calculated numbers and Charney climate sensitivity is introduced.

6. Krissansen-Totton and Catling (2017) has with 5.6°C a higher value for ESS than this study here (5.1°C, see your Supplementary Figure 3). Therefore, your claim given in the abstract, that you show higher values for ESS than other studies is wrong.

Minor issues in chronological order:

1. Page 1: I do not understand, for what purpose 3 key points have been given. This looks like a format for another journal (e.g. AGU journals).
2. line 16, Insert “surface” to global temperature changes.
3. line 22. Fast feedbacks in that context are those mentioned + “the Planck feedback”. So no need for “for example”, and please add “Planck feedback” here.
4. throughout, e.g. line 34, 37, 42: When you talk about “climate sensitivity” instead of ESS it needs to be known what are the assumed forcings. It would be better to always refer to a specific $S_{[X]}$ and naming each X following the PALAEOSESN nomenclature, otherwise nobody knows how these numbers have to be understood.
5. Check out and discuss Farnsworth et al. (2019) for climate sensitivity of the last 150 Myr, which also plots temperature change over that time interval.
6. line 179: The list of various model parameters is not helpful for the reader, if they are not explained in detail.

7. line 288: Ice core data for the last 800 kyr show atmospheric CO₂ between 180 and 300 ppm. So why is “present-day CO₂ restricted to values above 280 ppm, and not above 180 ppm?
8. Supplementary Figure 1: How do you judge, which output agrees better with the CO₂ reconstructions?

References

- Farnsworth, A., Lunt, D. J., O’Brien, C. L., Foster, G. L., Inglis, G. N., Markwick, P., Pancost, R. D., and Robinson, S. A.: Climate Sensitivity on Geological Timescales Controlled by Nonlinear Feedbacks and Ocean Circulation, *Geophysical Research Letters*, 46, 9880–9889, doi:10.1029/2019GL083574, 2019.
- Foster, G. L., Royer, D. L., and Lunt, D. J.: Future climate forcing potentially without precedent in the last 420 million years, *Nature Communications*, 8, 14 845, 2017.
- Krissansen-Totton, J. and Catling, D. C.: Constraining climate sensitivity and continental versus seafloor weathering using an inverse geological carbon cycle model, *Nature Communications*, 8, 15 423, 2017.

Review responses

Reviewer #1 (Remarks to the Author):

This paper uses an extended version of the GEOCARB model, which simulates the long-term (multi-million years) evolution of the carbon cycle. Applying Bayesian statistics all model parameter are systemically altered followed by a model application over the last 420 Myr in order to give a revised estimated of the so-called “Earth system sensitivity (ESS)”, the annual and global mean long-term (millennial-scale and longer) temperature change due to a doubling of atmospheric CO₂. They claim, that they find an updated ESS, that is higher and narrower than previous estimates.

I have some fundamental concerns with this study listed below, from which I have to conclude that I can not recommend to publish the paper.

We greatly appreciate your detailed comments, which have greatly helped to improve the clarity, quality and thoroughness of the work.

Fundamental remarks:

1. ESS in their approach is not an output of the model, but one of the parameters (input). Thus, the improvement reported here is, how Bayesian statistic improves one of the model parameters. The target for model improvement is the CO₂ reconstruction from data (Fig 3), which is visibly shown in comparison to an older (2014) version (parametrisation) of the same model. Maybe I missed it, but I can not remember a quantitative criterion which tells the reader that and by how much the new parametrization leads to simulated CO₂, that agrees better than the previous parametrization with the CO₂ reconstructions. Maybe the Nash-Sutcliffe efficiency (methods) is used for that, but I can not find any numbers, especially comparing both parametrizations. What disturbs me further are the findings in Foster et al. (2017) (their ref 28, on which one of the authors here (D Royer) is coauthor), in which presumably the same CO₂ data set is analysed by constructing a fitting time series, that is then compared in the Supplementary Figure 2 of ref 28 with (to my understanding) yet another version of GEOCARB, showing large discrepancies between the data fit and the model. As is I can not see if the model version presented here is the best of these 3 GEOCARB version, but it seems to me that all are still a bit away from meeting the fitted (mean) of the data. This model/data misfit of course is part of every forward modeling approach, that simulates one variable (here CO₂) independently from the reconstructions. However, since the agreement of simulated CO₂ with reconstructed CO₂ is used as a measure for model improvement, from which then the revised model parameter for ESS is taken as the main finding of the study, this still large disagreement bothers me, because it implies that ESS might also be still a way off from its “true” value, but this offset in ESS is not even shown in the given uncertainty range (which is claimed to have been narrowed).

Thank you for making several excellent points here that highlight key areas in which we have improved the analysis.

First, we note that there is an error in the Foster et al. (2017) paper in that the reference for the GEOCARB simulations therein should have been Royer et al. (2014), not Berner and Kothavala (2001). We apologize for the confusion this may have caused. We are indeed using the same version of GEOCARBSULF as presented in Royer et al. (2014), and as used by Foster et al. (2017). We clarify this in the revised text at line 96.

Our adopted GEOCARBSULFvolc model[13] (henceforth, “GEOCARB”) has 68 input parameters (including both constant and time-variable parameters; see Supplementary Materials) and is structurally identical to the model as presented in ref. 13 and used in ref. 25.

Second, in our revised analysis we have elected to use a simpler method for generating an ensemble of model simulations to match the CO₂ observations. The Markov chain Monte Carlo Bayesian calibration approach led to overconfidence in the model hindcast of CO₂. To avoid this issue, we have elected to use a precalibration windowing approach instead of the formal likelihood function. The windows are ranges of CO₂ (and temperature, discussed later) for each time slice (10 Myr) that fully encompass the available data within that time slice. We note that the precalibration windowing approach is akin to a Bayesian approach that uses a step function as the likelihood function. A similar approach is taken by the recent work of Mills et al. (2019). Those authors use the percentage of time-steps in which the model hindcast steps outside of the precalibration windows as a model goodness-of-fit measure. For example, if the model hindcast goes through 53 out of the 58 precalibration CO₂ windows (there are 58 time-steps of 10 Myr each), then in the notation of Mills et al., the “%outbound” is 8.6%. We adopt this goodness-of-fit metric here. The changes to the text associated with this revision are too numerous to reasonably copy here, but in the revised manuscript the new experimental set-up is described in Methods, and we describe it briefly in the main text at line 143:

First, we use a Monte Carlo precalibration approach to account for uncertainties in the 68 GEOCARB model parameters and the surface temperature and CO₂ proxy data (see Methods). The essence of this calibration method is to sample a large number of model parameter sets from their prior distributions – these are the a priori parameter values, taken before any data are fused with the model. Then, we rule out any combinations of parameters that yield simulations that do not agree well with the CO₂ proxy or temperature data, given their uncertainties. What remains are the a posteriori ensembles of parameters, including S. We use Latin hypercube sampling to draw samples of the constant parameters and inverse Wishart sampling to account for uncertainty and autocorrelation in the time series parameters (see Methods and Supplementary Figure 2).

We note also that other studies using precalibration for proxy-based reconstructions have used other measures for goodness-of-fit, for example reduction of error (e.g., Mann et al., 2005). Our use of the %outbound measure is motivated by the need to present the cleanest comparison with the work of Mills et al. (2019), who also examine CO₂ and temperature constraints on GEOCARBSULF (and COPSE) simulations.

Lastly, you are indeed correct in the observation that if the CO₂ hindcast remains biased, then this will lead to biases in the corresponding estimates of ESS. In our revised

analysis, the precalibration approach taken is meant to bound the set of plausible values, as opposed to deem one as the most likely or the best estimate.

2. What the authors analysed is not “pure” ESS, but the climate response to CO₂, solar evolution and changing geography, which, in the notation of the PALAEOSENS project (their ref 4) would have been S[CO₂,geo,solar], as such precisely mentioned in lines 62,63. However, the reader knows nothing about their assumptions on solar evolution and changing geography. This, in underlying changes and resulting radiative forcing needs to be shown, and then set in relation to CO₂ radiative forcing. It is also problematic, that throughout the paper the authors talk about ESS, while what they in fact analyse is S[CO₂,geo,solar]. It also needs to be clarified which versions of S (which are the assumed forcing changes X in the notation of S[X]?) are compiled in Supplementary Table 1.

We apologize for the confusion caused by our lack of specificity on the particular type of Earth system sensitivity we were referring to. In the revised manuscript, we have added text (see below) at line 74 to clarify the meaning of ESS throughout the remainder of the work. At that point, we mention that solar luminosity and changing geography are considered internal forcings and that GEOCARB assumes input time series for these forcings, and we point to a new Supplementary Figure that shows these input time series and their uncertainty ranges.

The GEOCARBSULFvolc model and its previous incarnations[23,24] have been widely used in previous studies [e.g., refs. 12,13,17,25,26], and includes a version of ESS where the only independent radiative forcings are CO₂, solar evolution and changing geography (S[CO₂, geog, solar]). GEOCARBSULFvolc assumes a linear increase in solar luminosity over time, corresponding to the parameter W_s , and uses results from general circulation model output to simulate the temperature change resulting from changes in paleogeography (GEOG; see Supplementary Figure 1)[27]. Our study focuses mainly on the form of ESS as used by GEOCARBSULFvolc (S[CO₂, geog, solar]). For brevity, we will use S when referring to ESS within the GEOCARBSULFvolc model, and reserve the term “ESS” for discussion of Earth system sensitivity more generally.

We have removed Figure 1 and Supplementary Table 1 in order to avoid confusion among the various types of ESS depicted in the literature. We are now more explicit in the text about what the forcings we consider in this work are, and the difference between “true” ESS and what is present in GEOCARB (S[CO₂,geo,solar], as you point out).

3. Another way of independently validating the results would have been to compare simulated temperature change with reconstructed temperature change. However, this has not been done so far, apart from a few points for ESS entered in Fig 1, for which it would be important to know if they are estimates of pure ESS or of S[CO₂,geo,solar]. However, all data points in Fig 1 are younger than 50 Myr (+ one assumption around 220 Myr). I believe that more temperature change estimates have been published, e.g. Krissansen-Totton and Catling (2017), their ref 20, contains data points until 100 Myr ago, and there are surely others, e.g. 150 Myr in Farnsworth et al. (2019). Without such independent temperature estimates the numbers on ESS given here are difficult to evaluate.

Thank you for this great suggestion. Our revised analysis consists of two main precalibration experiments: the first uses only CO₂ data to constrain the model hindcast, and the second uses both CO₂ and temperature data as constraints. This enables both an evaluation of the additional constraint offered by temperature, as well as an improved estimate of ESS, by reducing the simulations for analysis to only those that adequately match both temperature and CO₂. We describe the updated set of experiments beginning at line 165 in the revised manuscript. We use the temperature compilation from Mills et al. (2019) because it covers a deeper time period than Farnsworth et al. (2019), but we do use the latter study in our discussion of a set of temperature experiments around line 334. Additionally, we have removed Figure 1, which eliminates the need to discuss the individual temperature estimates in the studies there.

4. I am not very familiar with the GEOCARB models, but what I understood from reading a few of the paper is, that ocean alkalinity is not explicitly considered. Please correct me if I am wrong. However, especially weathering (one of the processes explicitly mentioned) changes atmospheric CO₂ via the alkalinity input. Higher alkalinity leads to an increased oceanic uptake of atmospheric CO₂, ultimately leading via carbonate compensation to a feedback of the ocean-sediment fluxes of carbonate. It might be that these effects are somehow “hidden” in the mentioned “long-term” effects, which are only considered. However, from my understanding atmospheric CO₂ is determined by the ocean (more precisely the surface ocean) state of the carbonate system, that has 2 degrees of freedom, therefore not only the carbon content itself, but also alkalinity needs to be known. This at least has been done in a different study (Krissansen-Totton and Catling, 2017), their ref 20, but to my understanding not in GEOCARB. For me, it is therefore difficult to understand how weathering effects can be represented at all within GEOCARB. However, since this a very general comment on all studies using GEOCARB I have the feeling, that the authors might have an answer for it, but right now I can not see evidence that would convince me to trust the results connected with any processes that change ocean alkalinity (and weathering is just one of these processes).

You bring up a fair and interesting point about modeling the long-term carbon cycle. We follow the approach of Berner (2004, pp. 9-11) and consider the surficial system: rocks, ocean, atmosphere, land and biosphere. The amount of carbon in the rock component far exceeds that in the other four components (cf. Berner (2004), table 1.1).

In GEOCARB, a mass balance governs changes in carbon over time among the reservoirs in the surficial system. This is laid out in (e.g.) Eq. 1 of Royer et al. (2014):

$$dM_c/dt = F_{wc} + F_{wg} + F_{mc} + F_{mg} - F_{bc} - F_{bg},$$

where M_c is the total carbon mass in the surficial system, F_{wc} represents the flux due to weathering of Ca and Mg carbonates, F_{wg} represents the weathering flux of sedimentary organic carbon, F_{mc} represents the degassing flux from carbonates, F_{mg} represents the degassing from organic carbon, F_{bc} represents the burial of carbonate and F_{bg} represents the burial of organic carbon. A carbon isotopic mass balance accompanies this as an additional constraint. Thus, the weathering processes (F_{wc} and F_{wg}) are parameterized to capture the average balance among these carbon sources and sinks, assuming a steady state balance over the course of a 10 Myr time step. *It has been shown that for modeling the long-term carbon cycle, no perturbations around the*

steady state can persist for more than 500,000 years (Sundquist, 1991), including for alkalinity.

We have added the italicized note above about the steady state assumption at line 128 and the mass balance equation beginning at line 120 of the revised manuscript.

5. Long-term effects: It is said in the abstract that ESS is the long-term global warming to CO₂ doubling, where long-term implies “millennial-scale” and longer. The effects investigated with GEOCARB are, however, more multi-million years with one time step being 10 Myr, which makes sense since weathering (one of the processes within GEOCARB) operates on million-year time-scales. However, this is a factor of 1000 or more longer than the mentioned millennial-scale mentioned for ESS. It might therefore indeed be worth to name it differently (see # 2 above), or reframe this time-scales importance. Furthermore, the question is, how relevant multi-million years effects are for ongoing anthropogenic future warming, for which mainly the so-called Charney climate sensitivity considering only fast feedbacks is of importance. As such I find it hard to consider the calculated ESS to be an important policy relevant issue, since no connection between calculated numbers and Charney climate sensitivity is introduced.

We completely agree with the point you raise, that in the near-term (say, less than thousands of years) the Charney feedbacks are of course the important ones for calculating the warming response to anthropogenic CO₂ emissions. However, even if all human CO₂ emissions were to cease, global temperatures would not decrease appreciably for thousands of years, which would be enough time to see the longer-term non-Charney feedbacks have an effect. This is still much less than the 10 Myr GEOCARB time step. But, it is important to characterize the ESS accounting for different combinations of forcings (to evaluate the impacts of each) and for different time scales. Thus, the ESS value we produce is relevant for providing context for both shorter-term sensitivity estimates and estimates with different combinations of external forcings considered (e.g., comparison with S[CO₂, solar], and with other assumed time series for paleogeography for S[CO₂, solar, geog], could shed light on the importance of the geographical forcing). In light of these comments, we have removed the phrase “policy-relevant” from the abstract. We retain mentioning policy development in the Introduction (first paragraph) because there we are introducing why any ESS/ECS is worth studying at all.

6. Krissansen-Totton and Catling (2017) has with 5.6°C a higher value for ESS than this study here (5.1°C, see your Supplementary Figure 3). Therefore, your claim given in the abstract, that you show higher values for ESS than other studies is wrong.

This point highlights an important area for clarification - there is some nuance to the values of ESS presented in our work and those of Krissansen-Totton and Catling (2017). Specifically, the ESS values of Krissansen-Totton and Catling (2017) include both glacial and non-glacial periods (past 100 Myr), so they are not directly comparable to our non-glacial ESS estimate, nor to our glacial (GLAC* ΔT_{2X}) ESS estimate. For this reason, we elected to present the comparison on its own as a supplementary figure (Supplementary Figure 4). The fact that the ESS estimates of Krissansen-Totton and Catling (2017) fall between our glacial and non-glacial estimates is expected, given that the estimates of those authors should be some mixture of both glacial and non-glacial

ESS - falling somewhere in between. That being said, in our revised set of experiments, our central ESS estimate falls below the predictive range from Krissansen-Totton and Catling (2017), so we have removed this statement from the revised manuscript.

Minor issues in chronological order:

1. Page 1: I do not understand, for what purpose 3 key points have been given. This looks like a format for another journal (e.g. AGU journals).

We apologize for this oversight - we meant for those to focus the manuscript and presentation as it was developed and did not mean for them to distract from the text. They have been removed in the revised manuscript.

2. line 16, Insert “surface” to global temperature changes.

Thank you for this useful suggestion, which is of elevated importance given the addition of the global temperature comparisons. This has been added at this point, and other relevant spots throughout the manuscript to hopefully strike a balance between specificity and verbosity.

3. line 22. Fast feedbacks in that context are those mentioned + “the Planck feedback”. So no need for “for example”, and please add “Planck feedback” here.

Thank you for pointing this out. We have added “Planck feedback” and removed “for example” from the revised manuscript.

4. throughout, e.g. line 34, 37, 42: When you talk about “climate sensitivity” instead of ESS it needs to be known what are the assumed forcings. It would be better to always refer to a specific S[X] and naming each X following the PALAEOSESN nomenclature, otherwise nobody knows how these numbers have to be understood.

This is a nice suggestion to improve the clarity of the manuscript. There is some overlap in addressing this concern as well as your Fundamental Remark #2 (above), so we apologize for any redundancy in our response here. In the revised manuscript at line 80, we have added a statement to declare that for brevity’s sake, we use “S” to refer to S[CO₂, solar, geog] as used by our GEOCARB model and “ESS” to refer more generally to Earth-system sensitivity in a broader sense:

Our study focuses mainly on the form of ESS as used by GEOCARBSULFvolc (S[CO₂, geog, solar]). For brevity, we will use S when referring to ESS within the GEOCARBSULFvolc model, and reserve the term “ESS” for discussion of Earth system sensitivity more generally.

5. Check out and discuss Farnsworth et al. (2019) for climate sensitivity of the last 150 Myr, which also plots temperature change over that time interval.

Thank you for suggesting this reference. We are of course aware of the work and our revised analysis that includes temperature as a constraint on the paleo simulations elevates the importance of discussing the Farnsworth et al. (2019) paper. We have

added the following text to the discussion of our temperature sensitivity experiments at line 333:

With these modifications, we find that the simulated temperature peak occurs at a time period (~100 Myr ago) and magnitude (~30 °C) as a recent analysis using paleoclimate model simulations and temperature proxy data³⁴ (c.f., their Fig. 2). Thus, we make no further adjustments here, but note that factors such as a time-varying plant-assisted weathering efficiency and uncertainty in the timing of the gymnosperm-angiosperm domination transition can be incorporated into future work as revised parameters or time series forcings become available.

6. line 179: The list of various model parameters is not helpful for the reader, if they are not explained in detail.

We apologize for not explaining the meaning behind the parameters in greater detail. We have removed this section from the revised manuscript (because the sensitivity analysis has changed), but in the new sensitivity discussion surrounding the paleo temperature constraint, we have added brief descriptions of the parameters as they arise in the revised manuscript. Some examples include the description of the temperature parameterization (beginning around line 122):

... where M_c is the total carbon mass in the surficial system, F_{wc} represents the flux due to weathering of calcium and magnesium carbonates, F_{wg} represents the weathering flux of sedimentary organic carbon, F_{mc} represents the degassing flux from carbonates, F_{mg} represents the degassing from organic carbon, F_{bc} represents the burial of carbonate and F_{bg} represents the burial of organic carbon.

and the discussion of temperature sensitivity experiments (beginning at line 289):

The parameters ACT (the activation energy for the dissolution of calcium and magnesium silicate rocks on land) and GYM (the rate of chemical weathering by gymnosperms, relative to angiosperms) have Spearman rank correlations with T_{Cret} of -0.39 and -0.44 (Pearson correlations of -0.41 and -0.40, respectively).

7. line 288: Ice core data for the last 800 kyr show atmospheric CO₂ between 180 and 300 ppm. So why is “present-day CO₂ restricted to values above 280 ppm, and not above 180 ppm?”

This is a very good point. We selected 280-400 ppmv as the range for “present-day” CO₂ limits following the work of Royer et al. (2014). In the revised analysis, precalibration windows for each time step are computed by adding +1sigma to the central estimate for each data point in a 10-Myr time step and taking the maximum among these upper bounds as the upper bound of the time step’s window; lower bounds are similarly found by subtracting -1sigma. This leads to a window of 40 to 669 ppmv for the present-day (last 5 Myr) CO₂ concentration. While this is much wider than the 180-300 ppmv from ice core records, the essence of a precalibration scheme is to only rule out simulations that can be “relatively uncontroversially classified as ‘non-physical’”. Thus, a wider precalibration window permits an approximate quantification of uncertainty

while not relying as heavily on subjective choice of data sources or form of (for example) the uncertainty distribution around those data points.

8. Supplementary Figure 1: How do you judge, which output agrees better with the CO2 reconstructions?

This is a great suggestion to clarify the multi-faceted comparisons in this figure. The difficulty in communicating this clearly is compounded now that we have added the temperature hindcast to our analysis. We do not want to impose any parametric assumption about the structure of the uncertainties about each data point (to avoid overfitting the data points with the lowest CO2 concentrations), so we use the %outbound (Mills et al., 2019) to cull each Latin hypercube ensemble down to only the simulations with at most 30% of the time-steps in CO2 or temperature outside of the precalibration windows. However, because we have simplified the (pre)calibration procedure, we have removed this supplemental figure from the analysis.

References

- [1] Farnsworth, A., Lunt, D. J., O'Brien, C. L., Foster, G. L., Inglis, G. N., Markwick, P., Pancost, R. D., and Robinson, S. A.: Climate Sensitivity on Geological Timescales Controlled by Nonlinear Feedbacks and Ocean Circulation, *Geophysical Research Letters*, 46, 9880–9889, doi:10.1029/2019GL083574, 2019.
- [2] Foster, G. L., Royer, D. L., and Lunt, D. J.: Future climate forcing potentially without precedent in the last 420 million years, *Nature Communications*, 8, 14 845, 2017.
- [3] Krissansen-Totton, J. and Catling, D. C.: Constraining climate sensitivity and continental versus seafloor weathering using an inverse geological carbon cycle model, *Nature Communications*, 8, 15 423, 2017.

Reviewer #2 (Remarks to the Author):

Review Wong et al. “Evidence for higher ESS from long-term carbon-cycle observations”

Let me start off with this: I think this is a novel study (while at the same time well-founded in previous work, which underlines its relevance) that has strong potential, and an important message. Eventually, I would like to see this published, and Nature Communications is a good outlet for it, with a broad readership. I will dig deep in the following, but this should not be seen as an attempt to reject the paper. On the contrary, I’m hoping to expose where the paper needs some extra thought, or where it needs extra clarity or detail to make it stand-alone and accessible to a broad readership. If/When I get things completely wrong, this likely is the case because I couldn’t follow or reproduce in my head what was being done; so, in those cases, better explanation is needed. Also, it seems to me that a structural rethink (maybe with a flow diagram) is needed to bring out exactly what the model starts with (and show this “master” time series), what it then does, and what is diagnosed for this study (CO₂, T, ESS). Then, a better/clearer comparison of those outputs with proxy data is warranted, including more thought on what mismatches may mean for the study’s conclusions – that level of critical (self-)assessment seems to be too weak to me.

Thank you for the words of encouragement and their many helpful comments. The constructive and thorough critique has greatly helped to improve the quality and clarity of the manuscript.

We have added a workflow diagram to help clarify the workflow and inputs/outputs. This is now Figure 1 in the revised manuscript, and is described in the new text at line 173.

We have also added a Supplementary figure showing the 12 time series forcings for the GEOCARB model (similar to Royer et al. 2014, their Fig 1). We explain these input forcings and describe how we account for uncertainty in them at line 408, in Methods, in the revised text.

So, I think the following aspects need in-depth attention:

First, the paper contains more technical jargon than it really needs to. I suggest going through it with a fine comb to minimise it to what is absolutely necessary. Bear in mind, that you’re aiming for a broad readership. A good example (but not the only one) of excessive jargon is in lines 93-99, where “prior distribution centers in the Monte Carlo resampling strategy” are mentioned before the Bayesian approach is even introduced, followed by some arcane sentences that make sense only to someone who has operated the specific model referred to. Such passages need to be written in a more accessible manner. Other examples are the captions of Suppl. Figures 1 and 2 – they are so terse that they are entirely inaccessible. And the end of line 180: “including the assumed time series of isotopic forcing”; what does that mean? Which isotopes (carbon I presume). Which forcing? And how is a time-series assumed? Is it based on data, or just made up? Please clarify in less terse and more intelligible terms.

Thank you for the many good suggestions here and highlighting an important way we can make the work more accessible to a broader audience. Specifically, the (pre)calibration approach that we use in the revised work is described beginning at line 143 in what we think are much more layperson's terms than our original description:

First, we use a Monte Carlo precalibration approach to account for uncertainties in the 68 GEOCARB model parameters and the surface temperature and CO2 proxy data (see Methods). The essence of this calibration method is to sample a large number of model parameter sets from their prior distributions – these are the a priori parameter values, taken before any data are fused with the model. Then, we rule out any combinations of parameters that yield simulations that do not agree well with the CO2 proxy or temperature data, given their uncertainties. What remains are the a posteriori ensembles of parameters, including S. We use Latin hypercube sampling to draw samples of the constant parameters and inverse Wishart sampling to account for uncertainty and autocorrelation in the time series parameters (see Methods and Supplementary Figure 2).

...

As a goodness-of-fit measure, we use the percentage of time steps in which a model simulation is outside the range of the precalibration windows around the data, termed “%outbound” following Mills et al.[9].

...

We generate model ensembles that agree with CO2 proxy data only, temperature reconstructions only, and both, by imposing limits of at most 50%, 45%, 40%, 35% and 30% of time steps to be out-of-bounds (for a total of 15 main experiments).

In light of our revised set of experiments, we have removed Supplementary Figures 1 and 2, but we apologize for the terseness of the captions. We have attempted to “soften” the delivery of the information in our revised captions, but we trust that the reviewers will kindly let us know where our efforts fall short.

The revised sensitivity experiments focus on controls on paleo-temperature during the Cretaceous, and the revised manuscript no longer includes mention of the assumed time series of isotopic forcing. The time series that we were referring to in the original manuscript are the isotope ratios for strontium, carbon and sulfur (shown in the revised Supplementary Figure 1a-c).

The paper also is presented too much in a manner that assumes that people know as much as the authors about the exact setup of the experiments. For example, tables of parameters are given, but not the values used, or the uncertainties. It is mentioned that values and uncertainty ranges were used, but the reader is nowhere shown what these values and ranges are in the experiments. That makes the work essentially unreproducible, and that's not good enough in my view. Parameters are mentioned in the text using their acronyms, but these are not always introduced/spelled out. This ought to be added, and/or reference needs to be made in appropriate places to the tables of parameter explanations.

We apologize for this oversight. We have removed the original Supplementary tables that were given in the supplemental text document and now include a supplementary

spreadsheet file that gives all of the parameter names, descriptions, the central estimates for the prior distribution ranges, the 5th and 95th percentiles of the prior distributions, and the medians, 5th and 95th percentiles of the precalibration ensemble using both CO₂ and temperature data, and a %outbound threshold of 30%.

We have also revised the text in numerous locations to give a brief explanation of each parameter as it comes up in the discussion. Examples can be found at line 122, where we discuss the temperature parameterization of GEOCARB or line 289, where we discuss the sensitivity of the simulated temperatures to the gymnosperm/angiosperm weathering efficiency parameters and transition timing, but there are a few other instances throughout the revised manuscript as well. For convenience, we pasted below the sections we mentioned above.

Line 122: ... where M_c is the total carbon mass in the surficial system, F_{wc} represents the flux due to weathering of calcium and magnesium carbonates, F_{wg} represents the weathering flux of sedimentary organic carbon, F_{mc} represents the degassing flux from carbonates, F_{mg} represents the degassing from organic carbon, F_{bc} represents the burial of carbonate and F_{bg} represents the burial of organic carbon.

Line 289: The parameters ACT (the activation energy for the dissolution of calcium and magnesium silicate rocks on land) and GYM (the rate of chemical weathering by gymnosperms, relative to angiosperms) have Spearman rank correlations with TC_{ret} of -0.39 and -0.44 (Pearson correlations of -0.41 and -0.40 , respectively).

And the paper migrates too freely (in a non-specific manner) across terms like ESS and climate sensitivity, especially in the first several paragraphs. Ranges are compared between studies, where some are ESS, and others determined over centuries and therefore clearly NOT ESS, but more like Equilibrium Climate Sensitivity (ECS) (lines 34-43). This level of non-specific description and comparison is confusing.

Thank you for this good suggestion, and we apologize for the several instances where we mistakenly used the term “climate sensitivity”. Those instances have been fixed to ESS, and we now use “S” when discussing the specific form of ESS as represented by the GEOCARB model and reserve “ESS” for more general discussion of “true” Earth-system sensitivity. We clarify this now at line 80 of the revised manuscript:

Our study focuses mainly on the form of ESS as used by GEOCARBSULFvolc ($S[CO_2, geog, solar]$). For brevity, we will use S when referring to ESS within the GEOCARBSULFvolc model, and reserve the term “ESS” for discussion of Earth system sensitivity more generally.

For Figure 1, a supplementary table is needed that documents exactly what values are used (e.g., some studies determined SST, not mean global surface temperature, so that a correction must have been made to generate Figure 1, but no details are provided). In short here, the terminology needs to be sorted out very carefully throughout the manuscript. And composite figures like Fig 1 need to be backed up with a supplementary table that documents exactly how the plotted values were obtained. It is very important that this is checked carefully to ensure that

the plotted values are indeed ESS (not ECS), and that any conversion that were made between the two are documented in the supplementary table. This is all too “loose” at the moment.

We have removed Figure 1 to avoid much of the confusion, but retain some of the discussion of these previous studies in Earth-system sensitivity. We make a note of the different flavors used in two studies specifically mentioned in text at line 52:

For example, the geochemical model from Royer et al.[17] uses a form of ESS that assumes external forcings from CO₂, solar luminosity changes and paleogeography, or S[CO₂, geog, solar], in the notation of Rohling et al.[4], whereas the ESS estimate of Anagnostou et al.[19] is based on external forcing from CO₂ alone (S[CO₂]). Both of these studies’ sensitivity estimates are based on global mean surface temperatures.

Similarly, I think that the model code (including all modifications such as GLAC) needs to be made publicly available, and if it is available somewhere already, then a URL needs to be provided.

Thank you for emphasizing this important point. We have moved the Data Availability section to before the References, per Nature Communications formatting. All files are available on Github, and will be put on Zenodo with a stable URL and DOI specific to the final code versions after the review process.

Second, I would like to see the temperature values used in this study to calculate ESS. I think they’re all just obtained from the model somehow. But how do they compare with data? And then, what is the confidence in the temperature responses, and how does the model determine this?

This is a great suggestion and highlights a key angle that we had not looked at in the original work. The temperature in GEOCARB is calculated as a function of the three external forcings: CO₂, solar luminosity and paleogeography. The calculation of temperature in GEOCARB is:

$$T(t)-T(0) = ESS \cdot RCO_2(t)$$

where $T(t)-T(0)$ denotes the global mean land surface temperature at time t (Myr ago) relative to present ($t=0$), RCO_2 is the CO₂ concentration in year t relative to present, W_s is a parameter taking into account the linear change in solar luminosity and $GEOG$ is the global mean land surface temperature in year t relative to present assuming present-day values for solar luminosity and CO₂. W_s , $GEOG$ and ESS are taken as uncertain input parameters. We have added a description of this temperature calculation to the revised manuscript beginning at line 110:

The temperature in GEOCARB is computed as

$$T(t)-T(0) = S * RCO_2(t), \tag{1}$$

where $T(t)-T(0)$ denotes the global mean land surface temperature at time t (Myr ago) relative to present ($t=0$) and $RCO_2(t)$ is the CO₂ concentration in year t

relative to present. We follow ref. 32 and assume a present (last 5 Myr) mean global land surface temperature of 15 °C.

We have expanded the analysis to examine the model-data mismatch between temperature as well as CO₂. We use the temperature compilation of Mills et al. (2019) as our model validation data for temperature. The updated calibration procedure for comparing the model to the data is described in greater detail below in responding to your comment beginning “Third” (regarding model overconfidence), so we only review it briefly here.

The temperature data of Mills et al. (2019) provide an upper and lower uncertainty range. We quantify how well a model simulation matches the data using the proportion of time steps in which a model simulation is within the given error range. We describe this in the revised text at line 158:

We establish a maximal +/-1 window around all of the time series data for each of temperature and CO₂. For the CO₂ data, we use the proxy compilation of Foster et al.[25], and for the temperature data, we use the Phanerozoic temperature compilation of Mills et al.[9]. As a goodness-of-fit measure, we use the percentage of time steps in which a model simulation is outside the range of the precalibration windows around the data, termed “%outbound” following Mills et al.[9].

We generate model ensembles that agree with CO₂ proxy data only, temperature reconstructions only, and both, by imposing limits of at most 50%, 45%, 40%, 35% and 30% of time steps to be out-of-bounds (for a total of 15 main experiments).

Also, how good is the temperature determination from the model on a structural level; for example, does it assume constant and similar efficacies between all radiative influences to determine temperature (i.e., how does the model cope with feedback efficacy changes and/or potential non-linear addition of feedback processes)? In the complete absence of discussion on this aspect, and especially of the rather large uncertainties that are likely associated with inferred T changes and/or their comparison with data (not shown), I am uncomfortable with the conclusions as they stand. It may well be that all of this is covered and perfectly in order. But I am given absolutely nothing to judge it by.

The revised calibration procedure includes using global mean land surface temperature as a constraint alongside the atmospheric CO₂ concentrations. We now present the percentage of model time-steps in which the simulation is out of agreement with the given temperature compilation (from Mills et al., 2019).

We discuss the sensitivity of the GEOCARB temperature output during the early Cretaceous period, where the largest bias in model output temperature relative to the data compilation of Mills et al. (2019) can be seen (cf. Figure 4b in the revised manuscript and pasted below for convenience).

We conduct a sequence of sensitivity experiments in order to evaluate the GEOCARB parametric controls on early Cretaceous temperature by computing the correlations (both linear Pearson and Spearman rank) between the 56 GEOCARB constant parameters and the mean temperature between 140 and 90 Myr ago. These experiments highlight GYM, the parameter that represents the gymnosperm weathering efficiency rate relative to angiosperms, as a parameter that is most strongly associated with higher Cretaceous temperature. GEOCARB parameterizes the transition from gymnosperm to angiosperm domination by using a linear shift in the weathering efficiency rate from GYM to a value of 1 (a rate of 1 relative to angiosperm-dominated weathering implies angiosperm domination). This shift happens over the 130-80 Myr ago time frame. Our supplemental experiments include (1) reducing GYM by a factor of $\frac{1}{4}$ during the early Cretaceous, (2) shifting the timing of the gymnosperm-angiosperm transition 20 Myr later, to occur between 110-60 Myr ago, which is still in agreement with more recent assessments of the onset of angiosperm domination (e.g., Coiffard et al., 2012). We find that GEOCARB is much better able to match the sharp increase in Cretaceous temperature seen in the data with these parametric adjustments, highlighting the importance of the plant-assisted weathering in the temperature response to changes in atmospheric CO₂. We discuss this sensitivity experiment in the revised manuscript at line 271, in a new section that replaces the old sensitivity experiment section. The new section is titled “Controls on Cretaceous temperature biases”.

Importantly, simple referring to a bunch of previous studies when revising will not be good enough; the arguments should be at least synthesised in the supplement, so that all critical information is brought together nicely for the reader. This should not be too onerous, even if should be done in a transparent manner that is well-referenced and presented with as little jargon as possible.

Again, many thanks for the helpful comments, and we apologize for the overuse of technical terminology in the original manuscript. This point gets at many of the other comments you have raised, so we will point to a few of the major criticisms and the corresponding spots in the revised manuscript where substantial edits and additions have been made.

- We have added discussion of both the Mills et al. (2019) paper which also compares CO₂ and temperature reconstructions for both the GEOCARB and COPSE long-term carbon cycle models, and the Farnsworth et al. (2019) paper using paleo temperature reconstructions, and we position our work relative to these previous studies at (for example) lines 333 and 362 in the revised manuscript.
- On a related note, we have added a brief section to describe the carbon balance and temperature calculation within GEOCARB beginning at line 109, but for the sake of brevity in a short-form journal article must refer the reader to previous, more extensive, documentation of the GEOCARB model structure for further details (Berner, 2004; referred to at line 110).
- Later, we describe the temperature and CO₂ precalibration experiments (line 142). At line 154, we reference avoiding the issue the original work faced of overconstraining the system/overconfidence and include a Supplementary Figure to show the diverse uncertainties that pose this issue.
- We discuss the issue of state dependence (per the von der Heydt et al. (2016) paper) now in the revised manuscript at line 102 and again in the Discussion at line 359 (discussed in greater detail to the specific comment on this matter below).

Third, and this comes back a bit (in different wording) to my second point, I remain confused about the serious amount of mismatch between the data and the model hindcast for CO₂ (Figure 3), which is also visible throughout Supplementary Figure 1. The model hindcast seems to strongly underestimate major, dense clusters of data. Yet this lack of agreement is not discussed. The study seems to simply proceed with the model hindcast. Does this reduced variability in hindcast CO₂ relative to the data not risk underestimating ESS (DeltaT/F_2xCO₂)? Lines 113-116 give us some vague inkling on what's happening: "the essence of the calibration method is to iteratively sample model parameters, and "steer" the sampling toward values of parameters (including ESS) that yield simulations that agree well with the CO₂ proxy data." This suggests to me that the model effectively varies (for calibration) a bunch of parameters (as is well described, and also in the tables), which include ESS. It then uses the calibration to determine which CO₂+Temp histories best agree with the "assumed time series of isotopic forcing", which is mentioned only as late as Line 180. In this sense, it appears to me that it is the carbon isotope record that the model really tries to fit, and that CO₂, ESS, and T histories are what's diagnosed (and compared with the CO₂ proxy data; but why not also compared with T data?). This – or the correct description, if I got it wrong – really, really needs explaining. At the moment, the paper is like a bowl of spaghetti that twists and twirls around without apparent beginning or end.

Thank you for bringing attention to this important point. It indeed highlights one of the dangers of the parametric approach we took in using the Bayesian calibration scheme in the original analysis. Namely, the selection of the mixture model likelihood function, which was strongly multimodal during certain time periods (e.g., about 240 Myr ago), leading to overconfidence in the major mode for low-CO₂ concentrations and underconfidence in the minor mode with high-CO₂ concentrations. You are quite correct that biases that linger in the CO₂ simulation risk leading to biases in the ESS estimate.

We avoid this model overconfidence in the revised work by opting to use a precalibration approach, in which calibration “windows” are constructed for each 10 Myr time slice as follows. We pool all of the data points in each time slice. For these data, we construct the upper bound for the time slice’s precalibration window by taking each data point and adding +1sigma to it (where each data point may have a difference uncertainty sigma), then taking the maximum among all of the data+1sigma values. A lower bound for each window is calculated in a similar manner, taking -1sigma from each data point. We then measure the model goodness-of-fit for a given set of input parameters using the percentage of time steps where the model misses a precalibration window; lower values of this “%outbound” are better. This follows the work of Mills et al. (2019), who pursue a similar type of precalibration, although those authors do not use a Monte Carlo approach (preserving the novelty of our approach). We use this precalibration for both temperature and CO2. We concisely describe the precalibration procedure in the revised text at line 145, specifically, how our estimates for ESS arise:

The essence of this calibration method is to sample a large number of model parameter sets from their prior distributions – these are the a priori parameter values, taken before any data are fused with the model. Then, we rule out any combinations of parameters that yield simulations that do not agree well with the CO2 proxy or temperature data, given their uncertainties. What remains are the a posteriori ensembles of parameters, including S.

We have also added a model workflow diagram to help clarify the role of different parameters, input time series and output time series, as well as what is adjusted during the (pre)calibration procedure. This new figure can be found near line 173 in the revised manuscript.

Moreover, where can I see the model’s grand master time series, the “assumed time series of isotopic forcing”? It’s downright strange not to show such a crucial piece of information, and to not explain (briefly) how it was developed.

This is a great suggestion. We have added a new Supplementary Figure 1 that shows the 12 time series inputs, including their central estimates and uncertainty ranges throughout the model hindcast period. We include both the a priori ranges and the a posteriori ranges from after the precalibration. We have added the following text (line 396) to briefly describe the 12 time series parameters, but we direct the reader to Royer et al 2014 for more information because these parameters have been described extensively elsewhere and to include fuller descriptions for even the 12 time series parameters (as in Royer et al 2014) would substantially add to the length of the manuscript.

There are 68 parameters in total: 56 constant parameters and 12 time series parameters. The time series parameters include isotopic ratios for strontium ($^{87}\text{Sr}/^{86}\text{Sr}$, to track the weathering fraction of volcanic rocks), carbon and sulfur isotope ratios ($\delta^{13}\text{C}$ and $\delta^{34}\text{S}$, to track burial, degassing and weathering fluxes); paleogeographical factors (including continental relief, total land area, land area susceptible to weathering, land area covered by carbonates, river runoff and the effect of paleogeographical changes on temperature); and degassing and

seafloor spreading. The parameters are described along with their prior and posterior ranges in the Supplemental Material accompanying this work, and in much greater detail in Royer et al.[13].

If the above is a reasonable approximation of what the model in its most basic form does, then CO₂ and T are per definition assumed to be highly correlated in the model, with only the glacial parameter (GLAC) imposing a change in time in that correlation. This brings me to a Fourth issue: such an imposed correlation may be flawed because there are two types of potential state dependence in climate sensitivity, and I am concerned that the model may be wrapping potentially (strongly) state-dependent estimates over a very long period of Earth history into one ESS range.

The GLAC parameter helps only to some extent with this problem. GLAC deals with so-called Type II state dependence (von der Heydt et al., 2016 Current Climate Change Reports; a study that really needs to be incorporated/addressed in some detail in the present work), which concerns bifurcations/multiple stable states issues between glaciated and non-glaciated states.

This is an excellent point to add to the framing of GEOCARB's GLAC parameter, and the assumptions and limitations of this formulation for the potential state-dependence of the Earth system sensitivity. We have added a note about this to the revised manuscript at line 102 (excerpt given below), and we apologize for the oversight in the initial version of the manuscript.

The two stable states, glacial and non-glacial, for S within GEOCARB provide a simple representation of the Type II state dependence described by von der Heydt et al.[31]. However, temporal variation in S within each of those stable states is not represented in GEOCARB. Some previous modeling efforts have assumed a single value of S for multiple climate states (e.g., glacial and non-glacial)[22]. This will generally increase the uncertainty in the resulting scalar parameter estimate for S because of the addition of temporal representation uncertainty.

However, the other type (Type I) of state dependence of von der Heydt et al (2016) is not considered at all well, in my view. Type I concerns the potential for changes in feedback efficacies, which can be gradual, or step-wise. For example, there may be water vapour and shortwave cloud feedback efficacy jumps in warm states (Caballero and Huber 2013 PNAS; Zhu et al., 2019 Science Advances; and refs therein), or ice-albedo efficacy decreases under increasing cold states (Stap et al., 2019 Earth System Dynamics; and refs therein). Such Type I efficacy changes would sit folded within the warm state (warm versus very warm) or cold state (cold versus very cold). This would cause changes/spread in the diagnosed ESS for warm states and cold states because the temperature responses to CO₂ changes would be strengthened or weakened. One might say: "so the estimates found for warm and cold states can be further refined/narrowed by looking in detail inside them." But that would be true only if the amalgamated warm or cold state values were randomly representative for the entire warm or cold range; i.e., if the information presented in the current manuscript had no bias at all into any direction within the warm state (e.g., toward very warm, or just warm), or within the cold state (e.g., toward very cold, or just cold). And I am not convinced about that. The model CO₂ seems to sit systematically below major clusters of proxy CO₂ in several intervals, suggesting that the model ESS may be systematically biased to low-forcing states in those intervals.

Thank you for bringing up this important point. Undoubtedly if the CO₂ simulation is indeed biased, the ESS values upon which that CO₂ time series is conditioned may be to blame for this mismatch. Similarly for the model output for temperature (which we now include, as described in our response to one of your (and the other reviewers') previous points). In the revised experimental setup, we use a precalibration windowing approach instead of the formal likelihood function, which led to the overconfidence and apparent low CO₂ bias in the original work. The revised precalibration approach has corrected the issue of overconfidence, but your point about Type I state dependence remains a valid criticism of the GEOCARB model and indeed other similar long-term carbon cycle models, in which a single (or two) values of ESS (S) is to be chosen which should be representative of an entire climate state - certainly there is temporal variability within each state. To characterize this temporal variability, Mills et al. (2019) conduct an experiment using GEOCARB in which they back-calculate what ESS value matches their temperature data set for each point in time. Their reconstructed ESS time series (Mills et al., 2019, Figure 6A) shows this variation. The glacial state in GEOCARB is assumed to be 330-260 Myr ago and 40-0 Myr ago, which coincides well with the periods during which the ESS estimates of Mills et al. (2019) are highest. Thus, this captures the notion of the Type II state dependence (per your previous comment), and the temporal variability seen within each state in the reconstruction of Mills et al. (2019) demonstrates the Type I state dependence.

However, the results of Mills et al. also show a Type I state dependence within the non-glacial state, with a higher ESS around 100 Myr ago. This coincides well with a period in which our results using the "out-of-box" GEOCARBSULFvolc model show a low bias relative to the temperature data compilation (the only spot where there is substantial disagreement). So we have added a Supplemental sensitivity experiment wherein the ESS factor (the parameter ΔT_X) in GEOCARB for the period 130-40 Myr ago is adjusted by a multiplicative factor that changes linearly from 1 at 130 Myr ago to be equal to the glacial scaling factor (GLAC) at 40 Myr ago. 40 Myr ago coincides with the beginning of the second glacial period in GEOCARB. This experiment constitutes a linear shift from non-glacial to glacial period as opposed to the step function in the "control" GEOCARB version. This choice is meant to strike a balance between over-parameterizing the model but still accounting for the large-scale state dependence changes seen in the results of Mills et al. While on its face, it may seem that this experiment falls victim to the very type of analysis you have cautioned us against (looking in greater detail inside the warm state), there is a bias in the model output temperatures, and we hypothesize in the revised work that this bias is attributable to poor representation of the Type I state dependence that you describe. However, the result of this sensitivity experiment shows little change in the model output temperatures, which suggests that our results are not sensitive to the coarse representation of state dependence within GEOCARB.

We discuss this in the revised manuscript at line 359:

As previously discussed[31], using a constant S for each of the glacial and non-glacial stable climate states risks missing Type I state dependence in S, wherein feedback efficiencies gradually change within one of these larger stable

climate states. The default glacial periods in GEOCARB (330-260 Myr ago and 40-0 Myr ago) correspond well to periods of highest S from the time series for S developed by Mills et al.[9]. However, beginning about 130 Myr ago, this S time series[9] shows an increasing trend. To evaluate the impacts of this Type I state dependence, we perform a supplemental experiment in which we linearly increase S from its non-glacial value at 130 Myr ago to its glacial value at 40 Myr ago. We find that the linear change in S (as opposed to the step function transitions in the control model) has very little effect on the model hindcast for temperature (Supplementary Figure 6). While this experiment is not an exhaustive treatment of Type I state dependence for ESS, this result suggests that this structural limitation of GEOCARB does not substantially affect our results.

Fifth, I sense that the sensitivity tests discussed around Figure 4 are all about instantaneous relationships within given time slices. However, for some of the slower processes, would you not expect the correlations to strengthen with some appreciable amount of time-lag? Could lagged correlations be included in the sensitivity tests?

We agree that examining correlation (and autocorrelation) in the processes involved would be an excellent additional sensitivity test. We have removed the global sensitivity analysis (former Figure 4) in light of suggestions by Reviewer 3, and in an effort to make the analysis more transparent and “digestible”. However, we have replaced it with a simpler analysis of correlations (Pearson and Spearman rank) for the input parameters and time series. The relative strength of examining the correlations is to provide a sense of the direction of each correlation, as opposed to simply magnitude, as in the original analysis. In the revised analysis, we have computed the Spearman and Pearson correlations between each time step’s value for each of the time series parameters and S, and find no correlations above about 0.05. We mention this in the revised manuscript at line 221 (below) but do not provide any tables for the reader because the matrices involved have 696 dimensions (12 time series parameters times 58 time steps).

At all time steps, all of the time series parameters have correlations with S weaker than about 0.05.

I think a lot of my problems/confusions come from the opaque presentation of the paper, which (1) does not outline how the model does what it does, (2) does not present the “master timeseries” from which the model through all its manipulations determines T, CO₂ and ESS, (3) does not carefully and consistently use terminology throughout; (4) obscures important information by using unnecessary jargon; (5) does not rigorously discuss the large offsets between model hindcast CO₂ and proxy CO₂, and how these mismatches may affect the conclusions; and (6) shows no temperature outputs with comparison to proxy data (or analysis of differences between these). All of this can be fixed. And then this could be a really important contribution.

Thank you for your many constructive comments and feedback. Our responses elsewhere provide more specific details on how the revised manuscript addresses these concerns, but here we provide a summary to the 6 items given.

- 1) We have added a description of the carbon mass balance and the temperature calculation in the GEOCARB model (beginning around line 109 in the revised manuscript).
- 2) We have added a Supplementary figure that displays the 12 input time series parameters and their uncertainty ranges (Supplementary Figure 1).
- 3) We have clarified what we term to be “S” (ESS as represented by GEOCARB) is $S[\text{CO}_2, \text{GEOG}, \text{solar}]$ and what is “ESS” (the general, true Earth-system sensitivity) throughout the manuscript (at line 80).
- 4) We have both simplified the analysis and attempted to simplify the explanation of the analysis in the revised manuscript. We were trying to pack too much into the original work (the calibration and sensitivity analysis), while sidestepping some issues (CO₂ mismatch to data) that are deserving of greater attention.
- 5) By simplifying the analysis, we are able to focus the revised manuscript on the CO₂ model-data mismatch, as opposed to focusing much of the discussion on the sensitivity experiments.
- 6) The simpler analysis also provides an opportunity to incorporate temperature data in an additional calibration experiment, alongside the CO₂ data. The revised manuscript now focuses on how tighter CO₂ and temperature constraints lead to a shift in and tighter constraint on estimates of ESS.

Finally, there is a really weird statement in line 169: “further investigation is outside the scope of this study.” Really? This statement is about overconfidence in the model’s CO₂ reconstruction, and offsets from proxy data. Those matters could blow out the reconstructed ESS ranges by quite some margin. I would argue that some of this comes into the Type I state dependence sector. If the authors wish to present believable ESS ranges, then this is exactly what should be tackled in the present study – it cannot be put off to a next study as it is fundamental to the ESS ranges reported, which are the central conclusion of the study.

This is a very good point and has helped to reshape the revised manuscript to focus on rectifying the model-data mismatch for both CO₂ and temperature. The revised manuscript digs deeper into a discussion of avoiding model overconfidence and bias issues, and what model improvements can reduce these biases. As the revised experimental set-up covers the range of plausible CO₂ values, the results do not indicate that a process is missing from GEOCARB, but rather the data are simply diverse and at times conflicting (e.g., the very wide uncertainty range seen around 240 Myr ago). We discuss this in the revised manuscript at line 380:

Our use of a precalibration approach to avoid overfitting data points with low CO₂ concentrations minimizes the low-CO₂ bias found throughout the Mesozoic Era characteristic of previous GEOCARB analyses[12,13]. Indeed, when we fit a mixture model distribution to the CO₂ proxy data, this distribution reveals strong multimodality in the CO₂ proxy record (see Supplementary Figure 7). This multimodality is a likely culprit for the low-CO₂ bias observed in previous work[12,13], as formal calibration procedures (e.g., Markov chain Monte Carlo[22]) will improve the model fit to the data by tuning the model to better represent modes in the data that have narrower uncertainty ranges at the expense of adequately representing data points with higher uncertainties (Supplementary Figure 7a).

Supplementary Figure 7a:

A most final niggle: The reference list is a bit of a mess, and needs consistent formatting, with correct journal abbreviations, and with the “2” in CO₂ and O₂ written in subscript. Also, in Ref 21, eocene should be capitalised.

We apologize for this oversight and thank the reviewer for pointing out the formatting issues. The references section has been carefully combed for consistent formatting and abbreviations.

Reviewer #3 (Remarks to the Author):

This paper describes a nice approach to fitting a long-term carbon-cycle model to Phanerozoic CO₂ estimates while varying a multitude of parameters in a MCMC procedure. Included in those parameters is the Earth System Sensitivity (ESS - the long term climate response to a doubling of carbon dioxide levels), and so a posterior pdf on ESS can be constructed. Overall this is worthwhile and interesting, but there are a few issues that could be better addressed.

1) There is an overall lack of clarity in what is being done (or rather, how the study is framed). ESS is not dependent on the carbon cycle - it is the theoretical long-term response to fixed CO₂ change. The work here uses variations in the CC to get a better match to the inferred CO₂ levels that then allow a best guess ESS to be determined. Thus this procedure to constrain ESS depends on the CC, but the ESS *sensu stricto* does not. There are multiple places where the text seems to imply there is a direct influence of the carbon cycle on ESS.

We apologize for the confusion here. We of course do not mean to imply that the ESS is dependent in some way on atmospheric carbon concentrations, or some other odd dependence structure. We have added a workflow schematic diagram (at line 173 of the revised manuscript) to depict the overall workflow and which factors are inputs and which are outputs. We have clarified in the revised manuscript (line 87, below) that we use the carbon cycle to diagnose what values of ESS are consistent with the observed temperature and CO₂ proxy records.

The inverse approach generates model simulations using many different plausible values for S to determine which values for S are likely, given the (mis)match between the resulting model output for simulated CO₂ and temperature.

2) There are a few issues with the model setup that I think could be more usefully explained:

i) Why is the response to solar insolation (Ws) different from ESS? (in some suitably scaled way). These should scale. If one wanted the flexibility to have the response to solar be more or less efficient than CO₂, it would be better to set $W_s = \text{Seff} \cdot \text{ESS}$ - but the prior on Seff would be quite tight. Otherwise one might end up with a situation where high ESS is bizarrely matched to low Ws which does not seem physical. What is Ws in any case? (I note the SM does not include units nor the 'best guess' values for any of the parameters other than ESS. This could usefully be added I think).

Thank you for bringing up this great point for clarification. First, we clarify in the revised manuscript that the ESS we are working with is $S[\text{CO}_2, \text{solar}, \text{geog}]$, so the solar luminosity is considered to be an external forcing (line 74):

The GEOCARBSULFvolc model and its previous incarnations[23,24] have been widely used in previous studies [e.g., refs. 12,13,17,25,26], and includes a version of ESS where the only independent radiative forcings are CO₂, solar evolution and changing geography ($S[\text{CO}_2, \text{geog}, \text{solar}]$).

If we instead treated luminosity as internal to the climate system, then indeed it would correlate strongly with ESS. Additionally, we clarify in the revised manuscript that W_s is a parameter to account for the increase in solar luminosity over time (c.f., Berner, 2004, p. 35 eq. 2.28), and uncertainty in this effect, at line 77 of the revised text:

GEOCARBSULFvolc assumes a linear increase in solar luminosity over time, corresponding to the parameter W_s , and uses results from general circulation model output to simulate the temperature change resulting from changes in paleogeography (GEOG; see Supplementary Figure 1)[27].

We have also revised the Supplementary tables of parameter values and names to include the central estimates and uncertainty ranges of the a priori parameter estimates and the a posteriori parameter estimates. These tables are now given in an Excel spreadsheet instead of textual tables to more comfortably fit more information.

ii) GEOG is the input time-series of normalized land temperature over 570 Ma. How are the uncertainties in this dealt with? It would seem to be that this is an order one uncertainty and yet it is not discussed at all (the inclusion of it in figure 4 is not particularly illuminating). I doubt that the authors are claiming that we know such temperatures exactly... Also how was it constructed since presumably one needs to know ESS and W_s to normalise actual temperatures to present day CO₂ and solar? Looking at the Cenozoic portion for instance, it appears that the authors assume that all Cenozoic variability at 10 My steps is driven by CO₂ - but isn't that begging the question? Why isn't the normalization part of the forward model? That geography can only have such a small impact is a little surprising - how does one deal with the isolation of Antarctica by the opening of drake passage and the impact that has on glaciation?

We apologize for not providing enough details about the many forcings and parameters of the GEOCARB model, specifically in this case the GEOG parameter. From Royer et al 2014: "GEOG is the change in temperature relative to the present-day, *assuming present-day CO₂ and solar luminosity*" (see also p. 35 in Berner, 2004) (emphasis added). Since this does not change with the CO₂ simulation, new normalization is not necessary. The GEOG time series was constructed based on the GCM simulations of Otto-Bliesner (1995). We have added a note from our response to the previous comment to clarify the origin of the GEOG parameter at line 78 of the revised text.

We have also added a Supplemental figure to show the 12 input time series and their uncertainty ranges (Supplemental Figure 1). We also note that GEOG is only meant to capture the subset of land surfaces that are undergoing appreciable chemical weathering, assuming no change in CO₂ or solar luminosity. Thus, uncertainty in paleogeographical changes such as the opening of the Drake Passage, while not explicitly represented in GEOG or other GEOCARB inputs or processes, indeed contributes to uncertainty in such parameters as GEOG. We mention this potential source of uncertainty in the revised text at line 371:

The assumed time series forcing parameters may also introduce biases. For example, uncertainty in paleogeographical changes such as the opening of the Drake Passage, while not explicitly represented in the GEOCARB inputs or

processes, indeed contributes to uncertainty in such parameters as GEOG (the temperature change resulting from changes in paleogeography, assuming fixed CO₂ and solar luminosity). Additionally, GEOCARB does not explicitly account for non-CO₂ greenhouse gases or aerosols. This limitation of GEOCARB and other similar models (e.g., ref. 22) may risk overestimating S by assuming that all of the observed temperature change is attributable to the CO₂ forcing (along with paleogeography and solar luminosity in the case of GEOCARB).

Regarding the Cenozoic variability, we anticipate that the new experimental set-up will be much more transparent. Specifically, the revised analysis examines the model-data mismatch in both CO₂ and temperature throughout the Phanerozoic and how improving the CO₂ and temperature simulations affects estimates of model parameters (with a focus on ESS). In this inverse modeling approach, the model output (temperature and CO₂) is used to determine the input (uncertain parameters), so we can see how this would appear confusing. However, the model workflow diagram (which we mentioned in our response to your first major point, and added at line 173) is intended to help clarify that in the model structure, it is the input constant and time series parameters that drive variability in both CO₂ and temperature.

iii) there are other time-series used as input (i.e. line 212+). What does the uncertainty sampling mean for them? Is each time point varied independently or is there some auto-correlation? What are the sources for the baseline values? This could all be usefully added to the SM.

Thank you for pointing out this nice opportunity to clarify an important aspect of the analysis. Each input time series is sampled as follows:

- The central estimate for the vector (time series) is the time series given by previous GEOCARB work (Royer et al., 2014), or the updated time series (in the case of fSR, the seafloor spreading rate relative to present (Domeier and Torsvik, 2019))
- The covariance matrix for the time series is sampled from an inverse Wishart distribution centered at the diagonal matrix of variances for each time point (representing no autocorrelation at all), but with uncertainty in each time step is given by previous GEOCARB modeling work (Royer et al., 2014). An inverse gamma distribution is the conjugate prior for the variance of a univariate normal distribution, and an inverse Wishart distribution is the higher-dimension analog to the inverse gamma, for multivariate normal distributions.
- In summary and simpler terms: each time series parameter assumes no autocorrelation as a central estimate, but the uncertainty sampling permits deviations from this.

We describe this time series sampling in the revised Methods section at line 408:

Each of the time series parameters takes on distinct values at each of the 58 model time steps. Following previous work, we assume the model and forcing time series parameters are in steady state between model time steps[12]. Each time series parameter is sampled from a 58-dimensional (number of time steps) multivariate normal distribution, whose mean is taken to match the central estimates from previous work[13].

We also now include a supplementary figure to show the autocorrelation function with its uncertainty range for each of the 12 time series parameters, in order to visually display the uncertainty in the parameters and the degree to which each is autocorrelated (Supplementary Figure 2).

3) I understand that the authors want to not only improve constraints, but also focus attention on key uncertainties. I am not sold on figure 4 as the main vehicle for showing this though. It doesn't explain how the parameters co-vary, merely that they do. Does a bigger GYM imply a bigger ESS or smaller? Is there a compensation between W_s and ESS? etc. A matrix of sensitivities might be more informative (i.e. a bar graph for all 25 parameters showing the sensitivity for ESS over the prior range of that value).

This is a great point, and we completely agree that capturing the sign of the covariance among parameters, as opposed to just the magnitude, is important. We also recognize that there is a lot to unpack in the former Figure 4, and there is value to a simple and clear analysis of the correlations and sensitivities.

In the revised manuscript, we have removed the formal global sensitivity analysis and replaced it with an examination of the correlations among the calibrated parameters. That matrix is quite large, particularly when considering the 12 time series parameters (each of which have 58 steps within them), so we only report parameters whose correlation with the ESS parameter S or the goodness-of-fit metric for either temperature or CO₂ is at least 0.1. The goodness-of-fit measure for CO₂, for example, in the revised analysis is “%outbound”, the percentage of time steps in which the modeled CO₂ hindcast does not fall between +1 standard error above the maximal CO₂ data point for that time step, or -1 standard error below the minimal CO₂ data point for that time step (similarly to the work of Mills et al., 2019). While 0.1 seems like a low threshold for correlation, there are many parameters and many model time steps/time series. With all of these competing influences, we do not expect to see correlations close to 1.

This revised approach has the benefit of highlighting which parameters are most important in reconstructing an adequate hindcast of CO₂ and/or temperature. We discuss the results of this experiment in the revised text in a new paragraph beginning at line 206, but to summarize here for convenience:

- We find the highest correlations with S are from the solar luminosity parameter and two weathering parameters
- The solar luminosity- S correlation demonstrates a compensating effect
- None of the time series parameters show strong correlation with S

4) I'm a little concerned that the model doesn't have physics to properly match everything that is reflected in the target CO₂ estimates, and that the residuals are in some sense irreducible because of that. For instance, at the K/Pg boundary something happens to perturb the CO₂ (i.e. an asteroid) that can't be captured (I think) in the model setup (or is it wrapped into GEOG - how though?). There could be a tendency for MCMC procedure to try and match that through other means, perhaps biasing the whole result. Similarly, at the P/T and whatever is happening around 300 Myr if any of that survives the next point.

We appreciate your perspective here, because your point about MCMC potentially biasing the whole result due to over-fitting particular data points and not others is indeed what we hypothesize led to the apparent model overconfidence in our original experiments. We have revised the analysis to use a precalibration windowing approach (described in our response to the previous comment). The updated results demonstrate that the model is indeed capable of capturing the breadth of the observed CO₂ proxy record, which we discuss in the revised manuscript at line 380:

Our use of a precalibration approach to avoid overfitting data points with low CO₂ concentrations minimizes the low-CO₂ bias found throughout the Mesozoic Era characteristic of previous GEOCARB analyses[12,13].

We note also that the 10 Myr time-step in GEOCARB is sufficiently long that events like asteroid strikes would be “smoothed out” over the entire time interval. That is, the model would not resolve such “rapid” events. On the other hand, studies with a time-varying ESS would likely see a signal at the K/Pg boundary. In fact, the results of both Mills et al. (2019) and Farnsworth et al. (2019) exhibit such a signal.

5) In the methods section the authors discuss the time-scale mismatch between the CO₂ observations where there is a lot of (relatively) short-term variability and the 10 My timestep in the model. Yet all of the results show the raw CO₂ data, not the CO₂ timeseries that is actually been fit. This could be misleading - please show the actual CO₂ curve that the procedure is targeting. My ability to fit a skew-normal mixture model to estimate the joint likelihood for each timestep by eye is a little rusty.

We apologize - You are quite right that our explanation of how that model-data fitting procedure works was lacking, and we certainly did not mean to expect the reader to fit a mixture model of any type by eye. The revised analysis uses what amounts to a step-function likelihood in each time slice. We show this as the gray regions in Figure 4 of the revised manuscript, which shows the CO₂ and temperature data windows throughout the model time period. We have added Supplementary Figure 8, which shows the CO₂ precalibration windows along with the actual proxy data points.

6) line 237. I don't understand this line. Nowhere in the text above have the authors described the paleo-temperature series influence, let alone a tight link - does this refer to GEOG? or some other input? Or indeed, the output? If temperature is being output then we should see it and have it compared to some independent estimates.

We apologize for the confusion here and recognize that we did not make the temperature sensitivities clear in the original work. In the revised work, we analyze the temperature model output, and show the paleogeographical influence on temperature in Supplemental Figure 1a.

7) nowhere is there any discussion of non-CO₂ forcings - CH₄, N₂O, aerosols etc. The authors might well be assuming that they are either zero, or scale proportionally to CO₂, but either way this assumption should be explicitly stated and the consequences for the eventual interpretation of the ESS noted.

Thank you for pointing out this important detail about the model. GEOCARB does not explicitly treat non-CO₂ greenhouse gases or aerosols. This limitation of GEOCARB and other similar works (e.g., Krissansen-Totton and Catling, 2017) may risk overestimating the Earth system sensitivity by assuming all of the observed temperature change is attributable to the CO₂ forcing (along with paleogeography and solar luminosity in GEOCARB's case). We now mention this caveat in the Discussion at line 375 (using almost exactly the phrasing above):

This limitation of GEOCARB and other similar models (e.g., ref. 22) may risk overestimating S by assuming that all of the observed temperature change is attributable to the CO₂ forcing (along with paleogeography and solar luminosity in the case of GEOCARB).

Minor points:

line 29: "carbon burial rates". No. this is not part of the ESS definition regardless of what the PALEOSENS paper said. (c.f. Lunt et al, 2010, Hansen et al. 2010).

We have removed "carbon burial rates" from the text.

line 29-32. Not sure I understand this sentence.

For convenience, the original sentence in question is this:

"Indeed, a growing body of evidence suggests large fluctuations in CO₂ and temperature during the last 420 Myrs, which would allow for improved quantification of climate sensitivity and insight into factors affecting the climate response across a wide range of climate states, including both icehouse and greenhouse conditions (Fig. 1 and Supplementary Table 1) [9–15]."

We apologize for the confusion regarding climate sensitivity versus Earth system sensitivity. We of course meant this sentence to say "Earth system sensitivity" and have fixed that in the revised text. Here, we are referring to the fact that by examining both greenhouse and icehouse regimes we see time periods with large fluctuations in both CO₂ and temperature. That is, there is a large enough signal in the CO₂ and temperature record on these time-scales that the signal is distinguishable from the considerable amount of noise. The revised text addresses these issues at line 30:

A growing body of evidence indicates large fluctuations in CO₂ and temperature during the last 420 Myrs[9]. This long-term record enables improved quantification of ESS and insights into factors affecting the climate response across a wide range of climate states, including both icehouse and greenhouse conditions[10–16]. This wide range of states and variations in temperature and CO₂ are also important to help distinguish the long-term climate signal from the noise.

line 41. This is not the same quantity as the ESS being discussed above. It is equivalent to assuming that orbital forcings have no effect on the glacial-interglacial changes.

Thank you for pointing this out. We have added a note to the revised text to clarify that on the time scales considered by GEOCARB (10-Myr time step), many orbital cycles (on the order of 10s to 100s of thousands of years) are averaged out. This is at line 41 of the revised manuscript:

Previous work also shows that ESS during glacial periods may be double that of non-glacial periods, with best estimates around 6 to 8 °C[12,16]. These studies use model time steps of 10 Myr, so mechanisms such as orbital forcings, which operate on time scales of 10s to 100s of thousands of years, are “averaged out” and are not explicitly represented.

Figure 1. 'glacial' boxes are not discussed in the caption. And I don't think they are credible in any case.

Thank you for pointing out this oversight. We have removed Figure 1 from the revised manuscript, but to address your concern about their credibility, we have added text to the revised manuscript that puts the ESS values into perspective relative to the range of Krissansen-Totton and Catling (2017) of 3.7-7.5 K per CO₂ doubling, and the time series of Mills et al. (2019) ranging from 5-15 K. Both of those works spanned both glacial and non-glacial periods, with glacial periods accounting for the higher ESS values.

We have added a note (below, at line 40 in the revised text) to the Introduction to put previous work examining glacial amplification of ESS into context. Glacial amplification of ESS is a concept that has been studied for nearly 50 years, dating back to (at least) Budyko (1974; 1982).

Additionally, arguments laid out in Hansen et al.[16] and Pagani et al.[11] support a glacial amplification in ESS, giving 6 °C or more warming per doubling of CO₂.

line 169. Seems too important to leave out entirely though...

We agree that this is an important point. In our revised set of data assimilation experiments, we use a precalibration windowing approach to quantify model-data mismatch and avoid the issue of model overconfidence/over-fitting. In the original set of experiments that used a Bayesian approach and formal likelihood function, the calibration algorithm over-fit the low-CO₂ data points because the uncertainties associated with them are relatively lower than the uncertainties associated with the high-CO₂ data points (e.g., the period around 220 Myr ago). In the revised analysis, we see that the model is in much better agreement with the proxy data points (c.f. Figure 4 in the revised manuscript). Of course, due to the many conflicting data points within some of the time steps, no model is capable of simultaneously reproducing all of the data points (for example, the time slice representing 225-215 Myr ago has CO₂ data ranging from 6 to 3722 ppmv; see our new Supplementary Figure 8, pasted below for convenience).

We discuss this issue of overconfidence and the model-data mismatch, particularly when the data do not even match with other data, in the revised text at line 380:

Our use of a precalibration approach to avoid overfitting data points with low CO₂ concentrations minimizes the low-CO₂ bias found throughout the Mesozoic Era characteristic of previous GEOCARB analyses[12,13]. Indeed, when we fit a mixture model distribution to the CO₂ proxy data, this distribution reveals strong multimodality in the CO₂ proxy record (see Supplementary Figure 7). This multimodality is a likely culprit for the low-CO₂ bias observed in previous work[12,13], as formal calibration procedures (e.g., Markov chain Monte Carlo[22]) will improve the model fit to the data by tuning the model to better represent modes in the data that have narrower uncertainty ranges at the expense of adequately representing data points with higher uncertainties (Supplementary Figure 7a).

SM fig 4. This could be usefully converted into a joy plot (i.e. like the famous Joy Division cover) of stacked pdfs to allow for more data to be shown (i.e. one every 30 Ma or so).

Thank you for the excellent suggestion. Alas, the revised manuscript uses a much simpler windowing approach, so we do not pursue the mixture model likelihood function anymore. Instead, the windowing approach uses what amounts to a simple step-function likelihood, so there is no “texture” to the likelihood to be shown in a joy plot. However, we are thankful for the nice suggestion and will keep it in mind for future work.

References:

- Berner, R. A. (2004), *The Phanerozoic carbon cycle: CO₂ and O₂*, Oxford University Press, New York, NY, USA.
- Berner, R. A., and Z. Kothavala (2001), *Geocarb III: A revised model of atmospheric CO₂ over phanerozoic time*, *Am. J. Sci.*, doi:10.2475/ajs.301.2.182.
- Budyko, M. I., 1974, *Climate and Life*: New York, Academic Press, 495 p.
- Budyko, M. I., 1982, *The Earth's Climate: Past and Future*: New York, Academic Press, 304 p.
- Coiffard, C., B. Gomez, V. Daviero-Gomez, and D. L. Dilcher (2012), Rise to dominance of angiosperm pioneers in European cretaceous environments, *Proc. Natl. Acad. Sci. U. S. A.*, doi:10.1073/pnas.1218633110.
- Hansen, J., M. Sato, P. Kharecha, D. Beerling, R. Berner, V. Masson-Delmotte, M. Pagani, M. Raymo, D. L. Royer, and J. C. Zachos (2008), Target Atmospheric CO₂: Where Should Humanity Aim?, *Open Atmos. Sci. J.*, doi:10.2174/1874282300802010217.
- Farnsworth, A., D. J. Lunt, C. L. O'Brien, G. L. Foster, G. N. Inglis, P. Markwick, R. D. Pancost, and S. A. Robinson (2019), Climate Sensitivity on Geological Timescales Controlled by Nonlinear Feedbacks and Ocean Circulation, *Geophys. Res. Lett.*, doi:10.1029/2019GL083574.
- Foster, G. L., D. L. Royer, and D. J. Lunt (2017), Future climate forcing potentially without precedent in the last 420 million years, *Nat. Commun.*, 8, doi:10.1038/ncomms14845.
- Krissansen-Totton, J., and D. C. Catling (2017), Constraining climate sensitivity and continental versus seafloor weathering using an inverse geological carbon cycle model, *Nat. Commun.*, doi:10.1038/ncomms15423.
- Mann, M. E., S. Rutherford, E. Wahl, and C. Ammann (2005), Testing the fidelity of methods used in proxy-based reconstructions of past climate, *J. Clim.*, 18(20), 4097–4107, doi:10.1175/JCLI3564.1.
- Mills, B. J. W., A. J. Krause, C. R. Scotese, D. J. Hill, G. A. Shields, and T. M. Lenton (2019), Modelling the long-term carbon cycle, atmospheric CO₂, and Earth surface temperature from late Neoproterozoic to present day, *Gondwana Res.*, doi:10.1016/j.gr.2018.12.001.

Otto-Bliesner, B. L., 1995, Continental drift, runoff and weathering feedbacks: Implications from climate model experiments: *Journal of Geophysical Research*, v. 100, n. D6, p. 11537–11548, <http://dx.doi.org/10.1029/95JD00591>

Pagani, M., Z. Liu, J. Lariviere, and A. C. Ravelo (2010), High Earth-system climate sensitivity determined from Pliocene carbon dioxide concentrations, *Nat. Geosci.*, doi:10.1038/ngeo724.

Royer, D. L., Y. Donnadieu, J. Park, J. Kowalczyk, and Y. Godderis (2014), Error analysis of CO₂ and O₂ estimates from the long-term geochemical model GEOCARBSULF, *Am. J. Sci.*, doi:10.2475/09.2014.01.

Sundquist, E. T. (1991), Steady- and non-steady-state carbonate-silicate controls on atmospheric CO₂, *Quat. Sci. Rev.*, doi:10.1016/0277-3791(91)90026-Q.

Von der Heydt, A. S. et al. (2016), Lessons on Climate Sensitivity From Past Climate Changes, *Curr. Clim. Chang. Reports*, doi:10.1007/s40641-016-0049-3.

Reviewer comments, second round -

Reviewer #1 (Remarks to the Author):

This MS has largely improved and I can now also consider that this might be published. I have only the following minor points which need nevertheless clarification before acceptance.

line 23-25: Consider citing one of the most recent and condensed reviews on climate sensitivity (Knutti et al 2017), probably instead of Knutti et al 2002 (ref 8) which is a bit outdated.
Knutti, R.; Rugenstein, M. A. A. & Hegerl, G. C. Beyond equilibrium climate sensitivity Nature Geoscience, 2017, 10, 727-736

line 40: „Additionally, arguments laid out in Hansen et al.16 and Pagani et al.11 support a glacial amplification in ESS, giving 6 oC or more warming per doubling of CO2. „
and

line 102: „During glacial periods, the effective ESS is then $GLAC \times S_{SEP}$ “

I am not sure I understand the argument here, or what I understand I disagree with my knowledge.

There has been quite some work on the state-dependency of climate sensitivity, and nearly all studies show, that climate sensitivity is larger for interglacials than for glacials. This is included in the Knutti et al 2017, also in von der Heydt et al 2016 (your ref 31), and has been extensively analysed in papers by Köhler et al. (2015, 2017).

In all this efforts $S = dT/dR$ needs to account for the land ice (LI) albedo feedback, in details $S[CO_2,LI] = dT / dR[CO_2,LI]$ to become a climate sensitivity that is meaningful for comparison with the Charney sensitivity, which considers only the fast feedbacks. Thus, $S[CO_2,LI]$ considering glacials is smaller than $S[CO_2]$ in which this is ignored. Here, to my understanding your S is amplified by GLAC to a HIGHER values during glacial times.

Thus, this seems to be the opposite of what is written here and needs some explanation or correction.

I hope these opposite views can be merged into one. This at least needs to be discussed to give the readers a chance to understand it, and not to think the whole time „it should be the other way round...“.

Side-remark: The Pagani paper (ref 11) cited in line 41 has been reevaluated in Köhler et al (2015), and the deduced CO₂ in Pagani are difficult to bring into agreement with known temperature changes. Maybe revise.

Köhler, P., de Boer, B., von der Heydt, A. S., Stap, L. B., and van de Wal, R. S. W.: On the state dependency of the equilibrium climate sensitivity during the last 5 million years, *Clim. Past*, 11, 1801–1823, <https://doi.org/10.5194/cp-11-1801-2015>, 2015.

Köhler, P., Stap, L. B., von der Heydt, A. S., de Boer, B., van de Wal, R. S. W., & Bloch-Johnson, J. (2017). A state-dependent quantification of climate sensitivity based on paleodata of the last 2.1 million years. *Paleoceanography*, 32, 1102–1114. <https://doi.org/10.1002/2017PA003190>

line 96: The part

„has 68 input parameters (including both constant and time-variable parameters; see Supplementary Materials) and„

can be deleted, since this is mentioned in detail in line 132 and further below, otherwise we read a lot that there are indeed 68 parameters...

line 116: „RCO₂(t) is the CO₂ concentration at time t relative to present. „

I believe RCO_2 is the radiative forcing of CO_2 at time t relative to present (not the CO_2 concentration), and the units should be W/m^2

line 221: I do not understand this sentence „ At all time steps, all of the time series parameters have correlations with S weaker than about 0.05.“ S is not a time series itself, so how do you calculate a correlation of a scalar (single value) with a time series? Or should this mean „correlation with CO_2 ? Just guessing.”

Reviewer #2 (Remarks to the Author):

2nd review of Wong et al. “A tighter constraint on ESS from long-term temperature and carbon-cycle observations”

I find that this manuscript has considerably improved, and that overall good responses were made to the various reviewers’ comments. Still, I think some work is needed, but it’s moving rapidly in the right direction. I will go through things line by line.

L2-3. “Developing sound strategies to manage climate risks hinges critically on Earth-system properties, including the Earth-system sensitivity (ESS).” I think this is a gross overstatement, in an attempt to give a 100% palaeo-study some sort of relevance to modern climate change. First, I think that it should not be needed to give such a contrived relevance statement – the study has its own distinct relevance for understanding how the world works (on very long timescales), and doesn’t need to be somehow “made relevant” by artificially linking it to what’s happening on decadal to centennial timescales. Sound strategies for the future do not depend on ESS, which is measured over hundreds of thousands to several millions of years. They depend on transient climate response and equilibrium (fast feedback) climate sensitivity. It is unreasonable to argue that complete Earth System responses that take 10^5 or more years would be relevant to policy. In most (all, I would argue) places, policy struggles to look forward a decade or more. Also, the paper launches into ESS without ever explaining what it actually stands for. This is partly related to the “relevance” statement because it will be immediately obvious that ESS is not the “critical hinge;” there are other numbers with much greater relevance to climate management w.r.t. the modern changes.

L3. “Large and deep uncertainties.” I can understand large uncertainties, but what are deep uncertainties, and how are deep uncertainties different from large uncertainties? Large and deep sound nice and dramatic, but the meaning of this statement is rather opaque.

L11. “better agreement.” Better than what?

Abstract in general: this launches into rather specific concepts, such as “ESS”, “chemical weathering of gymnosperms” (I guess you mean mediated by gymnosperms, and not of the gymnosperms themselves, which your statement suggests?), “shift in timing of gymnosperm- to angiosperm-dominated vegetation” (which shift? Many people won’t be familiar with that in the first place, let alone understand why this might affect ESS). The abstract needs to be stand-alone, and capture the essence of the study without prior knowledge of key concepts and processes. It needs a good rewrite to get to that level.

L17. “deep uncertainties.” Do you mean large here? If not, then explain what you mean with deep uncertainties.

L20. “terms climate sensitivity.” This should be more specific and say “termed equilibrium climate

sensitivity." In the following, I abbreviate that to ECS.

L20-23. "The climate sensitivity is critical for mapping projections of CO₂ forcing to global temperature, and is based on "fast" feedback responses to changes in radiative forcing. These fast feedbacks include changes in water vapor, lapse rate, cloud cover, snow/sea-ice albedo and the Planck feedback." This is somewhat awkwardly phrased, I think. The climate sensitivity is not just critical for mapping projections of CO₂ forcing, but of all radiative forcing. ECS is a function of all forcing, and in the palaeo domain that includes "forcing" by the slow feedbacks. In palaeo, the slow feedbacks are considered forcing to make ECS estimates from palaeo studies as close as possible approximations of ECS from modern/actuo studies. It is somewhat awkward, therefore, to say that ECS (rather than "the climate sensitivity") is functional for mapping CO₂ impacts. It is functional for mapping all radiative impacts.

L25. "large and deep uncertainties" again. This need explanation if it is to be used so frequently.

L27-29. "Consideration of longer-term responses offers a glimpse into the deep-time paleoclimate evolution of climate sensitivity and can inform estimates of the related, longer-term equilibrium response, termed "Earth-system sensitivity" (ESS)." First, there is a sound definition of ESS, and it is not as loose as what is given here. You can find it in Palaeosens (2012), and in their ref. 44 (the original study to coin ESS). Second, there seems to be a considerable confusion here, in that the "longer-term equilibrium response" is here used to differentiate between "climate sensitivity" and ESS. Yet this is a unfortunate choice of words, given that the climate sensitivity referred to here is equilibrium climate sensitivity (ECS) versus ESS. It gets even worse in L29-30, when it is stated that "The ESS can include slower responses such as changes in vegetation cover, land-ice or carbon burial rates." This is nonsense. ESS DOES include all that – it wraps everything up in terms of a total response to CO₂ changes only. Moreover, the list provided are the typical slow feedbacks included in ECS already (arguably with exception of carbon burial rates). So it's not a good list to explain the difference between ECS (which the paper continues to unqualifiedly call "climate sensitivity") and ESS.

What all this demonstrates is that the paper still shifts far too freely between definitions, and stealthily makes up its own definitions that are different from established ones (without really digging into the details of why and how this is done). This is not a sound way forward, but actually steps back to the days when everything under the sun was called climate sensitivity. We sorted that out in Palaeosens, and stepping back would only be stepping back into confusion and therefore poor replicability. This should be avoided at all costs. This paper needs to outline the exact definitions, and then work in that framework. This becomes extremely important when the study starts to compare its findings with previous ones, as it then goes quite badly off the rails.

L34. "are also important" should be "is also important" (the subject in the sentence is "range", which is singular)

L39. "ESS falls between 1.6 and 5.5 deg. C (95% confidence). This is well within the range of the fast-feedback climate sensitivity from other studies using climate data from the last millennium." This is simply comparing apples with oranges. The "fast-feedback climate sensitivity" – yet another term used without explanation – in my reading refers to ECS. So ECS, in which slow feedbacks are considered forcings that are explicitly account for, is being compared here with ESS, which considered only total response versus CO₂ changes. As is clearly illustrated in palaeosens (2012), ESS values should be expected to be differ considerably from ECS values, because the latter explicitly removes the actions of slow feedbacks, whereas the former keeps those included. So when the present study sees that ESS values fall within the ranges of previous ECS values, a correction needs to be made for the slow feedback "forcings" before such a comparison even makes sense. As a result. The statement as give is meaningless. A table should be made when comparing between studies, indicating which forcings and "slow feedback "forcings" are included in the various estimates discussed. Otherwise we're discussing all these things in a confusing vacuum, and that doesn't help anybody.

L52-55. When discussing the difference between estimates from Royer and Anagnostou, it is important to be clearer still. The implication of what is stated is the two S notations is that the Royer study explicitly considered (and accounted for) solar luminosity and geographic influence

forcings, and that Anagnostou does not. When I say accounted for, this means that the Royer values were according to this notation explicitly calculating the forcing changes from these two processes, whereas Anagnostou ignored them (and thus would have found biased values). I think that's precisely also what the authors want to say here. It would be good to spell it out a bit for non-initiated readers. It's not an easy thing to wrap one's head around. Again, presenting things in a table would be even clearer when comparing studies. But there is also a problem. ESS was defined to express everything in terms of CO₂ forcing only. That means that the authors now shift their ESS definition to the Royer-style approach. This may be sensible when looking over very long timescales. But it's an ESS redefinition nonetheless. This needs to be clarified in unambiguous terms, and we also need to be shown how much that change affect comparability with previous "ESS sensu stricto" estimates (e.g., Anagnostou). Perhaps the present study could evaluate both versions of ESS and demonstrate how important it is to go with Royer's definition when looking over very long timescales?

L69-71. "By contrast, we quantify the influence of using both CO₂ and temperature proxy data, which allows us to highlight the impacts and ability of different sources of information to continue to improve our understanding of ESS." This is a very cryptic sentence when looking at its grammatical coherence: "the impact ... of different sources of information to continue to improve our understanding ..." What does that even mean? Impact upon what?

L81-82. This is not sufficiently precise. I would suggest that you say here "we will use S when referring to the specific version of ESS defined as S[CO₂, geog, solar] obtained from the GEOCARBSULFvolc model, and reserve the term "ESS" for discussion of Earth system sensitivity more generally, including estimates of the formal ESS definition, which is S[CO₂] (Lunt et al 2010; Palaeosens, 2012)." That would be more precise. It requires a statement earlier on that Royer's work redefines ESS for studies over very long timescales to S[CO₂, geog, solar] instead of the previously defined S[SO₂].

Paragraph in L154-163. Please check and state whether the Mills T-series is entirely independent of the GEOCARB model. I don't know, and don't have the reference at hand. It's just a simple check to ensure there is no circularity.

Figure 1. Very nice and helpful summary of the approach.

L211. What do you mean with CO₂ (S)?

L210-215. The passage "The positive correlation between Sand the solar luminosity parameter, Ws, seems counterintuitive at first, given that both increased CO₂ (S) and increased luminosity (Ws) should both serve to increase the temperature. However, given that the temperature at present is assumed to be fixed, higher values for Ws imply lower temperatures deeper into the past. Given that the CO₂ simulations have been calibrated to match the proxy records, to compensate for this temperature deficit, higher values for S are required" needs to be slightly extended to allow for a somewhat slower and more elaborated discussion of this point. It's too easy for a reader to get lost here. One question that comes to mind, for example, is: Isn't this purely a mathematical construct – should we not be talking about underpinning processes a bit to justify that?

Page 8 and subsequent. There's a lot of talk here about "error rates." This is an unfortunate terminology, I think, as a rate is a quantification of change with time. I don't think you mean that w.r.t. the error. I would recommend keeping the language more simple and correct, and simply say "error margins" or something like that.

Figure 4 caption. (5% range, and 90% range. I guess you mean "probability range?" It's important to remain precise and specific throughout. The 95% range terminology is colloquial and works well in discussions, but it's not as precise and specific, as written word needs to be.

The discussion on Page 9 flips between percentage outbound values (25%, 30%, 50%). I think this can be written more systematically, to avoid confusion because of flipping around between percentage values. It may only need a simple opening sentence such as "first we will look at this

% range, then relax it further to that % range," or something like it. Help the reader to stay with you. It's not an easy paper to digest.

L302-303. GYM -> 0.25 GYM. That equation has an internal conflict. I guess you mean GYM_new - > 0.25 GYM_previous or something like that. Please update to make the writing specific and precise.

L328-329. "We find that a four-fold reduction in gymnosperm weathering efficiency leads to a much better match with the paleotemperature data (Fig. 5d)." This is another example of sloppy, colloquial writing. Gymnosperm weathering efficiency mean the efficiency of weathering of gymnosperms. But you don't mean that. You mean the efficiency of weathering MEDIATED BY gymnosperms. You are not writing only for people who know the model and what you're doing. You need to be very precise and specific to say things exactly in the way that you mean them. This means that you cannot use colloquialisms as you would in discussions in the lab. You need to be much more specific and precise. Also, it would be good if somewhere in the paper you would explain to the readers what makes the difference between gymnosperm and angiosperm mediated weathering.

More importantly, I know quite well what you are doing and why, but still am left wondering if you mean here that it really is the efficiency of the weathering process by gymnosperms that reduces (i.e., you keep the same gymnosperm coverage, but for whatever reason they don't affect weathering to a lesser extent), or that the gymnosperm coverage itself drops and thus the weathering intensity that they drive (even though the weathering efficiency by each gymnosperm may have remained constant). The writing is simply too imprecise for readers to understand these subtleties. This lack of precision and specificity really needs to be fixed. I suggest that senior authors go through next versions much more seriously and critically than they apparently have done.

L365-369. This about what you're saying here from the point of view of a non-specialist who is very interested in your work. Do you think this is sufficiently elaborated, specific and precise to make your point convincingly? I would argue that it isn't. This bit needs work in my view.

L371. Insert "of" between "time series" and "forcing."

Overall: getting there, but please work on accessibility for non-specialists, by elaborating relevant points, presenting and working with established definitions/terms, avoiding colloquialisms, and making the writing more specific and precise. It will gain you many, many citations because people will understand what you're doing and discuss the subtleties in your work. Use your experienced co-authors. Make them work for the privilege, and don't put the burden of trying to clarify how you're saying things on reviewers. In other words: it's a great study, so present it more carefully so that people will see and appreciate it for what it is, rather than feeling confused.

Reviewer #3 (Remarks to the Author):

This is a completely different paper than the one I first reviewed. It is, to be clear, a better one, but it does require almost a de novo review.

The clarity of the exposition is much better, and the results presented (including in the supplemental material) are far more comprehensive and appropriate. I think that the authors are still overconfident in their results, but given the new presentation it is much easier to see (and say) why.

The overall methodology is sound, and I think these results are interesting. I would however suggest that more revisions are required before publication.

detailed critiques:

I 9. "implying increased learning" or overconfidence? Maybe delete.

I 9-14. The sensitivity analyses to deal with the clear mismatches in the Cretaceous are neither exhaustive, nor particularly convincing. I personally would not elevate this to the lead paragraph. Rather I would describe the caveats to the uncertainty estimate given above. I expand on this further below.

I 25. It's only just come out, but the WCRP assessment on climate sensitivity (Sherwood et al, 2020, doi: 10.1029/2019RG000678) is an excellent reference for the state of the art in ECS estimation.

I 30. ESS is predicated on fixed atmospheric CO₂ concentration. It does not depend on carbon burial rates. (Again).

I. 79. "resulting *solely* from paleogeography"...(does this include ice sheets?), and do the results really rely on 25 year old model results? (O-B, 1995)? Models have changed a lot since then - that model was very coarse resolution and only had a 50m mixed layer ocean! Talk to Paul Valdes and Dan Lunt for more up-to-date numbers.

I. 174 Figure 1. This probably belongs in the Supp. Material.

I 211. a correlation btw W_s and S is not at all surprising, but I think the authors would have been better building this into the model at the beginning. Specifically, the temperature response to increasing solar should not be that different to the temperature response to increasing CO₂ - both are forcings and both impact similar feedbacks. Thus W_s should be $w \cdot S$, where w would be an efficacy related to the potentially different response of the system to 4 W/m² of solar forcing compared to CO₂ forcing. The value of w would be constrained to be 0.5 to 1.5 (times whatever normalisation might be used with W_s - see point below about the Supp. data table - is it per 570My?).

A quick back of the envelope calculation shows the need. The magnitude of the solar forcing from 420 Myr, is around 5 to 6 W/m² (assuming that the sun has become 30% more active over the last 4.5 billion years) (i.e. ~ 30 W/m² change in solar output at 1 AU), which is equivalent to $\sim 3 \times$ CO₂ forcing, and so the temperature change W_s (assuming that is in K per 570Myr), would be ~ 7.3 K for the 'best guess' S of 3.4 K/2xCO₂ - close to the mean constrained value. Setting this up as part of the model will reduce the uncertainties.

I 213. "assumed to be fixed"  "is known".

I. 271 onwards. The authors are correct in highlighting the clear mismatch between the baseline cases shown in Figure 4 and the inferred Cretaceous temperatures. This mismatch is not small (up to 20 deg C!!). However, their sensitivity studies are not that informative. It's useful to realise that ACT and GYM go some way to reducing the error, but the improvement is minor and the mismatch is still clear in Figure 5h. Thus the authors need to go further - if the model cannot match the obs, then either the model is wrong or the obs are biased. The evidence for a very warm Cretaceous is pretty strong, so this points to a structural problem in the model. What could it be? CH₄? higher atmospheric pressure? an error in the GEOG estimates? I don't really know, but this should be explored by the authors in addition to the GYM/ACT variation.

Supplementary data table. This table really needs units. For instance, the mean of W_s is 7.4. Is that K? F? K per some solar luminosity change? K per billion years? Makes a big difference! (I think I can infer from the code that it is K per 570Myr, but I'm not sure that would be obvious to most readers!

Review responses

Reviewer #1 (Remarks to the Author):

This MS has largely improved and I can now also consider that this might be published. I have only the following minor points which need nevertheless clarification before acceptance.

Thank you for the encouraging feedback and the constructive comments.

line 23-25: Consider citing one of the most recent and condensed reviews on climate sensitivity (Knutti et al 2017), probably instead of Knutti et al 2002 (ref 8) which is a bit outdated.
Knutti, R.; Rugenstein, M. A. A. & Hegerl, G. C. Beyond equilibrium climate sensitivity Nature Geoscience, 2017, 10, 727-736

Thank you for the good suggestion. We have updated this reference as recommended.

line 40: "Additionally, arguments laid out in Hansen et al.16 and Pagani et al.11 support a glacial amplification in ESS, giving 6 oC or more warming per doubling of CO2. "
and

line 102: "During glacial periods, the effective ESS is then $GLAC \times S$."

I am not sure I understand the argument here, or what I understand I disagree with my knowledge.

There has been quite some work on the state-dependency of climate sensitivity, and nearly all studies show, that climate sensitivity is larger for interglacials than for glacials. This is included in the Knutti et al 2017, also in von der Heydt et al 2016 (your ref 31), and has been extensively analysed in papers by Köhler et al. (2015, 2017).

In all this efforts $S = dT/dR$ needs to account for the land ice (LI) albedo feedback, in details $S[CO_2,LI] = dT / dR[CO_2,LI]$ to become a climate sensitivity that is meaningful for comparison with the Charney sensitivity, which considers only the fast feedbacks. Thus, $S[CO_2,LI]$ considering glacials is smaller than $S[CO_2]$ in which this is ignored. Here, to my understanding your S is amplified by $GLAC$ to a HIGHER values during glacial times.

Thus, this seems to be the opposite of what is written here and needs some explanation or correction.

I hope these opposite views can be merged into one. This at least needs to be discussed to give the readers a chance to understand it, and not to think the whole time "it should be the other way round...".

We apologize for the lack of clarity in the previous version of the manuscript - this is a subtle point, and a good area for further explanation. You are, of course, correct that during glacial periods, higher ice-albedo feedback leads to additional forcing from the ice sheets, and (consistent with previous studies) $S[\text{CO}_2] > S[\text{CO}_2, \text{LI}]$. The additional land-ice forcing arises during glacial periods. Since land-ice is not one of the externally considered forcings in the GEOCARB family of models, the GEOCARB form of ESS is on the $S[\text{CO}_2]$ side of that inequality (neglecting for the moment the geography and solar forcings). During non-glacial periods, the difference is assumed to be negligible. But during glacial periods, the GLAC amplification parameter is meant to account for this inequality (amplification of $S[\text{CO}_2]$ relative to $S[\text{CO}_2, \text{LI}]$ in GEOCARB). This interpretation and argument is corroborated by the results from Mills et al. (2019), who “back-calculated” ESS for GEOCARB and found results generally consistent with these values (their Fig 6A). (“Back-calculated” here referring to rearranging the GEOCARB equations so that data for temperature and CO_2 are used to compute ESS, as opposed to ESS and CO_2 used to compute temperature.) Based on the specific climate sensitivity values plotted around 300 Mya in Fig. 2C from Rohling et al. (2012; the PALEOSENS perspective piece), values of about 1.25-1.5 for the GLAC glacial scaling factor may be reasonable values *a priori*.

We have added text to clarify this point at line 51 of the revised manuscript:

“Additionally, during glacial periods, a given CO_2 forcing will lead to a stronger temperature change due to the land ice-albedo feedback. Thus, estimates of ESS that do not explicitly account for land-ice feedbacks will necessarily be higher than those that do. Arguments in (for example) Park and Royer¹⁴ and Hansen et al.¹⁸ support such a “glacial amplification” in ESS, giving 6 °C or more warming per doubling of CO_2 .”

We have also added two recent references relevant to the state-dependence of Earth-system and climate sensitivity:

Anagnostou, E., John, E. H., Babila, T. L., Sexton, P. F., Ridgwell, A., Lunt, D. J., Pearson, P. N., Chalk, T. B., Pancost, R. D., & Foster, G. L. (2020). Proxy evidence for state-dependence of climate sensitivity in the Eocene greenhouse. *Nature Communications*. <https://doi.org/10.1038/s41467-020-17887-x>

Tierney, J. E. et al. (2020). Past climates inform our future. *Science*, 370(6517), eaay3701. <https://doi.org/10.1126/science.aay3701>

Side-remark: The Pagani paper (ref 11) cited in line 41 has been reevaluated in Köhler et al (2015), and the deduced CO_2 in Pagani are difficult to bring into agreement with known temperature changes. Maybe revise.

Köhler, P., de Boer, B., von der Heydt, A. S., Stap, L. B., and van de Wal, R. S. W.: On the state dependency of the equilibrium climate sensitivity during the last 5 million years, *Clim. Past*, 11, 1801–1823, <https://doi.org/10.5194/cp-11-1801-2015>, 2015.

Köhler, P., Stap, L. B., von der Heydt, A. S., de Boer, B., van de Wal, R. S. W., & Bloch-Johnson, J. (2017). A state-dependent quantification of climate sensitivity based on paleodata of the last 2.1 million years. *Paleoceanography*, 32, 1102–1114. <https://doi.org/10.1002/2017PA003190>

Thank you for pointing this out. In response to your comment about Lines 40 and 102, we have expanded the text to better explain the glacial amplification. We have also removed the reference to Pagani et al. (2010).

line 96: The part

"has 68 input parameters (including both constant and time-variable parameters; see Supplementary Materials) and"

can be deleted, since this is mentioned in detail in line 132 and further below, otherwise we read a lot that there are indeed 68 parameters...

This has been removed from the revised manuscript.

line 116: "RCO₂(t) is the CO₂ concentration at time t relative to present. "

I believe RCO₂ is the radiative forcing of CO₂ at time t relative to present (not the CO₂ concentration), and the units should be W/m²

Thank you for making the valuable point about RCO₂. We apologize for the apparent lack of clarity in the manuscript. The use of RCO₂ that you describe is in line with more common notation for CO₂ (radiative) forcing. Our manuscript employs the notation of Berner (2004, see his Table 1) which defines RCO₂ as the ratio of the mass of CO₂ in the atmosphere at time t to the mass of CO₂ in the atmosphere at time 0 (present, or last model time-step). We have revised the text at Line 145 of the revised manuscript to be more in line with the Berner definition by using the word “mass” instead of “concentration”:

“... where \$T(t)-T(0)\$ denotes the global mean surface temperature at time t (Myr ago) relative to present (\$t=0\$ ) and \$RCO_2(t)\$ is the mass of atmospheric CO₂ at time t relative to present.”

line 221: I do not understand this sentence "At all time steps, all of the time series parameters have correlations with S weaker than about 0.05." S is not a time series itself, so how do you calculate a correlation of a scalar (single value) with a time series? Or should this mean "correlation with CO₂"? Just guessing.

We apologize for the confusion. We were referring to the correlation for each point in time, between that time series' value *at that time* and ESS. For example, if v is a time series, then we would examine the correlations at time $t=570, 560, \dots, 0$ Mya: $\text{corr}(v(t), \text{ESS})$. We have removed this from the revised manuscript in order to make space for the new experiment examining the temperature bias in the Cretaceous period.

Reviewer #2 (Remarks to the Author):

I find that this manuscript has considerably improved, and that overall good responses were made to the various reviewers' comments. Still, I think some work is needed, but it's moving rapidly in the right direction. I will go through things line by line.

We thank you for your detailed and encouraging comments. They continue to be extremely constructive and helpful in highlighting areas of the work where we can better explain things and make the work more accessible to a broader audience.

L2-3. "Developing sound strategies to manage climate risks hinges critically on Earth-system properties, including the Earth-system sensitivity (ESS)." I think this is a gross overstatement, in an attempt to give a 100% palaeo-study some sort of relevance to modern climate change. First, I think that it should not be needed to give such a contrived relevance statement – the study has its own distinct relevance for understanding how the world works (on very long timescales), and doesn't need to be somehow "made relevant" by artificially linking it to what's happening on decadal to centennial timescales. Sound strategies for the future do not depend on ESS, which is measured over hundreds of thousands to several millions of years. They depend on transient climate response and equilibrium (fast feedback) climate sensitivity. It is unreasonable to argue that complete Earth System responses that take 10^5 or more years would be relevant to policy. In most (all, I would argue) places, policy struggles to look forward a decade or more.

Also, the paper launches into ESS without ever explaining what it actually stands for. This is partly related to the "relevance" statement because it will be immediately obvious that ESS is not the "critical hinge;" there are other numbers with much greater relevance to climate management w.r.t. the modern changes.

Thank you for the suggestion - this is helpful for framing the study. We appreciate your point of view that the paleo perspective is useful in its own right. We have revised the first sentence to now read:

"The long-term temperature response to a given change in CO_2 forcing, or Earth-system sensitivity (ESS), is a key parameter quantifying our understanding about the

relationship between changes in Earth's radiative forcing and the resulting long-term Earth-system response. Current ESS estimates ..."

L3. "Large and deep uncertainties." I can understand large uncertainties, but what are deep uncertainties, and how are deep uncertainties different from large uncertainties? Large and deep sound nice and dramatic, but the meaning of this statement is rather opaque.

We again apologize for the confusion. By "deep uncertainty", we are referring to the state of affairs in which experts cannot or will not agree on the set of possible outcomes, the consequences of those outcomes, or the probabilities associated with each outcome (Langlois & Cosgel, 1993; Walker et al., 2013). However, this confusion has led us to determine that the subject of deep uncertainty is not central to our study or its findings, and is only adding confusion. So, to hopefully improve the clarity of our manuscript, we have removed this topic. This line now reads:

"Current ESS estimates are subject to sizable uncertainties."

L11. "better agreement." Better than what?

Our experiments for examining and reducing the temperature bias during the Cretaceous period have changed, so this is no longer in the revised manuscript.

Abstract in general: this launches into rather specific concepts, such as "ESS", "chemical weathering of gymnosperms" (I guess you mean mediated by gymnosperms, and not of the gymnosperms themselves, which your statement suggests?), "shift in timing of gymnosperm- to angiosperm-dominated vegetation" (which shift? Many people won't be familiar with that in the first place, let alone understand why this might affect ESS). The abstract needs to be stand-alone, and capture the essence of the study without prior knowledge of key concepts and processes. It needs a good rewrite to get to that level.

Thank you for the good suggestion. We have heavily revised the abstract in an effort to bring it to the level of broad accessibility that you describe.

L17. "deep uncertainties." Do you mean large here? If not, then explain what you mean with deep uncertainties.

We apologize for not being clear here. We mean "deep uncertainty" in the same sense as the time this came up in the abstract (your comment on L3 above). We have revised the text to avoid the discussion of deep uncertainties altogether. We feel that while it is an interesting and relevant topic, it is presently distracting from our main message.

L20. "terms climate sensitivity." This should be more specific and say "termed equilibrium climate sensitivity." In the following, I abbreviate that to ECS.

Thank you for pointing out this typo. It has been corrected.

L20-23. “The climate sensitivity is critical for mapping projections of CO₂ forcing to global temperature, and is based on “fast” feedback responses to changes in radiative forcing. These fast feedbacks include changes in water vapor, lapse rate, cloud cover, snow/sea-ice albedo and the Planck feedback.” This is somewhat awkwardly phrased, I think. The climate sensitivity is not just critical for mapping projections of CO₂ forcing, but of all radiative forcing. ECS is a function of all forcing, and in the palaeo domain that includes “forcing” by the slow feedbacks. In palaeo, the slow feedbacks are considered forcing to make ECS estimates from palaeo studies as close as possible approximations of ECS from modern/actuo studies. It is somewhat awkward, therefore, to say that ECS (rather than “the climate sensitivity”) is functional for mapping CO₂ impacts. It is functional for mapping all radiative impacts.

Thank you for this helpful suggestion. We have revised this to read (at Line 22 in the revised manuscript):

“The ECS is critical for mapping changes in radiative forcing, including CO₂ and other greenhouse gases, to changes in global temperature. ECS is based on “fast” feedback responses to changes in radiative forcing, including changes in water vapor, lapse rate, cloud cover, snow/sea-ice albedo and the Planck feedback⁴.”

L25. “large and deep uncertainties” again. This need explanation if it is to be used so frequently.

This has been removed from the revised version of the manuscript.

L27-29. “Consideration of longer-term responses offers a glimpse into the deep-time paleoclimate evolution of climate sensitivity and can inform estimates of the related, longer-term equilibrium response, termed “Earth-system sensitivity” (ESS).” First, there is a sound definition of ESS, and it is not as loose as what is given here. You can find it in Palaeosens (2012), and in their ref. 44 (the original study to coin ESS). Second, there seems to be a considerable confusion here, in that the “longer-term equilibrium response” is here used to differentiate between “climate sensitivity” and ESS. Yet this is a unfortunate choice of words, given that the climate sensitivity referred to here is equilibrium climate sensitivity (ECS) versus ESS.

Thank you for pointing out this area to make clear the consistent application of this previous definition. We have revised this first sentence to (1) make clear the distinction between ESS and the fast-feedback sensitivity discussed in the previous paragraph, and (2) make clear the definition for ESS from Lunt et al. (2010) (ref. 10 in the text below). This in the revised manuscript at Line 34:

“In contrast to the shorter-term ECS that responds to relatively fast feedback processes, consideration of longer-term responses offers a glimpse into the deep-time paleoclimate

evolution of the sensitivity of the Earth-system temperature response to both fast and slow feedbacks. In particular, a deep-time perspective offers insight into the ‘Earth-system sensitivity’ (ESS) - the long-term equilibrium surface temperature response to a given CO₂ forcing, including all Earth-system feedbacks¹⁰.”

It gets even worse in L29-30, when it is stated that “The ESS can include slower responses such as changes in vegetation cover, land-ice or carbon burial rates.” This is nonsense. ESS DOES include all that – it wraps everything up in terms of a total response to CO₂ changes only. Moreover, the list provided are the typical slow feedbacks included in ECS already (arguably with exception of carbon burial rates). So it’s not a good list to explain the difference between ECS (which the paper continues to unqualifiedly call “climate sensitivity”) and ESS. What all this demonstrates is that the paper still shifts far too freely between definitions, and stealthily makes up its own definitions that are different from established ones (without really digging into the details of why and how this is done). This is not a sound way forward, but actually steps back to the days when everything under the sun was called climate sensitivity. We sorted that out in Palaeosens, and stepping back would only be stepping back into confusion and therefore poor replicability. This should be avoided at all costs. This paper needs to outline the exact definitions, and then work in that framework. This becomes extremely important when the study starts to compare its findings with previous ones, as it then goes quite badly off the rails.

Please accept our apologies for the confusion here. Of course, it is not our intention to stealthily make up our own definitions. We appreciate the opportunity to clear up the apparent misunderstandings.

Our ESS is the same ESS from the Royer et al. (2012) and Park and Royer (2011) studies which were cited in the PALEOSENS perspective paper (Rohling et al., 2012). There is no redefinition of ESS. Here, we would have paleogeography and solar luminosity, in addition to CO₂ of course, as explicitly leading to the temperature response computed from GEOCARB. The calculation gives the temperature deviation relative to present as our ESS parameter (ΔT_{2x}) times the log of the atmospheric CO₂ concentration relative to present, plus the paleogeography effect on temperature (the GEOG parameter) and the solar luminosity effect on temperature (the Ws parameter, times a linear effect from time). This is described in Berner (2004), his Equation 2.28:

$$T(t) - T(0) = \Gamma \ln RCO_2 - Ws(t / 570) + GEOG(t) \quad (2.28)$$

So, the ESS in GEOCARB is accounting for changes in paleogeography and solar luminosity. You are of course quite right that to be completely consistent with the Lunt et al. and PALEOSENS definition for S[x,y,z], everything must be wrapped up in terms of a total response to CO₂ changes only. We completely agree that the $\Gamma \ln(RCO_2)$ term, where RCO₂ includes only the CO₂ forcing explicitly, is in some sense similar to what should be S[CO₂]. However, the total temperature response computed by our model

includes the effects of solar luminosity and paleogeography changes. We infer probable values for ESS (related to the Γ parameter in Eq. 2.28 above) by matching model output temperatures to observational constraints. If we were using $S[\text{CO}_2]$ to do this, ignoring solar evolution, for example, would lead to a lower estimated CO_2 , because the assumed temperature input would be higher. This, in turn, would lead to more silicate weathering (and thus lower CO_2). So, if we ignore changes in radiative forcing due to solar evolution and continental paleogeography, the estimated CO_2 would be quite different, which would in turn affect ESS because the tuning between GEOCARB and proxy CO_2 and temperature would be different. Thus, $S[\text{CO}_2]$ is not quite the right way to describe the GEOCARB form of ESS.

We also realized that our notation was not consistent with that laid out in Lunt et al. (2010) and Rohling et al./PALEOSENS (2012) because we used $S[x,y,z]$ to denote a version of our ΔT_{2x} parameter. The units are not consistent. We apologize for this regrettable notational error. As is obvious from the units in both Lunt et al. and the PALEOSENS paper, ESS has units of temperature change [deg C], and S (the specific (paleo)climate sensitivity) has units of temperature change per radiative forcing [deg C/(W/m²)].

To summarize, neither $S[\text{CO}_2]$ nor $S[\text{CO}_2, \text{geog}, \text{solar}]$ gives a precise description of how GEOCARB accounts for solar and paleogeography forcings explicitly in its calculation of the temperature response to a given CO_2 forcing, and the use of S to describe our resulting temperature response for a doubling of CO_2 forcing leads to an unfortunate mix-up of units. These issues have been addressed in the revised manuscript by using the term “ESS parameter (within GEOCARB)” and/or “ ΔT_{2x} ” to refer to the specific form of ESS represented in GEOCARB, and using “ESS” (alone, with no qualifiers) when discussing Earth-system sensitivity more generally. While “ ΔT_{2x} ” is not an established convention, we anticipate that you will agree that using something like $\text{ESS}[\text{CO}_2, \text{geog}, \text{solar}]$ would also be a bad way to go, as would using simply $S[\text{CO}_2, \text{geog}, \text{solar}]$ (or $S[\text{CO}_2]$) because of the units issue and conflating ESS with specific (paleo)climate sensitivity as defined in Lunt et al. (2010) and the PALEOSENS paper (2012). Additionally, one of the recent studies with which our study is comparable (Krissansen-Totton and Catling, 2017; published in Nature Communications) employs the ΔT_{2x} terminology, and their ESS estimates are also from modeling work based on $S[\text{CO}_2, \text{geog}, \text{solar}]$ in the same manner as our GEOCARB estimates. Since the ESS in GEOCARB does not fit tidily into either $S[\text{CO}_2]$ or $S[\text{CO}_2, \text{geog}, \text{solar}]$, we are of course happy to defer to editorial recommendations for which notation, if any, to employ.

There are many specific revisions to the text to clear this matter up and make clear our distinction between ESS as represented within the GEOCARB model world (ΔT_{2x}) and ESS more generally. We draw attention to the following text that we have added to the Introduction (Line 96 in the revised manuscript) to make this distinction more clear.

“The GEOCARBSULFvolc model (henceforth, “GEOCARB”) and its previous incarnations^{24,25} have been widely used in previous studies [e.g., refs. 14,15,19,26,27], and includes a version of ESS where the only independent radiative forcings are CO₂, solar evolution and changing geography. In the notation of refs. 4 and 10, this ESS would be computed from the specific paleoclimate sensitivity, S[CO₂, geog, solar]. However, within GEOCARB and other such models (e.g. ref. 21), an ESS model parameter links CO₂ radiative forcing to the associated temperature response, but also accounts internally for other forcings in computing the total temperature response to radiative forcing. In GEOCARB, the ESS model parameter ΔT_{2x} corresponds to the long-term temperature change resulting from doubling CO₂ relative to preindustrial levels, accounting for changes in solar evolution and continental geography. GEOCARB assumes a linear increase in solar luminosity over time, corresponding to the parameter W_s , and uses results from general circulation model output to simulate the land temperature change resulting from changes in paleogeography (GEOG; see Supplementary Figure 1)^{28,29,30}. Thus, appropriate choices for the ESS parameter within GEOCARB are influenced by the balance of forcing between CO₂, solar luminosity, and paleogeographic changes. For brevity, we will use ΔT_{2x} when referring to ESS within the GEOCARB model, and reserve the term “ESS” for discussion of Earth system sensitivity more generally.”

L34. “are also important” should be “is also important” (the subject in the sentence is “range”, which is singular)

Thank you for catching this grammar mistake. It has been fixed in the revised manuscript.

L39. “ESS falls between 1.6 and 5.5 deg. C (95% confidence). This is well within the range of the fast-feedback climate sensitivity from other studies using climate data from the last millennium.” This is simply comparing apples with oranges. The “fast-feedback climate sensitivity” – yet another term used without explanation – in my reading refers to ECS. So ECS, in which slow feedbacks are considered forcings that are explicitly account for, is being compared here with ESS, which considered only total response versus CO₂ changes. As is clearly illustrated in palaeosens (2012), ESS values should be expected to be differ considerably from ECS values, because the latter explicitly removes the actions of slow feedbacks, whereas the former keeps those included. So when the present study sees that ESS values fall within the ranges of previous ECS values, a correction needs to be made for the slow feedback “forcings” before such a comparison even makes sense. As a result. The statement as give is meaningless. A table should be made when comparing between studies, indicating which forcings and “slow feedback “forcings” are included in the various estimates discussed. Otherwise we’re discussing all these things in a confusing vacuum, and that doesn’t help anybody.

We apologize for the confusion here. We have revised this section to only compare values for similar “flavors” of ESS. The 1.6-5.5 deg C range from Royer et al. (2007) is based on $S[\text{CO}_2, \text{geog}, \text{solar}]$ within GEOCARB. Other GEOCARB modeling studies thus, provide suitable comparisons, but of course are not independent estimates for ESS. Nevertheless, we can use previously established GEOCARB results to frame what our expected results might look like (Park and Royer, 2011), as well as the effect of time-variability/multi-state dependency of ESS (Mills et al., 2019). For comparisons not based on the GEOCARB model, we also compare to Krissansen-Totton and Catling (2017) and Farnsworth et al. (2019). All three of these studies also explicitly account for CO_2 , paleogeography and solar luminosity in their ESS calculations, similarly to GEOCARB. We also note that Anagnostou et al. (2020) indeed goes through the correction that the reviewer suggests. However, we compare our ESS estimates to the versions from these studies that explicitly account for the same forcings external to CO_2 that our GEOCARB ESS does. We have revised the text here in a few ways, and can be found at Lines 63-74 in the revised manuscript. First, we have removed the line about fast-feedback climate sensitivity. Then, we have added discussion of the specific studies mentioned above to the end of the paragraph. We discuss this further in response to your next comment. We also have included a table as requested as Supplementary Material, showing multiple studies and ESS estimates based on different forcings.

L52-55. When discussing the difference between estimates from Royer and Anagnostou, it is important to be clearer still. The implication of what is stated is the two S notations is that the Royer study explicitly considered (and accounted for) solar luminosity and geographic influence forcings, and that Anagnostou does not. When I say accounted for, this means that the Royer values were according to this notation explicitly calculating the forcing changes from these two processes, whereas Anagnostou ignored them (and thus would have found biased values). I think that’s precisely also what the authors want to say here. It would be good to spell it out a bit for non-initiated readers. It’s not an easy thing to wrap one’s head around. Again, presenting things in a table would be even clearer when comparing studies. But there is also a problem. ESS was defined to express everything in terms of CO_2 forcing only. That means that the authors now shift their ESS definition to the Royer-style approach. This may be sensible when looking over very long timescales. But it’s an ESS redefinition nonetheless. This needs to be clarified in unambiguous terms, and we also need to be shown how much that change affect comparability with previous “ESS sensu stricto” estimates (e.g., Anagnostou). Perhaps the present study could evaluate both versions of ESS and demonstrate how important it is to go with Royer’s definition when looking over very long timescales?

Thank you for pointing this out. We have removed the comparison with Anagnostou et al. (2016) because of the differences that you have noted. We have elected to discuss instead the work of Krissansen-Totton and Catling (2017) at this point in the manuscript for a couple of reasons. First, this offers a cleaner comparison of ESS with our GEOCARB results later in the paper, and their model is independent of GEOCARB but also uses $S[\text{CO}_2, \text{geog}, \text{solar}]$. Eq. 9 from Krissansen-Totton and Catling (2017)

(repeated below) gives the temperature change resulting from CO₂ forcing (pCO₂), solar luminosity (the t/228 Myr term) and paleogeography (the ΔP term):

$$\Delta T_S = \Delta T_{2x} \left(\frac{\ln(p\text{CO}_2/p\text{CO}_2^{\text{mod}})}{\ln(2)} - \frac{t}{228 \text{ Myr}} \right) + \Delta P \left(\frac{t}{100 \text{ Myr}} \right) \quad (9)$$

Comparing this with the GEOCARB temperature equation from Berner (2004) (repeated below), illustrates that ESS from their study and ours are analogous:

$$T(t) - T(0) = \Gamma \ln R\text{CO}_2 - Ws(t/570) + \text{GEOG}(t) \quad (2.28)$$

Second, discussing Krissansen-Totton and Catling (2017) here in the revised manuscript provides an opportunity to point out a quantitative example of how S[CO₂, geog, solar] ought to be greater than S[CO₂, geog, solar, land-ice] (per the glacial amplification remark by Reviewer 1). The work of Anagnostou et al. (2020) provides a good example for S[CO₂, geog, solar, land-ice] of the importance of being explicit about which forcings are (not) considered. We have added the following text to the revised manuscript at Line 63:

“For example, the geochemical model from Royer et al.¹⁹ uses a form of ESS that computes the overall global mean surface temperature response by explicitly accounting for forcings from changes in CO₂, solar luminosity, and paleogeography. In the notation of Rohling et al.⁴, this ESS is based on the specific paleoclimate sensitivity S[CO₂, geog, solar]. Krissansen-Totton and Catling²¹ also account explicitly in their model for CO₂, solar, and paleogeographic forcing over the past 100 Myr, and compute a median ESS of 5.6 °C (3.7-7.5 °C 90% credible interval). By contrast, Anagnostou et al.²² (2020) account explicitly for CO₂, solar luminosity, paleogeography, and land ice, and find ESS estimates varying from about 5-7 °C 53 Myr ago to about 2 °C 30 Myr ago. Following the argument above, we expect that the inclusion of land-ice feedbacks leads the ESS estimate of ref. 22 (based on S[CO₂, geog, solar]) to be lower than that of ref. 21 (based on S[CO₂, geog, solar, land ice]).”

L69-71. “By contrast, we quantify the influence of using both CO₂ and temperature proxy data, which allows us to highlight the impacts and ability of different sources of information to continue to improve our understanding of ESS.” This is a very cryptic sentence when looking at its grammatical coherence: “the impact ... of different sources of information to continue to improve our understanding ...” What does that even mean? Impact upon what?

We apologize for the confusion here and thank you for pointing out an opportunity to improve the clarity of the manuscript. Here, we are referring to the fact that we conduct experiments using CO₂ data and using temperature data separately, as well as together, to constrain ensembles of GEOCARB model simulations. This set-up offers a glimpse into how well each source of data constrains our model simulations and parameters (like

the ESS parameter, ΔT_{2x}). We have revised this text as follows, which appears at Line 88 in the revised manuscript.

“Here, we perform experiments in which we use CO_2 and temperature data both separately and together, to constrain ensembles of model simulations and model parameters. This experimental set-up allows us to characterize the ability of each source of information to better constrain estimates of model parameters, thereby informing our understanding of ESS.”

L81-82. This is not sufficiently precise. I would suggest that you say here “we will use S when referring to the specific version of ESS defined as S[CO₂, geog, solar] obtained from the GEOCARBSULFvolc model, and reserve the term “ESS” for discussion of Earth system sensitivity more generally, including estimates of the formal ESS definition, which is S[CO₂] (Lunt et al 2010; Palaeosens, 2012).” That would be more precise. It requires a statement earlier on that Royer’s work redefines ESS for studies over very long timescales to S[CO₂, geog, solar] instead of the previously defined S[SO₂].

Thank you for pointing out this area where we were not consistent with previously defined terminology. First, we would like to emphasize our earlier response (see page 8 of this response letter), that we do not intend to invent a new ESS, and we apologize for the confusion here. We hope that our response above can help clear up the apparent confusion. In short, neither S[CO₂] nor S[CO₂, geog, solar] is a perfect fit for the sort of ESS that is used within GEOCARB. The same is true for other deep time paleoclimate modeling studies such as Mills et al. (2019) and Krissansen-Totton and Catling (2017). Additionally, there is the notational/units inconsistency between S (specific (paleo)climate sensitivity) and ESS that we note above. In light of these issues, we have elected to adopt the notation following Krissansen-Totton and Catling (2017) as well as other previous GEOCARB modeling studies and use “ESS [parameter] within GEOCARB” or (2) ΔT_{2x} , when referring to the specific form of ESS within GEOCARB, and reserve “ESS” for discussing Earth-system sensitivity more broadly. We have included the following text in the revised manuscript to make this distinction (Line 96):

“The GEOCARBSULFvolc model (henceforth, “GEOCARB”) and its previous incarnations^{24,25} have been widely used in previous studies [e.g., refs. 14,15,19,26,27], and includes a version of ESS where the only independent radiative forcings are CO_2 , solar evolution and changing geography. In the notation of refs. 4 and 10, this ESS would be computed from the specific paleoclimate sensitivity, S[CO_2 , geog, solar]. However, within GEOCARB and other such models (e.g. ref. 21), an ESS model parameter links CO_2 radiative forcing to the associated temperature response, but also accounts internally for other forcings in computing the total temperature response to radiative forcing. In GEOCARB, the ESS model parameter ΔT_{2x} corresponds to the long-term temperature change resulting from doubling CO_2 relative to preindustrial levels, accounting for changes in solar evolution and continental geography. GEOCARB

assumes a linear increase in solar luminosity over time, corresponding to the parameter W_s , and uses results from general circulation model output to simulate the land temperature change resulting from changes in paleogeography (GEOG; see Supplementary Figure 1)^{28,29,30}. Thus, appropriate choices for the ESS parameter within GEOCARB are influenced by the balance of forcing between CO_2 , solar luminosity, and paleogeographic changes. For brevity, we will use ΔT_{2x} when referring to ESS within the GEOCARB model, and reserve the term “ESS” for discussion of Earth system sensitivity more generally.”

Paragraph in L154-163. Please check and state whether the Mills T-series is entirely independent of the GEOCARB model. I don't know, and don't have the reference at hand. It's just a simple check to ensure there is no circularity.

You are correct that some of the later figures in Mills et al. (2019) do indeed rely on the temperature functions in the GEOCARB and COPSE models. However, the temperature windows time series that we use is from Figure 4B in Mills et al. (2019). From that work:

“Here, the dark shaded area is derived from the $\delta^{18}O_{lowlat}$ record assuming either a linear or quadratic variation in $\delta^{18}O_{seawater}$ in combination with the ice line proxy for long glacial periods and the ± 1 st. dev. uncertainty in the benthic $\delta^{18}O$ record. The lighter shaded area represents the ± 1 st. dev. uncertainty in the $\delta^{18}O_{lowlat}$ record and encompasses the scatter in the TEX86 record and shorter-lived glacial periods.”

Figure 1. Very nice and helpful summary of the approach.

Thank you for the kind words, and the constructive and thorough first review, which led to the inclusion of this figure.

L211. What do you mean with CO_2 (S)?

We are referring to the parameter (S) that relates the changes in CO_2 to changes in temperature. Later in that sentence, we do the same with the solar luminosity parameter W_s , as it relates the changes in luminosity to changes in temperature. We have removed this section from the revised manuscript because we have replaced the discussion of parameter correlations with our more robust Cretaceous temperature-matching experiment (described in greater detail in our later remarks here, and in the revised manuscript at Line 300).

L210-215. The passage “The positive correlation between S and the solar luminosity parameter, W_s , seems counterintuitive at first, given that both increased CO_2 (S) and increased luminosity (W_s) should both serve to increase the temperature. However, given that the temperature at present is assumed to be fixed, higher values for W_s imply lower temperatures deeper into the past. Given that the CO_2 simulations have been calibrated to match the proxy records, to

compensate for this temperature deficit, higher values for S are required” needs to be slightly extended to allow for a somewhat slower and more elaborated discussion of this point. It’s too easy for a reader to get lost here. One question that comes to mind, for example, is: Isn’t this purely a mathematical construct – should we not be talking about underpinning processes a bit to justify that?

This is related to a comment by Reviewer 3 about how this relationship indeed arises as a result of our mathematical model for temperature change as a result of changes in CO₂ (and the ESS parameter) and the external forcings from changes in solar luminosity and paleogeography (Berner 2004, eq. 2.28). In the revised manuscript, we have removed this discussion of correlations for what we view as a more robust experiment to assess which parameters drive the model-data (mis)match in the Cretaceous.

Page 8 and subsequent. There’s a lot of talk here about “error rates.” This is an unfortunate terminology, I think, as a rate is a quantification of change with time. I don’t think you mean that w.r.t. the error. I would recommend keeping the language more simple and correct, and simply say “error margins” or something like that.

We have taken your suggestion and revised “error rate” to “error margin”. We have also changed “success rate” to “proportion of success”.

Figure 4 caption. (5% range, and 90% range. I guess you mean “probability range?” It’s important to remain precise and specific throughout. The 95% range terminology is colloquial and works well in discussions, but it’s not as precise and specific, as written word needs to be.

You are exactly correct, and thank you for pointing this out. We have updated the caption here to read “95% probability range... 90% range... 50% probability range... 95% range...”. The figure itself has been updated to show results for our modified Cretaceous temperature-matching experiment.

The discussion on Page 9 flips between percentage outbound values (25%, 30%, 50%). I think this can be written more systematically, to avoid confusion because of flipping around between percentage values. It may only need a simple opening sentence such as “first we will look at this % range, then relax it further to that % range,” or something like it. Help the reader to stay with you. It’s not an easy paper to digest.

Thank you for the good suggestion. We have moved the statement at the end of the second paragraph of this section, which gives our reasoning for relaxing the 30%-outbound threshold to the 50%-outbound threshold, to the beginning of this second paragraph. Since the first paragraph continues (from the previous section) to talk about 25%-outbound, this revision has the nice continuity that you suggest. We have also revised the text for this statement linking the two to read as follows, which has been

moved to what is now the second paragraph in the section “Controls on Cretaceous temperature biases” (Line 309 in the revised manuscript).

“In light of these biases, we perform an additional sensitivity experiment to investigate the controls on early Cretaceous (140-90 Myr ago) temperature using the GEOCARB model. First, the point of this exercise is to examine the relationship between Cretaceous temperatures and the model parameters (in particular, the ESS parameter ΔT_{2x}), so we relax the %outbound threshold from 30% to 50%. This change allows more variation in the model’s temperature simulations. Later, after making further changes to improve the goodness-of-fit in the Cretaceous temperature simulations, we tighten the error margin back to 30%, to show that the GEOCARB model is indeed quite capable of matching well the Cretaceous temperature record from Mills et al. (2019). Our initial sensitivity experiment is similar to the 50 %outbound experiment from our main set of simulations, where the ensemble for analysis consists only of simulations that match the CO₂ temperature data windows in at least 50% of the time steps. In our new experiment, however, we retain only those simulations that pass through the temperature data window at 90 Myr ago.”

L302-303. GYM -> 0.25 GYM. That equation has an internal conflict. I guess you mean GYM_new -> 0.25 GYM_previous or something like that. Please update to make the writing specific and precise.

Thank you for the suggestion. This experiment has been removed from the revised manuscript.

L328-329. “We find that a four-fold reduction in gymnosperm weathering efficiency leads to a much better match with the paleotemperature data (Fig. 5d).” This is another example of sloppy, colloquial writing. Gymnosperm weathering efficiency mean the efficiency of weathering of gymnosperms. But you don’t mean that. You mean the efficiency of weathering MEDIATED BY gymnosperms. You are not writing only for people who know the model and what you’re doing. You need to be very precise and specific to say things exactly in the way that you mean them. This means that you cannot use colloquialisms as you would in discussions in the lab. You need to be much more specific and precise. Also, it would be good if somewhere in the paper you would explain to the readers what makes the difference between gymnosperm and angiosperm mediated weathering.

More importantly, I know quite well what you are doing and why, but still am left wondering if you mean here that it really is the efficiency of the weathering process by gymnosperms that reduces (i.e., you keep the same gymnosperm coverage, but for whatever reason they don’t affect weathering to a lesser extent), or that the gymnosperm coverage itself drops and thus the weathering intensity that they drive (even though the weathering efficiency by each gymnosperm may have remained constant). The writing is simply too imprecise for readers to understand these subtleties. This lack of precision and specificity really needs to be fixed. I

suggest that senior authors go through next versions much more seriously and critically than they apparently have done.

Thank you for the great points. We apologize for the confusion. We, of course, are referring to *mediating* the weathering, as opposed to our original wording, which implied causing, or weathering of, gymnosperms. We have removed this experiment from the revised manuscript. In its place, there is a more thorough new calibration experiment forcing the GEOCARB model simulations to agree during the Cretaceous period.

L365-369. This about what you're saying here from the point of view of a non-specialist who is very interested in your work. Do you think this is sufficiently elaborated, specific and precise to make your point convincingly? I would argue that it isn't. This bit needs work in my view.

Thank you for highlighting this. We agree, and the revised paragraph now reads (at Line 381 in the revised manuscript):

“We adopt a well-studied, state-of-the-art, yet still relatively simple model. This model simplicity provides the advantages of transparency and the ability to perform careful and exhaustive uncertainty and sensitivity analyses³⁸. These advantages come, however, with several caveats that point to fruitful research directions. One key caveat stems from the fact that GEOCARB is a coarse-resolution and highly parameterized model with a long (10-Myr) time step and many (68) parameters (including 12 time series). A second related caveat arises from the still highly stylized representation of feedbacks and processes that is characteristic of such models (e.g., refs. 21,31). As previously discussed (e.g., refs. 12,22,34), the current assumption in the model of using a constant ΔT_{2x} for each of the glacial and non-glacial stable climate states risks missing processes leading to gradual changes in ΔT_{2x} within one of the larger stable climate states. The work of ref. 12 further points to the potential importance of capturing this Type I state dependence in ΔT_{2x} because their results indicate an increasing trend in ΔT_{2x} beginning about 130 Myr ago. In the GEOCARB model, however, the ΔT_{2x} ESS parameter is assumed to be constant at its non-glacial value from 260 to 40 Myr ago, then shifts immediately to its glacial value from 40 to 0 Myr ago. We evaluate the impacts of this Type I state dependence in an experiment where we linearly increase ΔT_{2x} from its non-glacial value 130 Myr ago to its glacial value 40 Myr ago; the parameter remains constant at its glacial value from 40 to 0 Myr ago. This linear change in ΔT_{2x} (as opposed to the step function transitions in the base-case version of the model) has little effect on the temperature hindcast (Supplementary Figure 6). This simple experiment, of course, scratches only the surface of the challenge to represent Type I state dependence for ESS. This result suggests, however, that a simple refinement of Type I state dependency does not substantially impact our results.”

L371. Insert “of” between “time series” and “forcing.”

This has been updated in the revised manuscript.

Overall: getting there, but please work on accessibility for non-specialists, by elaborating relevant points, presenting and working with established definitions/terms, avoiding colloquialisms, and making the writing more specific and precise. It will gain you many, many citations because people will understand what you're doing and discuss the subtleties in your work. Use your experienced co-authors. Make them work for the privilege, and don't put the burden of trying to clarify how you're saying things on reviewers. In other words: it's a great study, so present it more carefully so that people will see and appreciate it for what it is, rather than feeling confused.

Again, many thanks for the continued thorough and constructive feedback.

Reviewer #3 (Remarks to the Author):

This is a completely different paper than the one I first reviewed. It is, to be clear, a better one, but it does require almost a de novo review.

The clarity of the exposition is much better, and the results presented (including in the supplemental material) are far more comprehensive and appropriate. I think that the authors are still overconfident in their results, but given the new presentation it is much easier to see (and say) why.

The overall methodology is sound, and I think these results are interesting. I would however suggest that more revisions are required before publication.

Thank you for the constructive feedback, and we very much appreciate you taking the time to do nearly an entirely new review. Your comments have certainly helped to improve the work itself, as well as our descriptions/discussion of the work.

detailed critiques:

I 9. "implying increased learning" or overconfidence? Maybe delete.

This has been removed from the updated manuscript.

I 9-14. The sensitivity analyses to deal with the clear mismatches in the Cretaceous are neither exhaustive, nor particularly convincing. I personally would not elevate this to the lead paragraph. Rather I would describe the caveats to the uncertainty estimate given above. I expand on this further below.

Thank you for making this suggestion. We have revised the Cretaceous experiments to start with a new full precalibration experiment. This is described in greater detail in response to your specific points below.

I 25. It's only just come out, but the WCRP assessment on climate sensitivity (Sherwood et al, 2020, doi: 10.1029/2019RG000678) is an excellent reference for the state of the art in ECS estimation.

Thank you for pointing out this important reference. We have added it to the Introduction of the revised manuscript at Line 27 in the revised manuscript:

“Based on the understanding of feedback processes, historical climate and paleoclimate records, Sherwood et al.⁹ estimate the most likely range (66% confidence) for the effective sensitivity (defined in terms of the 150-year temperature response to a quadrupling of CO₂ forcing) is 2.6 to 3.9 °C. Similar to the ECS, the effective sensitivity does not include the long-term feedback such as ice sheets, vegetation and carbon cycle (ref. 9 and references therein).

In contrast to the shorter-term ECS that responds to relatively fast feedback processes, consideration of longer-term responses offers a glimpse into the deep-time paleoclimate evolution of the sensitivity of the Earth-system temperature response to both fast and slow feedbacks. In particular, a deep-time perspective offers insight into the “Earth-system sensitivity” (ESS) - the long-term equilibrium surface temperature response to a given CO₂ forcing, including all Earth-system feedbacks¹⁰. Sherwood et al. estimate the ESS as their effective sensitivity multiplied by an inflation factor, $(1+f_{ESS})$, where f_{ESS} is sampled from a normal distribution with mean value of 0.5 and standard deviation of 0.25^{10,11}.”

I 30. ESS is predicated on fixed atmospheric CO₂ concentration. It does not depend on carbon burial rates. (Again).

We apologize for the confusion here. Reviewer 2 makes the point:

“ESS DOES include all that [carbon burial] – it wraps everything up in terms of a total response to CO₂ changes only.”

You are of course correct - ESS is the *response to* CO₂ forcing; it does not depend on the forcing itself. In light of Reviewer 2's point, we believe that the confusion here is that we meant this to be a discussion of the processes that are considered to be internal versus external forcings, in the sense of the Paleosens/Lunt et al. (2010) definition for ESS. In light of the comments by both Reviewers, we have removed the note about carbon burial and modified the text at Line 34 in the revised manuscript to read:

“In contrast to the shorter-term ECS that responds to relatively fast feedback processes, consideration of longer-term responses offers a glimpse into the deep-time paleoclimate evolution of the sensitivity of the Earth-system temperature response to both fast and slow feedbacks. In particular, a deep-time perspective offers insight into the “Earth-system sensitivity” (ESS) - the long-term equilibrium surface temperature response to a given CO₂ forcing, including all Earth-system feedbacks¹⁰. Sherwood et al. estimate the ESS as their effective sensitivity multiplied by an inflation factor, (1+f_{ESS}), where f_{ESS} is sampled from a normal distribution with mean value of 0.5 and standard deviation of 0.25^{10,11}.”

I. 79. "resulting *solely* from paleogeography"...(does this include ice sheets?), and do the results really rely on 25 year old model results? (O-B, 1995)? Models have changed a lot since then - that model was very coarse resolution and only had a 50m mixed layer ocean! Talk to Paul Valdes and Dan Lunt for more up-to-date numbers.

The paleogeographical effects are not assumed to include ice sheets. We apologize for the confusion but thank you for noting this regrettable writing error on our part - the forcing is from Godderis et al. (2012, 2014) and was first incorporated into the GEOCARB family of models by Royer et al. (2014). We have added the word “solely” and updated the references at Line 107 in the revised manuscript.

Godd ris, Y., Donnadi u, Y., Lefebvre, V., Le Hir, G., and Nardin, E., 2012, Tectonic control of continental weathering, atmospheric CO₂, and climate over Phanerozoic times: *Comptes Rendus Geoscience*, v. 344, p. 652– 662, <http://dx.doi.org/10.1016/j.crte.2012.08.009>

Godd ris, Y., Donnadi u, Y., Le Hir, G., Lefebvre, V., and Nardin, E., 2014, The role of palaeogeography in the Phanerozoic history of atmospheric CO₂ and climate: *Earth-Science Reviews*, v. 128, p. 122–138, <http://dx.doi.org/10.1016/j.earscirev.2013.11.004>

Royer, D. L., Donnadi u, Y., Park, J., Kowalczyk, J., & Godd ris, Y. (2014). Error analysis of CO₂ and O₂ estimates from the long-term geochemical model GEOCARBSULF. *American Journal of Science*, 314(9), 1259–1283. <https://doi.org/10.2475/09.2014.01>

I. 174 Figure 1. This probably belongs in the Supp. Material.

In light of the comments from Reviewer 2, particularly during the first round of review, we would like to keep this schematic of the overall workflow in the main text. We are of course happy to follow the editorial team’s guidance on this issue.

I 211. a correlation btw W_s and S is not at all surprising, but I think the authors would have been better building this into the model at the beginning. Specifically, the temperature response to increasing solar should not be that different to the temperature response to increasing CO_2 - both are forcings and both impact similar feedbacks. Thus W_s should be $w \cdot S$, where w would be an efficacy related to the potentially different response of the system to 4 W/m² of solar forcing compared to CO_2 forcing. The value of w would be constrained to be 0.5 to 1.5 (times whatever normalisation might be used with W_s - see point below about the Supp. data table - is it per 570My?).

A quick back of the envelope calculation shows the need. The magnitude of the solar forcing from 420 Myr, is around 5 to 6 W/m² (assuming that the sun has become 30% more active over the last 4.5 billion years) (i.e. ~30 W/m² change in solar output at 1 AU), which is equivalent to ~ 3 x CO_2 forcing, and so the temperature change W_s (assuming that is in K per 570Myr), would be ~7.3 K for the 'best guess' S of 3.4 K/2x CO_2 - close to the mean constrained value. Setting this up as part of the model will reduce the uncertainties.

You are quite correct here, and we apologize for the confusion. A correlation between W_s (solar forcing) and ESS should not be surprising at all. In fact, Equation 2.28 from Berner (2004) (provided below) gives the relationship between temperature changes, ESS, and solar forcing (W_s):

$$T(t) - T(0) = \Gamma \ln RCO_2 - W_s(t / 570) + GEOG(t) \quad (2.28)$$

Because the overall temperature change relative to present ($t=0$), $T(t)-T(0)$, and the CO_2 trajectory (RCO_2) are both constrained by matching the data, a three-dimensional relationship between ESS (which is a scaled version of Γ), W_s and GEOG (the temperature change driven by changes in paleogeography) must arise as a result of how GEOCARB is set up as a mathematical model.

We suspect that the confusion here has arisen because our description of the correlation was meant to walk the reader to the fact that higher values of the W_s parameter actually serve to decrease solar luminosity deeper into the past. However, in the revised version of the manuscript, we have removed the section on correlations to make space for the new Cretaceous temperature-matching experiment. As a result, this section has been removed.

I 213. "assumed to be fixed"  "is known".

Thank you for this suggestion. However, this section has been removed from the revised manuscript (in our replacement of the correlations examination by the improved Cretaceous temperature-matching experiment, described in greater detail below).

I. 271 onwards. The authors are correct in highlighting the clear mismatch between the baseline cases shown in Figure 4 and the inferred Cretaceous temperatures. This mismatch is not small (up to 20 deg C!!). However, their sensitivity studies are not that informative. It's useful to realise that ACT and GYM go some way to reducing the error, but the improvement is minor and the mismatch is still clear in Figure 5h. Thus the authors need to go further - if the model cannot match the obs, then either the model is wrong or the obs are biased. The evidence for a very warm Cretaceous is pretty strong, so this points to a structural problem in the model. What could it be? CH₄? higher atmospheric pressure? an error in the GEOG estimates? I don't really know, but this should be explored by the authors in addition to the GYM/ACT variation.

Thank you for the constructive comment. We have decided to replace the examination of parameter correlations with a more thorough experiment to determine the underlying factors that are responsible for the model-data mismatch in temperature during the Cretaceous period. Our new experiment is similar to the 50%-outbound experiment from our main set of simulations, where the ensemble for analysis consists only of simulations that match the temperature and the CO₂ data windows in at least 50% of the time steps. In our new experiment, however, we retain only those simulations that pass through the temperature data window at 90 Myr ago.

In the new "Cretaceous-matching" experimental results, the estimated distribution for ESS shifts down slightly, by only about 0.1 deg C, relative to the original results for the 50%-outbound experiment. One of the benefits of this sort of analysis is that it enables us to examine how the distributions of the model parameters have changed, relative to the original experiment. We do not find any substantial changes in any of the 56 constant parameters, but several of the time series parameters' values do display notable changes. Specifically, the time series for the land area relative to present (f_A), global river runoff relative to present (f_D), the response of temperature change on river runoff (RT), and the fraction of land area that undergoes chemical weathering relative to present (f_{AW}/f_A). In the figure below (the new Supplementary Figure 5), we show the mean for each time series among the control simulations in black and the means from the Cretaceous-matching experiment in red. The decreases in f_A , f_D and RT at 90 Mya are perhaps plausible, but the decrease in f_{AW}/f_A (weatherable land surface area) is, in our view, less convincing.

The purpose of this experiment, however, is to assess the degree to which our ESS estimate may (not) be trusted, based on potential model temperature biases, and whether these biases indicate model shortcomings in GEOCARB.

So we generate a new set of 10,000 simulations that all match the Cretaceous temperatures, by changing the centers of the time series parameters' multivariate normal distributions to match the mean time series shown in the figure below. We now use the 30%-outbound threshold because we want to assess how much our best ESS estimates (the 30%-outbound experiments) are biased by the Cretaceous temperatures, and to

improve this estimate if possible. To retain only the plausible simulations in our experiment, we removed any simulations where the value for the f_{Aw_fA} time series at 90 Mya was more than 1 standard deviation away from the *original* central value. This leaves 2,139 simulations out of the original 10,000. This updated set of Cretaceous-matching simulations has a median ESS of 3.3 deg C, as compared to 3.4 deg C in the original set of experiments. The 5-95% range also shifts about 0.1-0.2 deg C cooler at 2.5-4.5 deg C, as compared to 2.6-4.7 deg C in the original experiments.

These results address the Cretaceous temperature bias and have not permitted the time series parameters to stray too far from their original central estimates. From the fact that the distribution of estimated ESS changes by less than 0.2 deg C, we conclude that our estimates of ESS are not unduly influenced by the biases in the temperature simulation. Further, we conclude that GEOCARB is in fact capable of reproducing the temperature data, although sampling via brute force appears to require a very large number of samples and some statistical care in order to bring the modeled and observed temperatures into better agreement.

Supplementary data table. This table really needs units. For instance, the mean of W_s is 7.4. Is that K? F? K per some solar luminosity change? K per billion years? Makes a big difference! (I think I can infer from the code that it is K per 570Myr, but I'm not sure that would be obvious to most readers!

We apologize for this oversight. We have added units to the supplemental table of parameter descriptions.

References

Anagnostou, E., John, E. H., Babila, T. L., Sexton, P. F., Ridgwell, A., Lunt, D. J., Pearson, P. N., Chalk, T. B., Pancost, R. D., & Foster, G. L. (2020). Proxy evidence for state-dependence of climate sensitivity in the Eocene greenhouse. *Nature Communications*.
<https://doi.org/10.1038/s41467-020-17887-x>

Berner, R. A. (2004). *The Phanerozoic carbon cycle: CO₂ and O₂*. New York, NY, USA: Oxford University Press.

Goddéris, Y., Donnadiou, Y., Lefebvre, V., Le Hir, G., and Nardin, E., 2012, Tectonic control of continental weathering, atmospheric CO₂, and climate over Phanerozoic times: *Comptes Rendus Geoscience*, v. 344, p. 652– 662, <http://dx.doi.org/10.1016/j.crte.2012.08.009>

Goddéris, Y., Donnadiou, Y., Le Hir, G., Lefebvre, V., and Nardin, E., 2014, The role of palaeogeography in the Phanerozoic history of atmospheric CO₂ and climate: *Earth-Science Reviews*, v. 128, p. 122–138, <http://dx.doi.org/10.1016/j.earscirev.2013.11.004>

Krissansen-Totton, J., & Catling, D. C. (2017). Constraining climate sensitivity and continental versus seafloor weathering using an inverse geological carbon cycle model. *Nature Communications*, 8, 15423. <https://doi.org/10.1038/ncomms15423>

Langlois, R. N., & Cosgel, M. M. (1993). Frank Knight on Risk, Uncertainty, and the Firm: A New Interpretation. *Economic Inquiry*, 31(3), 456–465.
<https://doi.org/10.1111/j.1465-7295.1993.tb01305.x>

Lenton, T. M., Daines, S. J., & Mills, B. J. W. (2018). COPSE reloaded: An improved model of biogeochemical cycling over Phanerozoic time. *Earth-Science Reviews*, 178(December 2017), 1–28. <https://doi.org/10.1016/j.earscirev.2017.12.004>

Lunt, D. J., Haywood, A. M., Schmidt, G. A., Salzmann, U., Valdes, P. J., & Dowsett, H. J. (2010). Earth system sensitivity inferred from Pliocene modelling and data. *Nature Geoscience*, 3, 60–64. <https://doi.org/10.1038/ngeo706>

Mills, B. J. W., Krause, A. J., Scotese, C. R., Hill, D. J., Shields, G. A., & Lenton, T. M. (2019). Modelling the long-term carbon cycle, atmospheric CO₂, and Earth surface temperature from late Neoproterozoic to present day. *Gondwana Research*, 67, 172–186. <https://doi.org/10.1016/j.gr.2018.12.001>

Rohling, E. J., and others. (2012). Making sense of palaeoclimate sensitivity. *Nature*, 491, 683–691. <https://doi.org/10.1038/nature11574>

Royer, D. L., Donnadieu, Y., Park, J., Kowalczyk, J., & Godd ris, Y. (2014). Error analysis of CO₂ and O₂ estimates from the long-term geochemical model GEOCARBSULF. *American Journal of Science*, 314(9), 1259–1283. <https://doi.org/10.2475/09.2014.01>

Tierney, J. E. et al. (2020). Past climates inform our future. *Science*, 370(6517), eaay3701. <https://doi.org/10.1126/science.aay3701>

Walker, W. E., Haasnoot, M., & Kwakkel, J. H. (2013). Adapt or perish: A review of planning approaches for adaptation under deep uncertainty. *Sustainability (Switzerland)*, 5(3), 955–979. <https://doi.org/10.3390/su5030955>

Reviewer comments, third round -

Reviewer #1 (Remarks to the Author):

This version of the draft is fine with me. I have only 2 technical comments:

1)

line 29: „effective sensitivity (defined in terms of the 150-year ^{ISEPI} temperature response to a quadrupling of CO2 forcing),„

This is certainly only the description of the GCM experiments in Sherwood et al (2020) from which feedback processes have been analysed, but not for approach analysing historical and paleo data. Please revise.

2)

A lot of the text of the Fig 1 caption not only is repeated in the figure itself, but also in the main text, lines 202ff. I therefore suggest to shorten, potentially by reducing the caption of Fig 1 to something like „Schematic of the precalibration workflow.“

Reviewer #2 (Remarks to the Author):

First of all, I apologise for the delay.

I have now studied the revised files and the response letter, and I can state that I am fully satisfied with the manuscript now. It has sorted out all ambivalences, and makes a really strong and valid set of conclusions that are thoroughly borne out by the work presented.

One can always continue going on about tiny details, but this is a very solid piece of work now that is carefully presented in a (to me) crystal clear manner. I suggest acceptance as is.

Reviewer #3 (Remarks to the Author):

This too is a quite different paper than either version 1 or 2. But it is much improved, and I am impressed at the efforts made to address the valid points raised in prior reviews.

I would like to reiterate the one model specification that I think could be improved but otherwise I only have minor editorial comments.

Temperature responses to long-term solar and CO2 should be explicitly linked since the feedbacks that control ECS and ESS are not substantially different for solar forcing or for GHG forcing. However the variance in W_s and ΔT_{2x} is still uncoupled in this estimation. The authors should be strongly encouraged to rewrite the solar term in the model and replace W_s with $\Delta T_{2x} * S_{eff} * S_c$ where S_{eff} is term that could encompass the efficacy of solar forcing vis a vis CO2, and S_c is a suitable scale that translates time through the Phanerozoic to a normalized radiative forcing. My point here is not based on a misreading of the text, but deals with the specification of the model. If this isn't done for this paper, it should be fixed in future work. Even if this has persisted since Berner's original paper, it still could be improved!

line 412-414: It clearly can't be true that no uncertainties or model imperfections will ever impact the constrained ESS, so the line should read more like "ESS was seen to be robust to these variations associated with improving the Cretaceous temperatures".

Review responses

Reviewer #1 (Remarks to the Author):

This version of the draft is fine with me. I have only 2 technical comments:

We thank you for your continued constructive feedback and for your service to our community.

1)

line 29: „effective sensitivity (defined in terms of the 150-year temperature response to a quadrupling of CO₂ forcing),„

This is certainly only the description of the GCM experiments in Sherwood et al (2020) from which feedback processes have been analysed, but not for approach analysing historical and paleo data. Please revise.

We have updated this text to read as follows (italics denotes the added text):

“Based on the understanding of feedback processes, historical climate and paleoclimate records, a recent summary by Sherwood et al.⁹ concluded that the most likely range (66% confidence) for the effective sensitivity (defined in terms of the 150-year temperature response to a quadrupling of CO₂ forcing *in the context of their general circulation model experiments*) is 2.6 to 3.9 °C.”

2)

A lot of the text of the Fig 1 caption not only is repeated in the figure itself, but also in the main text, lines 202ff. I therefore suggest to shorten, potentially by reducing the caption of Fig 1 to something like „Schematic of the precalibration workflow.“

Thank you for this suggestion. We agree that the caption provides some redundant information. We have shortened the caption to read:

“Schematic of the precalibration workflow to produce ensembles by varying the %outbound threshold and the data sets employed.”

Reviewer #2 (Remarks to the Author):

First of all, I apologise for the delay.

I have now studied the revised files and the response letter, and I can state that I am fully satisfied with the manuscript now. It has sorted out all ambivalences, and makes a really strong and valid set of conclusions that are thoroughly borne out by the work presented.

One can always continue going on about tiny details, but this is a very solid piece of work now that is carefully presented in a (to me) crystal clear manner. I suggest acceptance as is.

We thank you for your many detailed and constructive comments throughout the review process, and for your service to our community.

Reviewer #3 (Remarks to the Author):

This too is a quite different paper than either version 1 or 2. But it is much improved, and I am impressed at the efforts made to address the valid points raised in prior reviews.

Thank you for your constructive comments and helpful feedback throughout the review process, and for your service to our community.

I would like to reiterate the one model specification that I think could be improved but otherwise I only have minor editorial comments.

Temperature responses to long-term solar and CO₂ should be explicitly linked since the feedbacks that control ECS and ESS are not substantially different for solar forcing or for GHG forcing. However the variance in W_s and ΔT_{2x} is still uncoupled in this estimation. The authors should be strongly encouraged to rewrite the solar term in the model and replace W_s with $\Delta T_{2x} \cdot S_{eff} \cdot S_c$ where S_{eff} is term that could encompass the efficacy of solar forcing vis a vis CO₂, and S_c is a suitable scale that translates time through the Phanerozoic to a normalized radiative forcing. My point here is not based on a misreading of the text, but deals with the specification of the model. If this isn't done for this paper, it should be fixed in future work. Even if this has persisted since Berner's original paper, it still could be improved!

Thank you for raising this important point. We completely agree that most models will never truly be “done”! You are quite right that exploring the correlations and dependencies among the model parameters is an important area for future research.

line 412-414: It clearly can't be true that no uncertainties or model imperfections will ever impact the constrained ESS, so the line should read more like "ESS was seen to be robust to these variations associated with improving the Cretaceous temperatures".

Thank you for this good suggestion. We have revised this sentence to read:

“However, our experiment examining the Cretaceous cool temperature bias suggests that our estimates of ESS are robust to these variations associated with improving the Cretaceous temperatures.”